# Airborne measurement of peroxy radicals using chemical amplification coupled with cavity ring down spectroscopy: the PeRCEAS instrument

[1]Midhun George, [1]Maria Dolores Andrés Hernández, [2]Vladyslav Nenakhov, [1]Yangzhuoran Liu, and [1]John Philip Burrows

[1] Institute of Environmental Physics, University of Bremen, Germany
[2] now at Flight Experiments, DLR Oberpfaffenhoffen, Germany

Correspondence to M. George (midhun@iup.physik.uni-bremen.de) and M.D. Andrés Hernández (lola@iup.physik.uni-bremen.de).

**Abstract.** Hydroperoxyl, $HO_2$, and organic peroxy, $RO_2$, radicals are reactive short-lived species, which play a key role in tropospheric chemistry. Measurements of the sum of $HO_2$ and $RO_2$ provide unique information about the chemical processing in an air mass. This paper describes the experimental features and capabilities of the Peroxy Radical Chemical Enhancement and Absorption Spectrometer (PeRCEAS). This is an instrument designed to make measurements on aircraft from the boundary layer to the lower stratosphere. PeRCEAS combines the amplified conversion of peroxy radicals to nitrogen dioxide, $NO_2$, with the sensitive detection of $NO_2$ using absorption spectroscopy (Cavity Ring Down, CRD) at 408 nm. PeRCEAS is a dual channel instrument, with two identical reactor-detector lines working out of phase with one another at a constant and defined pressure lower than ambient at the aircraft altitude. The suitability of PeRCEAS for airborne measurements in the free troposphere was evaluated by extensive characterisation and calibration under atmospherically representative conditions in the laboratory. The use of alternating modes of the two instrumental channels successfully captures short-term variations in the sum of peroxy radicals, defined as $RO_2^*$ ($RO_2^* = HO_2 + \sum RO_2 + OH + \sum RO$, being R an organic chain) in ambient air. For a 60s measurement, the $RO_2^*$ detection limit for a minimum ($2\sigma$) $NO_2$ detectable mixing ratio < 60 pptv, is < 2 pptv under laboratory conditions in the range of atmospheric pressures and temperatures expected in the free troposphere. PeRCEAS has been successfully deployed within the missions OMO (Oxidation Mechanism Observations) and EMeRGe (Effect of Megacities on the transport and transformation of pollutants on the Regional and Global scales) in different airborne campaigns onboard the High Altitude LOng range research aircraft (HALO) for the study of the composition of the free troposphere.

## 1. Introduction

Hydroperoxyl, $HO_2$, and organic peroxy, $RO_2$, radicals, having an unpaired spin are highly reactive free radicals. They play important roles in the tropospheric chemistry. During the day they are formed in the atmosphere following the oxidation of carbon monoxide, CO, methane, $CH_4$, and many volatile organic compounds, VOCs. They participate in catalytic cycles, which produce and destroy ozone, $O_3$. Their temperature dependent reactions form temporary reservoirs (e.g. peroxynitrates such as peroxyacetylnitrate, PAN, $CH_3COO_2NO_2$), which are transported in the troposphere. In the presence of sufficient $NO_x$ (nitrogen monoxide, NO, + nitrogen dioxide, $NO_2$), the reaction of $HO_2$ with NO forms $NO_2$ and hydroxyl radical, OH, which is the most important tropospheric oxidising agent. The organic-oxy radicals RO, which contain hydrogen atoms, often react with molecular oxygen, $O_2$, to form $HO_2$ and oxygenated volatile organic compounds, OVOC such as aldehydes and ketones. The latter are oxidised by OH and photolysed to ultimately produce $HO_2$ and $RO_2$.

Overall $HO_2$ and $RO_2$ influence the amounts and distributions of OH and $O_3$ and thus the oxidising capacity of the troposphere. Consequently, knowledge about the spatial distribution and concentration of $HO_2$ and $RO_2$ is essential to test our understanding of the tropospheric chemistry.

The $HO_2$ and $RO_2$ concentrations and mixing ratios are small because of their high reactivity. Consequently, their measurement requires sensitive and accurate techniques. With the exception of the freezing of air and subsequent use of the matrix isolation electron spin resonance technique (Mihelcic et al., 1985), there are no direct spectroscopic measurements of $HO_2$ or $RO_2$, which have been applied successfully in air. Alternatively, indirect measurement techniques have been developed. The chemical amplification technique (Cantrell and Stedman, 1982; Hastie et al., 1991) has been used to measure the sum of peroxy radicals. The Peroxy Radical Chemical Amplification (PeRCA) converts by addition of NO and CO, $HO_2$ and most atmospherically significant $RO_2$ to $NO_2$. The OH formed in the reaction cell reacts with CO to reform $HO_2$ in a chain reaction. Oxy, alkoxy, hydroxy and alkylperoxy radicals (OH + $\sum$RO + $HO_2$ + $\sum$RO$_2$) are converted into $NO_2$. As the RO and OH abundances in the troposphere are much lower than those of $HO_2$ and $RO_2$, PeRCA measures to a good approximation the sum of peroxy radicals collectively known as $RO_2^*$, ($RO_2^* = HO_2 + \Sigma\ RO_2$, being R any organic chain), which convert NO to $NO_2$. The rate coefficients of the $HO_2$ and $RO_2$ reactions with NO are very similar (Lightfoot et al, 1993). Large $RO_2$ which do not react with NO to form $NO_2$ are not detected, and are assumed to be negligibly small compared to the sum of $HO_2 + \sum RO_2$ concentrations. $HO_2$ and $CH_3O_2$ are the dominant peroxy radicals present in an air mass in most conditions.

A variant on the CO chemical amplification is used in the Ethane Chemical AMPlifier (ECHAMP). As its name implies this uses ethane, $C_2H_6$, rather than CO for the amplification of atmospheric peroxy radicals (Wood et al., 2017). Although the amplification is lower than for CO, the chain length appears to be less sensitive to humidity effects (Duncianu et al., 2020). Chemical amplification using a CO and $SO_2$ chain conversion in combination with chemical ionization mass spectrometry (CIMS) has been used for the measurement of $RO_2^*$ (Reiner et al., 1997; Hanke et al., 2002). In a further development, Edwards et al., (2003) and Hornbrook et al., (2011), described a PerCIMS instrument with two measurement modes, $HO_2$ and $\Sigma RO_2$. The separation is achieved by varying NO, $SO_2$ and $O_2$ concentrations, which changes the relative sensitivities to $HO_2$ and $RO_2$. Recently, the use of iodide and bromide as primary ions in CIMS for the measurement of $HO_2$ has been reported (Sanchez et al., 2016; Albrecht et al., 2019). Further investigation on the instrumental background signal is required before deploying this technique in the field.

$HO_2$ has also been successfully measured by the conversion of $HO_2$ to OH, which is then measured by laser induced fluorescence, LIF. The technique, also known as Fluorescence Assay by Gas Expansion, FAGE, was pioneered by Hard et al., (1984), and further modified by several scientific groups (Creasy et al., 1997; Kanaya et al., 1999; Holland et al., 2003, Faloona et al., 2004). Potential spectral and chemical interferences have been investigated in detail (Ren et al., 2004). The interference by some $RO_2$ radicals into the $HO_2$ signal reported by Fuchs et al., (2010, 2011) is minimised by controlling the NO concentration added for conversion into OH (Whalley et al., 2013; Lew et al., 2018).

In the last decades, ground based measurements of $RO_2^*$ and $HO_2$ have been successfully made in a variety of environments (Monks et al, 1998, 2009 and references herein; Burkert et al., 2001 a and b; Carslaw et al., 2002, Fleming et al., 2006 a and b; Emmerson et al., 2007; Qui et al., 2007; Kanaya et al., 2007, 2012; Hofzumahaus et al., 2009; Andrés-Hernández et al., 2009; 2010, Mao et al., 2010, Kukui et al., 2014; Lelieveld et al., 2018). The majority of measurements of $RO_2^*$ or $HO_2$ were made in field campaigns, which studied different aspects of the chemistry in the lower troposphere. These case studies have improved

considerably our knowledge of the role of $HO_2$ and $RO_2$ in tropospheric boundary layer chemistry. In contrast, the number of unequivocal measurements of peroxy radicals in the free troposphere is still quite limited.

Airborne measurements offer a unique opportunity to measure $HO_2$ and $RO_2$ in the free troposphere. However, the temporal and spatial variability in the chemical composition of the air masses make the measurement from airborne platforms challenging. High instrumental accuracy, sensitivity and specificity are required to unequivocally identify and quantify potential spectral and chemical interferences (see Green et al., 2003, 2006; Zanis et al., 2003; Clemitshaw, 2004 and references herein; Heard 2006, and references herein, Stone et al., 2012 and references herein; Ren et al., 2012). In addition, each particular airborne platform has unique capabilities but also limitations (e.g. mechanical, electrical and safety constraints) compared to ground based or ship board. As a result of the above, instruments to measure airborne $HO_2$ and $RO_2^*$ are usually designed and optimised for use on a specific aircraft platform.

The PeRCEAS (Peroxy Radical Chemical Enhancement and Absorption Spectrometer) instrument was designed by the Institute of Environmental Physics of the University of Bremen (IUP-UB) for the airborne measurement of $RO_2^*$ in the free troposphere and lower stratosphere and for its deployment on board the HALO (High Altitude LOng range) research aircraft (www.halo.dlr.de). PeRCEAS combines the **PeRCA** and the **CRDS** (Cavity Ring Down Spectroscopy) techniques in a dual channel instrument for the determination of $RO_2^*$. The principle of these well-known techniques and their application to the $RO_2^*$ measurement have been described in detail in a previous publication (Hortsjann et al., 2014). In an instrument using the PeRCA technique, the probed $RO_2^*$ are converted into an amplified amount of $NO_2$ by adding NO and CO in excess to the sampled air in the inlet. A modulated $NO_2$ signal is obtained by alternating the position of the CO addition between the so called amplified or amplification and non-amplified or background modes. These modes respectively facilitate or suppress the conversion of radicals into $NO_2$. Sampled $O_3$ is converted to $NO_2$ in the reactor. The background signal comprises the sum of ambient $NO_2$, $O_3$, and any $NO_2$ produced within the system (e.g. from the thermal decomposition of peroxyacyl nitrates like PAN). The instrumental amplification factor, the so called effective chain length ($eCL = RO_2^*/\Delta NO_2$; where $\Delta NO_2$ is the $NO_2$ formed by the chemical amplification) depends on loss of peroxy radicals during passage through the inlet and the termination chemical reactions or physical processes resulting in non-radical products. As a result, the specific instrumental characteristics and the measurement conditions determine the eCL (Cantrell and Stedman, 1982; Cantrell et al., 1984, 1996; Hastie et al., 1991; Clemitshaw et al., 1997, Kartal et al., 2010).

In CRDS (O'Keefe and Deacon, 1988; Atkinson, 2003; Brown, 2003; Berden and Engel, 2010 and references herein), which is now a well-established spectroscopic measurement technique, a monochromatic electromagnetic radiation pulse is trapped inside a high finesse optical cavity and the time decay of the intensity is measured. The concentration of an absorber of interest is calculated from the decay times of the electromagnetic radiation pulse to $1/e^{th}$ of its initial value, the so-called ring down time, for a resonator containing ($\tau$) or not containing ($\tau_0$) the absorber. In PeRCEAS the absorber of interest is $NO_2$ which is formed in both the amplification and the background modes.

The ambient $RO_2^*$ concentrations measured by PeRCEAS are then retrieved from the difference in the ring down times of the background and amplification modes of operation, provided that $\tau_0$ and the total scattering do not change substantially during two consecutive sampling modes:

$$\Delta\alpha = \alpha_2 - \alpha_1 = \frac{n}{c_o}\left(\frac{1}{\tau_2} - \frac{1}{\tau_1}\right) = \sigma_{NO_2}\,\Delta[NO_2] = \sigma_{NO_2}\,[RO_2^*] \times eCL \qquad \text{(Eq.1)}$$

where $\alpha_1, \tau_1$, and $\alpha_2, \tau_2$ are, respectively, the absorption coefficients and ring down times for the background and amplification modes in the inlet, n is the refractive index of the medium, $c_0$ is the speed of light in a vacuum, $\sigma_{NO_2}$ is the absorption cross section of $NO_2$, and eCL is characteristic for each particular set up.

PeRCA and absorption spectroscopy using high finesse optical cavities have been recently used for ground based measurements of $RO_2^*$ radicals (Liu et al., 2009; Wood and Charest, 2014; Chen et al., 2016). PeRCEAS addresses the particular constrains related to airborne measurement by optimising the conversion of probed radicals in the reactor and the accuracy of the $NO_2$ measurement.

In this study, the specifications and airborne performance of PeRCEAS are described based on thorough and extensive laboratory characterisations and calibrations. The present study builds on the experience gained from the PeRCEAS deployment in three airborne measurement campaigns in the framework of the HALO missions: Oxidation Mechanism Observations, OMO, (see: www.mpic.de/ forschung/ kooperationen /halo/omo-mission.html) in 2015, and Effect of Megacities on the transport and transformation of pollutants on the Regional and Global scales, EMeRGe, (see: www.iup.uni-bremen.de/emerge/) in 2017 and 2018 onboard of the HALO platform.

## 2. PeRCEAS general description: mechanical and electrical setup

The PeRCEAS airborne instrument, shown schematically in Figure 1, comprises essentially the DUALER inlet (DUal channel Airborne peroxy radical Chemical AmplifiER) installed inside a pylon located on the outside of the HALO fuselage, and two CRDS $NO_2$ detectors mounted in a rack inside the HALO cabin. The first laboratory prototype reported by Hortsjann et al. (2014) has been significantly improved using the experience gained from the deployments in HALO missions. The following description of the instrument focuses on the modified and optimised features of PeRCEAS.

Briefly, sampled air enters PeRCEAS through the DUALER pre-chamber, which is at a lower pressure than that outside of the HALO, through an orifice in a truncated cone, i.e. a nozzle. From this pre-chamber the air is pumped simultaneously through the two flow reactors and a bypass line. At the upper addition point a mixture of CO or $N_2$ and NO enters each reactor. At the lower addition point, a flow of $N_2$ or CO enters each reactor. This enables the CO and $N_2$ flows in the two reactors within the DUALER to be switched simultaneously but out of phase with one another from the upper to the lower addition point. At the addition points, the reagent gases enter the reactor through eight circular distributed 1 mm holes to facilitate the rapid mixing with the sampled air. During measurements, the pressure in the pre-chamber and both reactors is held constant. However, there is a small pressure fluctuation during the switching of flows between the upper and lower mixing point. The flow passing through each reactor enters a CRDS $NO_2$ detector. Afterwards, the sample flows together with the air from the bypass line are scrubbed for CO and $NO_x$ and exhausted by the pump.

The DUALER inlet comprises two PeRCA chemical reactors having alternating measurement modes, which are out of phase with one another. During the first part of the measurement cycle, the first reactor and detector are in amplification mode, while simultaneously the second reactor and detector are in background mode. In the second part of the cycle, the CO addition point in both reactors is switched. Consequently, the first reactor and detector are then in background mode while the second reactor and detector are in amplification mode. In the analysis of the measurements, the amplification and background signals from both detectors are combined appropriately. This improves accuracy and temporal resolution of the resultant $RO_2^*$ data set (see 3.1).

In the DUALER, a stable pressure in the pre-chamber is achieved by a pressure regulator, which controls the flow through the bypass line. As noted the flow rate through the reactors is held constant during measurements. Consequently, when the outside air pressure changes, the bypass flow rate from the pre-chamber is changed. The outer dimensions, shape, form and weight of the DUALER are constrained by the inlet pylon in use with the research aircraft HALO. After the first version of the DUALER (from now on called DUALER I) was flown, the inner dimensions of the pre-chamber were further optimised to reduce the wall losses and turbulence in the pre-chamber. For this, in the DUALER II the volume of the pre-chamber was increased by extending its vertical extent, the length of the truncated cone on top of the reactors was reduced in 3 mm, and the volume of the reactors was increased to 130.5 ml from the 112 ml in DUALER I. These changes resulted in a higher eCL and improved pressure stability in DUALER II as compared to DUALER I. Figure 2 shows the upper part of both DUALER I and DUALER II.

The improvements of the PeRCEAS CRDS detectors for $NO_2$ targeted the signal stability and the in-flight adjustment of the optical alignment. The optical cavity remains similar to that described in Horstjann et al., (2014), i.e., a V-resonator of ca. 100 $cm^3$ volume formed between glued highly reflective mirrors (reflectivity, R = 99.995 %, diameter, d = 0.5", radius of curvature, roc = 100 cm, AT Films, USA) on the side of a Teflon coated aluminium cuboid. As shown in figure 3, the current $NO_2$ detector houses a 100 mW continuous wave multimode diode laser (Stradus 405, wavelength ca. 408 nm, max 100 mW output power, Vortran Laser Technology Inc.). With this, the fine adjustment of the laser is simplified and improved, and the piezo electric stack used to achieve mode matching between the single mode laser and the optical cavity in Hortsjann et al., (2014) becomes unnecessary and is removed. The laser is aligned to the V-resonator using two motorised alignment mirrors (0.5'' aluminum mirrors mounted on Newport 8885 Picomotor Actuated Pint-Sized Center Mounts). These enable the correction of any mechanical displacement of the optical elements with respect to the V-resonator arising from misalignment due to vibration or mechanical shocks during transport, installation or in-flight measurement. During alignment procedures and for test purposes, a beam camera (BM-USB-SP907-OSI, Ophir Spiricon Europe GmbH) monitors the beam profile and simplifies the identification of misalignments or loss of performance of the optical system.

Concerning the data acquisition and processing, the system is equipped with the current National Instruments PXI-8840 computer with two PXI-6132 DAQ cards working with 1 M sample $s^{-1}$ to measure the ring down signal from both detectors. Other sensor data such as pressure, flow, temperature and humidity are measured with one PXI-6129 DAQ card at a rate of 1 sample $s^{-1}$.

Three identical interchangeable detectors (hereafter named Abbé: AB; Fraunhofer: FH; and Fresnel: FR) have been constructed and characterised at the IUP-UB, of which two are always simultaneously deployed in measurement campaigns.

Additional components used to operate the PeRCEAS such as mass flow and pressure sensors/ controllers, gas cylinders and electronics are mounted in the main rack, as described in Hortsjann et al, (2014). The instrument rack in the aircraft cabin is connected to the DUALER through an aperture plate. Other ancillary parts of the PeRCEAS, such as the vacuum pump, a secondary containment for dangerous gases (CO), a scrubber unit for $NO_x$/CO and the rest of gas cylinders are also installed in the aircraft cabin.

## 3.    PeRCEAS mode of operation

The mode of operation of PeRCEAS is optimised by systematically investigating the short and long term stability of the detector signal and the effect of potential interferences. Factors affecting the overall performance of PeRCEAS for airborne measurements are discussed in the following sections.

### 3.1. Measurement modes: integration time

The mode time is defined as the time selected for the measurement in either amplification or background mode. The modulation time is the time taken for a complete measurement cycle, which comprises the sum of one amplification and one background mode. The PeRCEAS measurement cycle is illustrated in Figure 4. The $\Delta NO_2$ for each detector is calculated from the ring down time of two consecutive modes using Eq.(1). If the mode time is adequately selected, the $RO_2^*$ retrieved per measurement cycle is identical in both measurement lines, as the two reactors are operated out of phase with one another. The final $RO_2^*$ data is calculated as the mean of the $RO_2^*$ determined from the $\Delta NO_2$ and eCL of both detectors for a given measurement cycle. The time resolution of the $RO_2^*$ measurement is then equal to the mode time. After switching modes, a small pressure pulse leads to an oscillation of the $NO_2$ signal. Consequently, the first 20 s of each mode are not used in data analysis. The time lag arising from the time taken for the sample flow between the CRDS detector and the point of switching is typically less than 8 s (see Table 3).

Typically, 650 to 800 ring down times of the $NO_2$ absorption are averaged per second and the measurement of $NO_2$ is made at 1 Hz. Individual ring down times are occasionally saved for sensitivity studies. Modulation and mode times are selected empirically. The optimised values are a compromise between the time taken for the detector signal to stabilise after the $CO/N_2$ flow is switched between the addition points, and the temporal variability of the chemical composition of the air probed.

To optimize the mode time and the modulation cycle, the Allan variance (Allan, 1966; Werle et. al., 1993) was analysed for PeRCEAS. Given a time series of N elements and a total measurement time $t_{acq}$, $t_{acq} = f_{acq} \cdot N$, where $f_{acq}$ is the frequency of acquisition, then the Allan variance is defined as:

$$\sigma_x^2(\tau) = \frac{1}{2} \langle (x_{i+1} - x_i)^2 \rangle_\tau \qquad (Eq.2)$$

where $x_i$ is the mean over a time interval of a length $\tau$, being $\tau = f_{acq} \cdot m$; and m the number of elements in a selected interval. The use of $\langle \ldots \rangle$ denotes the arithmetic mean. The square root of the Allan variance is the Allan deviation. For random noise, the Allan deviation at any given integration time determines the detection limit of the measurement.

The Allan variance plot for measurements of 5.6 ppbv $NO_2$ at 200 mbar and 23 °C is shown in figure 5. As can be seen, the optimal averaging time for the three PeRCEAS detectors is in the range between 20 s and 50 s. The corresponding minimum ($2\sigma$) detectable mixing ratio is < 60 pptv ($3.15 \times 10^8$ molecules $cm^{-3}$ for these P and T conditions). Slow temperature drifts over longer averaging times impact on both the laser and the resonator characteristics. This behaviour is observed for averaging times longer than 60 s.

In addition to random noise, systematic noise in the measurement arises from instability of the laser and or detector response over the modulation time. This is decisive for the overall accuracy of the $RO_2^*$ determination. As mentioned in the introduction, the ambient $RO_2^*$ concentrations are calculated from the CRDS detector signals using Eq. (1). This assumes that the variation of $\tau_0$ has a negligible impact over two consecutive modulation periods.

Temperature changes of the detector affect: i) the diode laser emission, both its amplitude and wavelength, and ii) the mode matching between laser and detector, and consequently the $\tau_0$. The effect of the variations in $\tau$, resulting from changes in room or HALO cabin air temperatures, on the accuracy and precision of the $\Delta NO_2$ determination was investigated by a series of laboratory experiments. For this, modulated concentrations of $NO_2$ in the flow were generated. This was achieved by alternating between two selected $NO_2$ concentrations once per minute. The temperature of the CRDS detector, T, and $\tau$ were then measured. Detector temperature gradients over a time t, i.e., $\Delta T/\Delta t$, determined by the temperature within the detector housing close to the photodiode, were induced by controlled changes in the room temperature.

Figure 6 shows the effect of introducing temperature perturbations in a modulated $NO_2$ signal between 11.5 and 12.1 ppbv measured at 200 mbar and 23 °C. As can be seen in the figure, a temperature perturbation affects both precision and accuracy of the retrieved $\Delta NO_2$. For temperature gradients up to $\Delta T/\Delta t \approx 7$ °C h$^{-1}$ the experimental precision of the $\Delta NO_2$ determination remains within ($2\sigma$) 150 pptv (= 7.3 x 10$^8$ molecules cm$^{-3}$ at 200 mbar and 23°C).

Using the results from the above sensitivity and calibration studies, a 60 s mode time and a 120 s modulation time is selected as optimal signal to noise ratio of the $\Delta NO_2$ at a significance level for $2\sigma$ error < 3.15 x 10$^8$ molecules cm$^{-3}$.

### 3.2. Sample flows and residence times

Sample and reagent gas flows have different and related impacts on the sensitivity of the PeRCEAS measurements. The rate of the sample flows determines the residence time in different parts of the flow system, which in turn determine the reaction time for the conversion of $RO_2^*$ to $NO_2$, the titration of the $O_3$ in the sampled ambient air, and the thermal decomposition of peroxynitrates, and peroxynitric acid, $HO_2NO_2$, which can produce an $NO_2$ interfering signal. Interferences are minimised by a short residence time, facilitated by a rapid flow. Conversely, the $RO_2^*$ to $NO_2$ conversion rate in the DUALER is determined by the concentration of the CO and NO reagent gases added. The eCL increases with the increase in CO added to the sample (Reichert et al., 2003). Laboratory tests comparing the performance of PeRCEAS using alternative gases showed that CO is the most suitable gas to convert OH back to $HO_2$ in the chain reaction used in the chemical amplification. However CO is a toxic and flammable gas with a lower explosive limit (LEL) in air of 12.5 % v/v at room temperature and atmospheric pressure. This LEL is the minimum concentration necessary to support the gas combustion along with an ignition source such as a spark or flame (Zabetakis, 1965). Consequently, safety considerations limit the maximum flow of CO.

NO participates in both the chain carrier and chain termination reactions as explained in detail elsewhere (e.g. Hastie et al 1991, Mihele et al., 1999). For a constant CO concentration, these reactions of NO determine the eCL at different pressures. Increasing NO in the reactor changes the sensitivity of the amplification to different peroxy radicals due to termolecular reaction of RO with NO forming $RONO_2$. The also termolecular reaction of $RO_2$ with NO leading to $RONO_2$ increases with increasing size for alkylperoxy radicals but remains < 20 % (Lightfoot et al, 1992, Tyndall et al., 2001).

The rate of titration of the sampled $O_3$ by NO to form $NO_2$ also depends on the concentration of NO added to the sample flow and the time for reaction before reaching the detector (Kartal et al., 2010).

As a result of the above, the flows of the sampled ambient air, NO and CO and the pressure in the DUALER are selected for each deployment of PeRCEAS. The selections are a compromise between safety requirements, which limit the amount and concentration of gases on board HALO, and the values of eCL achieved for a particular residence time.

### 3.2.1. Effective chain length

The eCL of the DUALER reactors is determined in the laboratory by using a calibrated source of peroxy radicals. The latter uses the photolysis of water vapour at 184.9 nm (see Schultz et al., 1995). Briefly, a known water vapour - air mixture is photolysed by a low pressure mercury (Hg) lamp. A nitrous oxide ($N_2O$) absorption filter attenuates the intensity of 184.9 nm radiation. This is achieved by varying the $N_2O/N_2$ ratio in the filter absorption zone. The photolysis of $H_2O$ makes an OH and H. In air, the H reacts with $O_2$ in a termolecular reaction to make $HO_2$. The photolysis of oxygen molecules yield oxygen atoms, O which react with $O_2$ in a termolecular reaction to make $O_3$ (see Reichert et al., 2003). CO is added to the gas mixture in the source to convert the OH into $HO_2$ radicals. As a result, each absorbed photon by a water vapour molecule generates two $HO_2$ molecules. Alternatively, the addition of a hydrocarbon, RH, leads to the conversion of OH to a $RO_2$, and consequently to a 1:1 mixture of $HO_2$ and $RO_2$, for calibration. The concentration of $HO_2$ or $RO_2$, and $O_3$ is thus proportional to the intensity of 184.9 nm electromagnetic radiation. As the absorption coefficient of $N_2O$ (Cantrell et al., 1997) does not change significantly around 185 nm ($\sigma_{N2O}$=14.05×10$^{-20}$ cm$^2$ molecule$^{-1}$ at 25 °C with a 0.02×10$^{-20}$ cm$^2$ molecule$^{-1}$ K$^{-1}$, temperature dependency), different $HO_2$ and $RO_2$ radical amounts can be produced for a constant $H_2O$ concentration. A flow reactor providing a known amount of $HO_2$ or $RO_2{}^*$ in a laminar flow is placed inside a pressure chamber, having a vacuum sealed connection to the DUALER inlet. This setup is described in detail elsewhere (Kartal et al., 2009; Horstjann et al., 2014). For the $HO_2$ calibration configuration, the $HO_2$ concentrations are calculated using:

$$[HO_2] = \frac{\sigma_{H_2O}^{184.9nm}}{\sigma_{O_2}^{184.9nm}} \times \frac{[H_2O]}{[O_2]} \times [O_3] \qquad \text{(Eq.3)}$$

The value for the absorption cross section of $H_2O$ at 184.9 nm, $\sigma_{H_2O}^{184.9nm} = (7.14 \pm 0.2)$ x 10$^{-20}$ cm$^2$ molec$^{-1}$ is taken from Cantrell et al., (1997) and Hofzumahaus et al., (1997), while the $O_2$ effective cross section $\sigma_{O_2 eff}^{184.9nm}$ is determined experimentally for the particular Hg lamp used for calibration and the measurement conditions (Creasey et al., 2000; Hofzumahaus et al.,1997; Kartal et al., 2010).

$HO_2$ and 1:1 $HO_2$: $CH_3O_2$ mixtures are generated at controlled pressures within expected airborne concentration ranges by adding 0.35 % of CO or $CH_4$ respectively to the humidified air in the calibration flow tube. Radical mixing ratios are changed every ten minutes and stepwise from 8 pptv to 150 pptv. The PeRCEAS eCL is determined as the slope of the measured $NO_2$ versus the calculated radical mixing ratios in the calibration flow tube. The $O_3$ generated by the radical source is converted in the DUALER to $NO_2$ by its reaction with NO, which is in excess. Therefore the $O_3$ entering the reactor during the radical calibration is detected as $NO_2$ in the background and amplified signals.

Figure 7 depicts the PeRCEAS eCL versus the NO concentration obtained experimentally for inlet pressures between 200 and 350 mbar. As expected, the eCL decreases with increasing NO concentration. This is attributed to the increase in the rate of the termination reactions forming HONO and $CH_3ONO$. The latter also causes the eCL to be lower for the 1:1 $HO_2$ : $CH_3O_2$ radical mixture. The experimentally determined eCL is higher for DUALER II, as expected from the reduction of radical losses in the pre-chamber respect to DUALER I. For a constant NO number concentration, eCL values increase with increasing pressure. The overall observed behaviour of eCL versus [NO] in these experiments is in good agreement with the results reported by Kartal et al., (2010).

A simple chemical box model was developed using the Kintecus software (Ianni, J. C, 2013; 2017; www.kintecus.com) to simulate the peroxyradical amplification in the DUALER inlets. The model comprises two consecutive modules to simulate the pre-chamber and the reactors separately. The first module takes into account radical terminating reactions prior to the addition of reagent gases. The second module includes the relevant amplification and terminating reactions taking place in the reactor, as listed in Table 1. The rate coefficients used are taken from Burkholder et al, (2015). The first module is initialised with 50 pptv $HO_2$ (6.07 x $10^8$ molecule $cm^{-3}$ at 500 mbar) or a 50 pptv $HO_2$ + 50 pptv $CH_3O_2$ mixture. The second module is initialised with the radical output of the first module and calculates the eCL at different pressures for the conditions used in the calibration set up at 500 mbar (9 % CO, and 3 ppb $O_3$) and a series of NO concentrations. According to sensitivity studies, the amount of $O_3$ used for initialising the model does not to significantly affect the eCL value calculated.

As in previous work (Kartal, 2009; Chrobry, 2013), the radical wall loss rates, $k_w$, in the DUALER reactors are estimated by using the expression from Murphy et al., (1987) and Hayman, (1997) for a cylindrical reactor:

$$k_w = 1.85 \left( \frac{v^{1/3} D^{2/3}}{d^{1/3} L^{1/3}} \right) \left( \frac{S}{V} \right) \qquad (Eq.4)$$

where S is the surface area in $cm^2$, V the volume in $cm^3$, L the length and d the diameter of the flow tube in cm, v the velocity of the gas in cm $s^{-1}$, and D is the diffusion coefficient, calculated to be $D_{HO_2}$=0.21 and $D_{CH_3O_2}$=0.14 in $cm^2 s^{-1}$.

Using (Eq 4) values for $k_{w_{HO_2}}$ and $k_{w_{CH_3O_2}}$ are estimated to be respectively 0.97 $s^{-1}$ and 0.74 $s^{-1}$ for the DUALER reactors at a pressure of 300 mbar. The $k_w$ for the pre-chamber cannot be calculated by Eq.4 due to its complex geometry and flow dynamics. Consequently, different values of $k_w$ are used in module 1 to account for radical losses in the pre-chamber matching the eCL obtained experimentally.

Figure 8 shows the eCL obtained experimentally for the DUALER II at 300mbar inlet pressure, 500 ml/min sample flow, and different NO mixing ratios added to the inlet. The best agreement between modelled and experimental data are obtained for the calculated $k_w$ in the reactor, and 64 % $HO_2$ and 54 % $CH_3O_2$ radical losses in the pre-chamber. This is in agreement with previous results reported by Kartal et al (2010) for a similar configuration.

Table 2 summarises the simulated PeRCEAS sensitivity for the $HO_2$ and $CH_3O_2$ detection for different NO mixing ratios in the reactor at 300 mbar. Up to 10 ppm NO ([NO] 7,29 x $10^{13}$ molecules $cm^{-3}$) the difference in sensitivity remains within the PeRCEAS uncertainty. The ratio of the $eCL_{CH_3O_2}$ /$eCL_{HO_2}$ is defined as α. The estimated values of α from modelling and measurements are given in table 2. For the assessment of air masses the measurements of $HO_2$ + α·$RO_2$, where α$RO_2$ ≈ α·$CH_3O_2$, are compared with atmospheric model values of $HO_2$ + α·$RO_2$.

The present results confirm that the determination of the eCL in the laboratory for each particular setup and measurement condition is essential.

### 3.2.2. Conversion of ambient $O_3$ into $NO_2$

As explained in section 3.1. the simultaneous use of two detectors measuring out of phase results in the temporal resolution of the $RO_2^*$ data being 60s. In this way, the horizontal resolution of the PeRCEAS airborne measurements, which depends on the speed and altitude of HALO, is typically between 7 and 15 km. Longer modulation cycles than 120 s result in noisy and

unrepresentative averages for ambient measurements in air masses having significant short term variability of $O_3$ and $NO_2$. To keep the temporal resolution of the $RO_2^*$ data to be equal to the mode time, the rapid and complete conversion of ambient $O_3$ into $NO_2$ within the PeRCEAS is required. For this, the NO concentration added at the inlet has to be sufficient for a complete titration of the sampled $O_3$ before reaching the detector. Figure 9 depicts the $O_3$ decay simulated for 100 and 200 ppb sampled $O_3$, i.e., $5 \times 10^{11}$ - $1.7 \times 10^{12}$ molecules $cm^{-3}$ at 200 and 300 mbar respectively, assuming the titration to be completed for a rest of $[O_3] = 5 \times 10^7$ molecules $cm^{-3}$. The $O_3$ absorption at 408 nm is assumed to be a negligible source of systematic error. These results are in agreement with a series of laboratory measurements made at 300 mbar for the DUALER II with a total flow of 500 ml $min^{-1}$ as shown in Figure 10. After 8 s in PeRCEAS the $O_3$ is titrated for mixing ratios above 10 ppm at the conditions investigated (i.e., $4.83 \times 10^{13}$ - $7.29 \times 10^{13}$ molecules $cm^{-3}$ at 200 and 300 mbar respectively). The sample residence times for both DUALER inlets are summarised in Table 3.

### 3.2.3. Peroxyacyl nitrates thermal decomposition

Peroxyacyl nitrates ($RC(O)OONO_2$) such as peroxyacetylnitrate, PAN and peroxypropionyl nitrate can decompose thermally inside PeRCEAS. The extent of the decomposition to peroxy radicals and $NO_2$ depends on the time and the temperature. If the thermal decomposition occurs at shorter time scales than the modulation time, they can be a significant interfering source of radicals which are chemically amplified and lead to additional $NO_2$. In a rapidly changing background the $RO_2^*$ determination might be affected according to the temperatures and sample residence times between the gas addition points in the DUALER (Table 3).

To evaluate this effect the production of peroxy radicals from the thermal decomposition of 1 ppb PAN at different temperatures and pressures has been simulated. The results obtained with a box model (Ianini, 2003) including the reactions:

$$CH_3COO_2NO_2 \; \rightarrow \; CH_3COO_2 + NO_2 \qquad (R1)$$

$$CH_3COO_2 + NO \rightarrow CH_3 + CO_2 + NO_2 \qquad (R2)$$

$$CH_3 + O_2 + M \; \rightarrow CH_3O_2 \qquad (R3)$$

are depicted in figure 11. The rate coefficients used are taken from Burkholder et al., (2015).

The $[CH_3O_2]$ produced does not vary significantly at the pressures investigated. As the temperature of the PeRCEAS reactors during flight generally remain under 290 K, this source of radicals is considered to be negligible for most operating conditions. The thermal stability of the PAN analogues is similar to that of PAN but they are usually at much lower concentrations than PAN in the atmosphere and also assumed to be a negligible source of error.

### 3.3. Operating pressure: radical losses and absolute humidity in DUALER

As explained in section 2, the PeRCEAS operating pressure is held constant and below ambient pressure to have a constant radical chemical conversion in the DUALER reactors during the flight. However, the $\Delta P = P_{ambient} - P_{inlet}$ is different at different flight altitudes and leads to changes in the physical losses and the humidity in the pre-chamber. These potentially may have a significant effect in the eCL, as reported by Kartal et al., (2010 and references herein).

To evaluate this effect for PeRCEAS, different ΔP were experimentally generated by changing the pressure in the pressure chamber while keeping inlet conditions like pressure, mixing ratios of the reagent gases (NO, CO and $N_2$), sampling gas velocity (flow) and relative humidity invariable.

Figure 12 shows the variation of the eCL for 10 and 45 ppm NO within a pressure range of 50 mbar ≤ ΔP ≤ 600 mbar. As can be seen in the figure, the eCL remains within 10 % of the mean value except for the values for ΔP < 100 mbar. This might be the result of variations in the relative importance of terminating processes (e.g. wall losses versus chemical reactions) with the sample velocity through the pre-chamber (Kartal et al., 2010) as indicated by the differences in the eCL pattern for NO 10 ppm and 45 ppm below 100 mbar. Consequently ΔP = 100 mbar is defined as the minimum operating pressure for PeRCEAS airborne measurements. With this limitation measurements of $RO_2^*$ at flight altitudes up to 12 km can be successfully made.

The effect of changes in the sampled air humidity on the eCL has been reported and studied by Mihele and Hastie, (1998) and Mihele et al., (1999). Reichert et al. (2003) investigated the dependency of the eCL for ground based measurements at 20 °C and 30 °C and standard pressure, i.e., keeping the relative humidity but almost doubling the absolute water concentration. The obtained eCL values did not differ within the experimental error and confirmed the dependency of eCL on the relative humidity. All these measurements were performed at a pressure of one atmosphere and for 3.3 ppm NO ([NO] 8.12 x $10^{13}$ molecules $cm^{-3}$).

In this work radical mixtures were sampled at 25 °C for relative humidity between 2 % and 25 %. This leads to a ca. 20 times increase in the absolute [$H_2O$]. These conditions cover the [$H_2O$] expected for a larger T range (-20 – 30 °C) during airborne measurements in the free troposphere at 200 and 300 mbar inlet pressures. Figure 13 shows the [$H_2O$] in the air probed versus the [$H_2O$] in the inlet for real measurements on board of the HALO aircraft. The results in figure 14 for 45 ppm ([NO] 3.28 x $10^{14}$ molecules $cm^{-3}$ at 300 mbar) indicate that variations in the sample humidity do not lead to additional uncertainty in the $RO_2^*$ retrieval as the PeRCEAS eCL remains invariable within the experimental error up to [$H_2O$] ~ 1.4 x $10^{17}$ molec $cm^{-3}$. In contrast, for 10 ppm and 30 ppm NO in the reactor ([NO] 7.29 x $10^{13}$ and 2.19 x $10^{14}$ molecules $cm^{-3}$ at 300 mbar) the eCL shows a clear dependency with the ambient [$HO_2$]. The comparison with the eCL values obtained by Reichert et al (2003) at 1 atmosphere indicate a eCL dependency on [$H_2O$], temperature and pressure having a different pattern for 45 ppm NO in the reactor. This is explained by invoking the competition in the amplification chain length, CL, between $HO_2$ and OH removal rates, as explained in Hastie et al., (1991) and Reichert et al., (2003). At [NO] ~ 3.28 x $10^{14}$ molecules $cm^{-3}$ the CL begins to be dominated by the rate of the termination termolecular reaction of OH with NO, which is independent of water vapour. This eCL dependency has to be taken considered in the analysis of ambient air $RO_2^*$ measurements.

## 4.    PeRCEAS $RO_2^*$ determination: Error calculation, detection limit and accuracy

The determination of $RO_2^*$ concentrations from PeRCEAS measurements is subject to two types of errors which either a) are intrinsic to  the CRDS and PERCA techniques and can be characterised under controlled conditions in the laboratory, or b) result from the in-flight variability in the temperature, velocity and pressure conditions and cannot be readily reproduced in the laboratory.

### 4.1.    Errors related to the CRDS technique

Provided that the $NO_2$ absorption is the dominant process leading to the extinction of light at ~ 408 nm in the optical cavity of each detector, the absorption coefficient can be calculated from Eq.1 by considering $\tau_1$ and $\tau_2$ as the cavity ring down times with and without sample respectively. However, the effective $\sigma_{NO_2}$, $\tau$ and $\tau_0$ can differ from one detector to another.

The effective $\sigma_{NO_2}$ for each PeRCEAS $NO_2$ detector has been determined by using the convolution of the $NO_2$ absorption cross section from Vandaele et al. (2002) with the normalised laser spectra from the corresponding detector. The values obtained have been verified by regular sampling of $NO_2$ mixtures of known concentration in synthetic air.

The PeRCEAS lasers are operated at the maximum 100 mW power to achieve the best Gaussian profile for the emission and are digitally modulated during operation. The laser emission spectrum is measured periodically in the laboratory by using a calibrated spectrometer (AvaSpec-ULS2048x64; 295-535 nm grating; 0.132 nm resolution) to verify the long term spectral stability. A sample comparison of spectra obtained for the three PeRCEAS detectors is included in the supplementary information (Figure SI-1).

By integrating $\sigma_{NO_2}$ under the normalized laser spectrum, the effective $\sigma_{NO_2}$ are calculated to be $6.0 \pm 0.3$ x $10^{-19}$, $6.3 \pm 0.3$ x $10^{-19}$ and $6.4 \pm 0.3$ x $10^{-19}$ $cm^2$ $molecule^{-1}$ for the AB, FH and FR detectors respectively. The errors are calculated from the $2\sigma$ variation in the 1 hour average of 10 samples $s^{-1}$ laser emission spectrum regularly measured and the error reported for the high resolution $NO_2$ spectra.

According to Eq. (5), the effective $NO_2$ absorption cross-section is $1/c_0$ times the slope of the inverse of the measured $\tau$ versus the $NO_2$ number concentration:

$$\frac{1}{\tau_x} = c_0 \sigma_{NO_2} [NO_2]_x + \frac{1}{\tau_0} \qquad \text{(Eq.5)}$$

The result of applying Eq. (5) to the PeRCEAS detectors is depicted in Figure 15. The detectors sampled known mixtures of $NO_2$ from commercial gas cylinders in synthetic air at 200 mbar as shown in the supplementary information (Figure SI-2). The effective $\sigma_{NO_2}$ obtained agrees within 5 % with the values derived by integrating $\sigma_{NO_2}$ under the normalized laser spectrum as described above.

The y intercept in figure 15 corresponds to $1/\tau_0$ which is different for each detector. This is attributed to slight differences in the mirror reflectivity and in the overall alignment of the optical cavities. The value of $\tau_0$ for a particular detector is not expected to vary significantly under laboratory conditions.

## 4.2. Errors related to the PeRCA technique

The determination of $RO_2^*$ mixing ratios from the $\Delta NO_2$ measurement requires accurate knowledge of the eCL which depends upon physical parameters, such as temperature, pressure, wall losses, residence time in the reactor and the operating conditions as discussed in section 3. Generally, in-flight variations in the HALO cabin temperature affect minimally the accuracy of the $RO_2^*$ determination.

The main sources of uncertainty in the eCL determination are the radical generation, and the $NO_2$ determination from CRDS due to the accuracy of the $\sigma_{NO_2}$, which is estimated to be 5 % ($2\sigma$) (see 4.1). In the current set up, the generation of peroxy radicals

(Eq.3) has a precision $< 3$ pptv ($2\sigma$). Based on the experimental reproducibility of radical calibrations the eCL precision is within 3 % under all conditions investigated. In addition to this, the experimental determination of eCL has a 15 % uncertainty, dominated by the 10 % uncertainty of both $[O_3]$ and $\sigma_{O_2 eff}^{184.9nm}$ determinations using the calibration setup (Creasy et al., 2000; Kartal et al., 2009). Other errors associated with the determination of $[H_2O]$ (0.05 %), $[O_2]$ (0.5 %) and the $\sigma_{H_2O}^{184.9nm}$ literature value (1.4 %) are significantly lower.

Figure 16 shows the calculated eCL from 6 radical calibrations carried out over six months for 300 mbar pressure, $\Delta P = 200$ mbar, and 1% relative humidity adding reagent gases to achieve 9 % CO and 30 ppmv NO within the DUALER II inlet. For $HO_2$ the obtained eCL values are $60 \pm 9$ and $61 \pm 9$ for reactor 1 and reactor 2, respectively. Those for radical mixtures are $46 \pm 7$ and $44 \pm 7$ respectively. The errors refer to the experimental precision ($2\sigma$) of the measurements.

### 4.3. Errors related to in-flight variability of air composition: DUALER approach and $RO_2^*$ retrieval

The in-flight dynamical stability of PeRCEAS is influenced by the stability of the sampling flows and pressures. This stability depends on pressure variations experienced by the instrument when the aircraft is turning, ascending or descending, as well as in the presence of turbulence. The noise in the $NO_2$ signal is generally larger in flight. This is attributed to the impact of mechanical vibration and temperature variation. Though the in-flight temperature in the HALO cabin remains reasonably constant, during the instrumental preparation on the ground prior to the flight the cabin temperature may increase up to 40 °C.

This affects the stability of the ring down time signal and the accuracy of the reference measurements.

In addition to the above, the retrieval of the $RO_2^*$ ambient mixing ratios requires a reliable discrimination of the interfering signals resulting from the variation of $NO_2$, $O_3$, PAN, and any other molecules in the sampled air leading to additional absorption or scattering at ~408 nm. As mentioned before, changes in the composition faster than two consecutive measurement modes might lead to erroneous peroxy radical retrievals. In the case of aircraft measurements, this effect might be important

due to the relative motion of the aircraft with respect to the air mass. The reliability of the PeRCEAS retrieval technique to effectively remove short term background variations was investigated in the laboratory by sampling $HO_2$ generated at a constant mixing ratio of $16 \pm 2$ pptv in synthetic air, while varying $O_3$ up to 30 ppbv. The DUALER I inlet was stabilised at 200 mbar and all other parameters like chamber pressure, mixing ratios of the reagent gases (30 ppmv NO, 9 % CO and 9 % $N_2$), sampling flows (500 ml/min) and relative humidity ($< 3$ %) were controlled and held constant.

As can be seen in figure 17 the $\Delta NO_2$ calculated from both detector signals remains around 700 pptv for a constant $O_3$ concentration, which is eCL times the $HO_2$ set value (i.e., $\approx 43$ x 16 pptv). $O_3$ variations within one minute lead to opposite deviations from the 700 pptv value in the $\Delta NO_2$ calculated from each system. This causes the $HO_2$ calculated from each system to deviate in the same manner from the actual value. Because the two reactors are operated out of phase with one another, the final $HO_2$ data is the mean of the $HO_2$ determined by each detector from their respective $\Delta NO_2$ using Eq.(1).

The $\Delta NO_2$ calculated over 1 minute has a standard deviation of the order of the variation of $O_3$ converted into $NO_2$ through the NO titration in the reactor, as shown in the retrieved $\Delta NO_2$ plot of figure 17. In the case of short term $O_3$ changes up to 30 ppbv, the 16 pptv $HO_2$ mixing ratio set (7.8 x $10^7$ molecules $cm^{-3}$ at 200 mbar and 25 °C) can be retrieved with a maximum deviation of 6 pptv (2.9 x $10^7$ molecules $cm^{-3}$ at 200 mbar and 25 °C). The error in the retrieved $HO_2$ data results from the 15 % uncertainty of the eCL and the background $NO_2$ variation within one minute caused by the $O_3$ variations. This result is valid for

all the background signal variations during a real-time measurement and proves the robustness of the DUALER approach for the retrieval of $RO_2^*$ in a rapidly changing environment.

### 4.4. $RO_2^*$ detection limit and accuracy

The PeRCEAS detection limit for $RO_2^*$ ($LOD_{RO_2^*}$) is calculated by dividing the $NO_2$ detection limit ($LOD_{NO_2}$) by the corresponding eCL for each measurement condition set in the laboratory. Provided that $LOD_{NO_2}$ is 60 pptv $NO_2$ ($3.15 \times 10^8$

molecules $cm^{-3}$ at 200 mbar and 23 °C), 2σ over one minute as mentioned in section 3.1, the $LOD_{RO_2^*}$ varies between 1 and 2 pptv for the eCL values expected under dominant conditions in the free troposphere. The $LOD_{RO_2^*}$ can additionally be determined from the eCL calibration curves at different measurement conditions, according to:

$$LOD_{RO_2^*} = 3 \cdot S_a/m \qquad \text{(Eq.6)}$$

$S_a$ is the standard deviation of the y-intercept and m is the slope of the $NO_2$ vs $HO_2$ calibration curve, (as in Fig 16, see 4.2). For

controlled laboratory conditions the $LOD_{RO_2^*}$ is $5.3 \times 10^6$ molecules $cm^{-3}$ ($\leq 2$ pptv in all conditions investigated for DUALER I and DUALER II). As stated in 4.2., the accuracy is mainly dominated by the uncertainty in the eCL determination for each condition and amounts ~15 %.

Conversely, as stated in previous sections, the in-flight PeRCEAS detector signals can be significantly affected by instabilities in physical parameters like pressure, temperature, flows, mechanical vibration and chemical composition which increase the

uncertainty of the $RO_2^*$ measurement. Therefore the in-flight error in the $RO_2^*$ measurement is calculated by taking into account the uncertainty of the eCL and the background variation in the signal within one modulation period as discussed in section 4.3.

The current sensitivity of PeRCEAS on HALO is competitive with similar airborne peroxy radical instruments. Table 4 summarises the specifications of state of art instruments for the airborne measurement of peroxy radicals. Ground based instruments are also included for comparison. Due to the differences in physical and chemical conditions used, a direct

comparison between methods is challenging and only possible for time resolution and detection limit for well-defined and controlled measurement conditions. As mentioned in the introduction, the Matrix Isolation and Electron Spin Resonance (MIESR) though being the only direct measurement technique of high precision, is not suitable for airborne measurements and is difficult to implement in field campaigns.

The pressure regulation in PeRCA based airborne instruments results in lower eCL than ground based ones. This is attributed to

radical losses in the pre-chamber prior to the addition of reagent gases for the radical chemical amplification. The modulation time limits the resolution, except in the case of continuous measurement of background and amplification signal by different detectors (e.g. Liu et al 2014). The increase in resolution is however associated with errors caused by differences in detector accuracy. In addition to this, during ambient measurement the detection limit and uncertainty of PeRCA based instruments are dependent on the variation of $O_3$ and $NO_2$ in the sampled air mass. The speciation between $HO_2$ and $\sum RO_2$ is challenging. LIF

based instruments have a better detection limit but are subject to interferences from $RO_2$ in the sample (Fuchs et al, 2011).

### 5. PeRCEAS for airborne measurements of $RO_2^*$

PeRCEAS has up to date been successfully deployed in 3 airborne measurement campaigns on board the HALO aircraft.

Figure 18 shows sample data of $RO_2^*$ measured on the 25.08.2015 over Egypt from 5 to 8.5 km during the first flight deployment of PeRCEAS in the OMO campaign. The $\Delta P$ ($\Delta P = P_{ambient} - P_{inlet}$) and [$H_2O$] in the inlet remained below the calculated yield values to affect the eCL stability.

As can be seen in the figure 18, the dynamic pressure variations experienced by the aircraft influence the stability of the inlet pressure. These changes are attributed to altitude changes, air turbulence, and changes in velocity, including turning, of the aircraft. The effect of inlet pressure instabilities on the retrieved $\Delta NO_2$ is not exactly identical for both detector signals. This leads to additional uncertainty in the $RO_2^*$ determination when using the procedure discussed in section 4.3.. For the data analysis, pressure spikes within 1 minute standard deviation higher than 2 mbar are identified and flagged. This approach enables data with large error due to dynamic pressure changes to be identified. Overall the error in the retrieved $RO_2^*$ is around 20 % in the measurement period shown in figure 18.

Two hours of measurements from the flight on the 19.03.2018 are shown in Figure 19 as an example of the third airborne deployment of PeRCEAS within the EMeRGe campaign in Asia. As can be seen in the figure, pressure fluctuations due to dynamic pressure changes have been reduced by up to 80 % in the improved PeRCEAS. Although the measured $\Delta NO_2$ is affected by altitude changes, the value of the retrieved $RO_2^*$ does not change significantly except for the maximum climbing rate directly after take-off. Furthermore, the beam camera and the motorised mirror mounts enable the identification and immediate correction of small misalignments. This improves significantly the instrumental performance while simplifying maintenance.

The results show the capability of PeRCEAS to capture $RO_2^*$ variations even in rapidly changing air masses from the boundary layer to the upper troposphere. The instrument performance is stable over the 7 hours of mission flight indicating the robustness of the instrument towards mechanical vibrations and temperature variations. Further analysis of $RO_2^*$ data obtained during measurement campaigns together with models and other trace gas measurements is ongoing.

## 6.   Summary and conclusion

The accurate measurement of peroxy radicals is essential to improve our understanding of the chemistry in the free troposphere. The PeRCEAS instrument has been designed, developed and thoroughly characterised for the measurement of the total sum of peroxy radicals onboard of airborne platforms. Parameters expected to affect the precision and accuracy of the measurement have been investigated in detail. Variations in the composition of the air mass within the modulation time are well captured when keeping the reactors out of phase and in alternating background/amplification modes with detectors of similar signal to noise ratio stability. Under controlled conditions in the laboratory the $RO_2^*$ detection limit remains around $5.3 \times 10^6$ molecules $cm^{-3}$ ($\leq$ 2 pptv) over a 60 s integration time for instrumental pressures from 160 to 350 mbar.

The performance of the PeRCEAS instrument has been proven to be suitable for airborne measurements during different campaigns onboard HALO. The in-flight precision and detection limit depends critically on the features of the flight like pressure, temperature, flows, mechanical vibration, water number concentration and short term variations in the chemical composition,  and must be calculated for each particular flight track. Therefore, the optimisation of the instrument had a particular focus on the robustness of the dynamical and detector signal stabilities, which makes PeRCEAS a reliable instrument for most flying conditions in the free troposphere.

**Author contribution:**

MG and MDAH designed the experiments, MG, VN, and YL carried out them. VN carried out the implementation in the HALO aircraft. MG prepared the manuscript with contributions from all co-authors. MDAH and JPB originated the measurement concept and participated in the research and the data analysis presented.

**Competing interests**:

The authors declare that they have no conflict of interest.

**Acknowledgements:**

The authors acknowledge funding for this study by the University of Bremen, the state of Bremen, and the HALO SPP 1294 grant from the DFG Deutsche Forschungsgemeinschaft. Special thanks are to the mechanical workshop of the University of Bremen and to Wilke Thomssen for the support in the construction and certification of the PeRCEAS instrument. The authors explicitly thank the Max Planck Institute in Mainz Air Chemistry Department (Director: Professor Dr. J. Lelieveld) for the provision of the secondary containment for HALO and the HALO SSC for recommending the funding its improvement and further optimisation. Also explicitly thank Enviscope GmbH for their flexible support in helping to deliver the flight worthiness certification of PeRCEAS, the secondary gas containment for use in HALO and the provision of support during the HALO OMO and EMeRGe campaigns. The authors also thank their colleagues in the OMO and EMeRGe communities for their collaboration, helpful discussions and support during the HALO flight campaigns.

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

| Amplification reactions | $k$ ($cm^3molec^{-1}s^{-1}$) | $k_0$ ($cm^3molec^{-3}s^{-1}$) | $n$ | $k_\infty$ ($cm^6molec^{-2}s^{-1}$) | $m$ |
|---|---|---|---|---|---|
| $HO_2 + NO \rightarrow NO_2 + OH$ | $8.0 \times 10^{-12}$ | | | | |
| $CO + OH + M \rightarrow HOCO + M$ | | $5.9 \times 10^{-33}$ | 1.0 | $1.1 \times 10^{-12}$ | - 1.3 |
| $HOCO + O_2 \rightarrow HO_2 + CO_2$ | $2.0 \times 10^{-12}$ | | | | |
| $CO + OH \rightarrow H + CO_2$ | $1.5 \times 10^{-13}$ | | | | |
| $H + O_2 + M \rightarrow HO_2 + M$ | | $4.4 \times 10^{-32}$ | 1.3 | $7.5 \times 10^{-11}$ | - 0.2 |
| $CH_3O_2 + NO \rightarrow CH_3O + NO_2$ | $7.7 \times 10^{-12}$ | | | | |
| $CH_3O + O_2 \rightarrow CH_2O + HO_2$ | $1.9 \times 10^{-15}$ | | | | |

| Termination reactions | $k$ ($s^{-1}$) | $k$ ($cm^3molec^{-1}s^{-1}$) | $k_0$ ($cm^3molec^{-3}s^{-1}$) | $n$ | $k_\infty$ ($cm^6molec^{-2}s^{-1}$) | $m$ |
|---|---|---|---|---|---|---|
| $OH + NO + M \rightarrow HONO + M$ | | | $7.0 \times 10^{-31}$ | 2.6 | $3.6 \times 10^{-11}$ | 0.1 |
| $OH + NO_2 + M \rightarrow HNO_3 + M$ | | | $1.8 \times 10^{-30}$ | 3.2 | $2.8 \times 10^{-11}$ | 0.0 |
| $OH + NO_2 + M \rightarrow HOONO + M$ | | | $1.0 \times 10^{-32}$ | 3.9 | $4.2 \times 10^{-11}$ | 0.5 |
| $CH_3O + NO + M \rightarrow CH_3ONO + M$ | | | $2.3 \times 10^{-29}$ | 2.8 | $3.8 \times 10^{-11}$ | 0.6 |
| $OH + HO_2 \rightarrow H_2O + O_2$ | | $1.1 \times 10^{-10}$ | | | | |
| $HO_2 + CH_3O_2 \rightarrow CH_3OOH + O_2$ | | $5.2 \times 10^{-12}$ | | | | |
| $OH + OH + M \rightarrow H_2O_2 + M$ | | | $6.9 \times 10^{-31}$ | 1.0 | $2.6 \times 10^{-11}$ | 0.0 |
| $OH + HONO \rightarrow H_2O + NO_2$ | | $4.5 \times 10^{-12}$ | | | | |
| $CH_3O_2 + CH_3O_2 \rightarrow CH_3O + CH_3O + O_2$ | | $3.5 \times 10^{-13}$ | | | | |
| $HO_2 + HO_2 \rightarrow H_2O_2 + O_2$ | | $1.4 \times 10^{-12}$ | | | | |
| $HO_2 + NO_2 + M \rightarrow HO_2NO_2 + M$ | | | $1.9 \times 10^{-31}$ | 3.4 | $4.0 \times 10^{-12}$ | 0.3 |
| $HO_2 (g) \rightarrow HO_2 (s)$ | 0.97 | | | | | |
| $CH_3O_2 (g) \rightarrow CH_3O_2 (s)$ | 0.74 | | | | | |

| Other reactions | $k$ ($cm^3molec^{-1}s^{-1}$) | $k_0$ ($cm^3molec^{-3}s^{-1}$) | $n$ | $k_\infty$ ($cm^6molec^{-2}s^{-1}$) | $m$ |
|---|---|---|---|---|---|
| $O_3 + NO \rightarrow O_2 + NO_2$ | $1.9 \times 10^{-14}$ | | | | |
| $CH_3COO_2NO_2 \rightarrow CH_3COO_2 + NO_2$ | $2.52 \times 10^{16} \exp(-1353/T)$ | | | | |
| $CH_3COO_2 + NO_2 + M \rightarrow CH_3COO_2NO_2$ | | $9.7 \times 10^{-29}$ | 5.6 | $9.3 \times 10^{-12}$ | 1.5 |
| $CH_3COO_2 + NO \rightarrow CH_3 + CO_2 + NO_2$ | $2.0 \times 10^{-11}$ | | | | |
| $CH_3 + O_2 + M \rightarrow CH_3O_2 + M$ | | $4.0 \times 10^{-31}$ | 3.6 | $1.2 \times 10^{-12}$ | - 1.1 |

Table 1: Reactions used in a box model for the eCL simulation of the PeRCEAS reactors.

| NO (ppm) | [NO] molecules cm$^{-3}$ | eCL$_{CH_3O_2}$ modelled | eCL$_{mix}$/eCL$_{HO_2}$ measured | eCL$_{mix}$/eCL$_{HO_2}$ modelled | $\alpha$ = eCL$_{CH_3O_2}$/eCL$_{HO_2}$ |
|---|---|---|---|---|---|
| 6 | 4.37E+13 | 93.5 | 0.89 | 0.97 | 1.04 |
| 10 | 7.29E+13 | 85.3 | 0.76 | 0.90 | 0.89 |
| 20 | 1.46E+14 | 46.8 | 0.73 | 0.79 | 0.65 |
| 30 | 2.19E+14 | 27.3 | 0.84 | 0.74 | 0.52 |
| 40 | 2.91E+14 | 17.7 | 0.77 | 0.70 | 0.43 |
| 45 | 3.28E+14 | 14.7 | 0.76 | 0.68 | 0.40 |

Table 2: PeRCEAS eCL simulated at 300 mbar for $HO_2$, $CH_3O_2$ and a 1:1 radical mixture (eCL$_{mix}$)

DUALER I

| Inlet pressure (mbar) | Reactor residence time (s) | | | Total residence time (s) | | |
|---|---|---|---|---|---|---|
| | 300 ml/min | 500 ml/min | 1000 ml/min | 300 ml/min | 500 ml/min | 1000 ml/min |
| 300 | 6.55 | 3.93 | 1.96 | 7.82 | 4.69 | 2.35 |
| 200 | 4.36 | 2.62 | 1.31 | 5.21 | 3.13 | 1.56 |
| 160 | 3.49 | 2.10 | 1.05 | 4.17 | 2.50 | 1.25 |
| 100 | 2.18 | 1.31 | 0.65 | 2.61 | 1.56 | 0.78 |
| 80 | 1.75 | 1.05 | 0.52 | 2.09 | 1.25 | 0.63 |
| 50 | 1.09 | 0.65 | 0.33 | 1.30 | 0.78 | 0.39 |

DUALER II

| Inlet pressure (mbar) | Reactor residence time (s) | | | Total residence time (s) | | |
|---|---|---|---|---|---|---|
| | 300 ml/min | 500 ml/min | 1000 ml/min | 300 ml/min | 500 ml/min | 1000 ml/min |
| 300 | 7.73 | 4.64 | 2.32 | 13.18 | 7.91 | 3.95 |
| 200 | 5.15 | 3.09 | 1.55 | 8.79 | 5.27 | 2.64 |
| 160 | 4.12 | 2.47 | 1.24 | 7.03 | 4.22 | 2.11 |
| 100 | 2.58 | 1.55 | 0.77 | 4.39 | 2.64 | 1.32 |
| 80 | 2.06 | 1.24 | 0.62 | 3.51 | 2.11 | 1.05 |
| 50 | 1.29 | 0.77 | 0.39 | 2.20 | 1.32 | 0.66 |

Table 3: Sample residence times in PeRCEAS for different operating total flows and pressures. Reactor residence time: residence time between the first and the second addition points in each reactor; total residence time: residence time between the first addition point in each reactor and the corresponding detector. Inner volumes up to the detector increased from 132 cm$^3$ in DUALER I to 220 cm$^3$ in DUALER II.


| Author | Year | Technique | eCL | $LOD_{NO_2}$ (pptv) | $LOD_{RO_2^*}$ (pptv) | Averaging time (s) | Pressure (mbar) |
|---|---|---|---|---|---|---|---|
| **Airborne instruments** | | | | | | | |
| Green et al. | 2002 | PeRCA + Luminol | 277 - 322 (3 ppmv NO + 7 % CO) | 180 | 1 | 20 | not controlled (from ground level to 7 km) |
| Kartal et al. | 2010 | PeRCA + Luminol | 45 ± 7 (3 ppmv NO + 7.4 % CO) | 130 ± 5 | 3 ± 2 | 60 | 200 |
| Horstjan et al. | 2014 | PeRCA + OF-CEAS | 110 ± 21 (6 ppmv NO + 9 % CO) | 300 | 3 - 5 | 120 | 300 |
| | | | 55 ± 10 (6 ppmv NO + 9 % CO) | 300 | 6 | 120 | 200 |
| Hornbrook et al. | 2011 | PeRCIMS | | | 2 | | 200 |
| Ren at al. | 2012 | LIF | | | 0.1 (2σ) | 60 | up to 300 |
| | | PerCIMS | | | 1 (2σ) | 15 | up to 300 |
| This work | | PeRCA + CRDS | 100 ± 15 (10 ppmv NO 9 % CO) 62 ± 9 (30 ppmv NO 9 % CO) 38 ± 4 (45 ppmv NO 9 % CO) | 60 | < 2 | 60 | 200 to 350 |
| **Ground based instruments** | | | | | | | |
| Cantrell et al. | 1984 | PeRCA + Luminol | 1010 (3 ppmv NO + 10 % CO) | | 0.6 | 300 | 1000 |
| Hastie et al. | 1991 | PeRCA + Luminol | 120 (2 ppmv NO + 4 % CO) | 50 | 2 | 10 | 1000 |
| Cantrell et al. | 1993 | PeRCA + Liminol | 300 (3 ppmv NO + 10 % CO) | | < 2 | 60 | 1000 |
| Reiner et.al. | 1997 | PeRCA + | 100 | | $10^6$ molec. cm$^{-3}$ | | 1000 |

| | | | | | | | |
|---|---|---|---|---|---|---|---|
| | | IMR-MS | | | | | |
| Sadanaga et al | 2004 | PeRCA + LIF | 190 (3 ppmv NO + 10 % CO) | 61 | 2.7 (50 % RH) 3.6 (80 % RH) | 60 | 1000 |
| Liu et al. | 2009 | PERCA + CRDS | 150 ± 50 (2σ) | 150 (3σ 10s) | 10 (3σ) | 60 | 1000 |
| Wood et al. | 2014 | PeRCA + CAPS | 168 ± 20 (3.75 ppmv NO 9.8 % CO) | 12 (1σ 30s) | 0.6 (40 % RH) | 60 | 1000 |
| Liu et al. | 2014 | PeRCA + CRDS | 190 | | 4 | 10 | 1000 |
| Chen et al. | 2016 | PeRCA + IBBCEAS | 91 ± 11 (7.7 ppmv NO 8.5 % CO) | 49 and 62 for differen chanels | 0.9 (10 % RH) | 60 | 1000 |
| Wood et al. | 2017 | ECHAMP + CAPS | 25 (dry) and 17(50 % RH) (1 ppmv NO 2.3 % $C_2H_6$) | 10 (1σ 45s) | 1.6 (50 % RH) | 90 | 1000 |
| Anderson et al. | 2019 | ECHAMP + CAPS | 23 (dry) and 12(58 % RH) (0.9 ppmv NO 1.3 % $C_2H_6$) | 10 (1σ 45s) | 1.6 (50 % RH) | 120 | 1000 |
| Edwards et al. | 2003 | PeRCIMS | | | 0.4 | 15 | 200 |
| Fush et al. | 2008 | LIF | | | 0.1 | 60 | 1000 |
| Mihelcic et al. | 2003 | MIESR | | | 2 | 1800 | 1000 |

Table 4: State of art instruments for the airborne measurement of peroxy radicals. Instruments for the ground based measurement are also included for comparison.

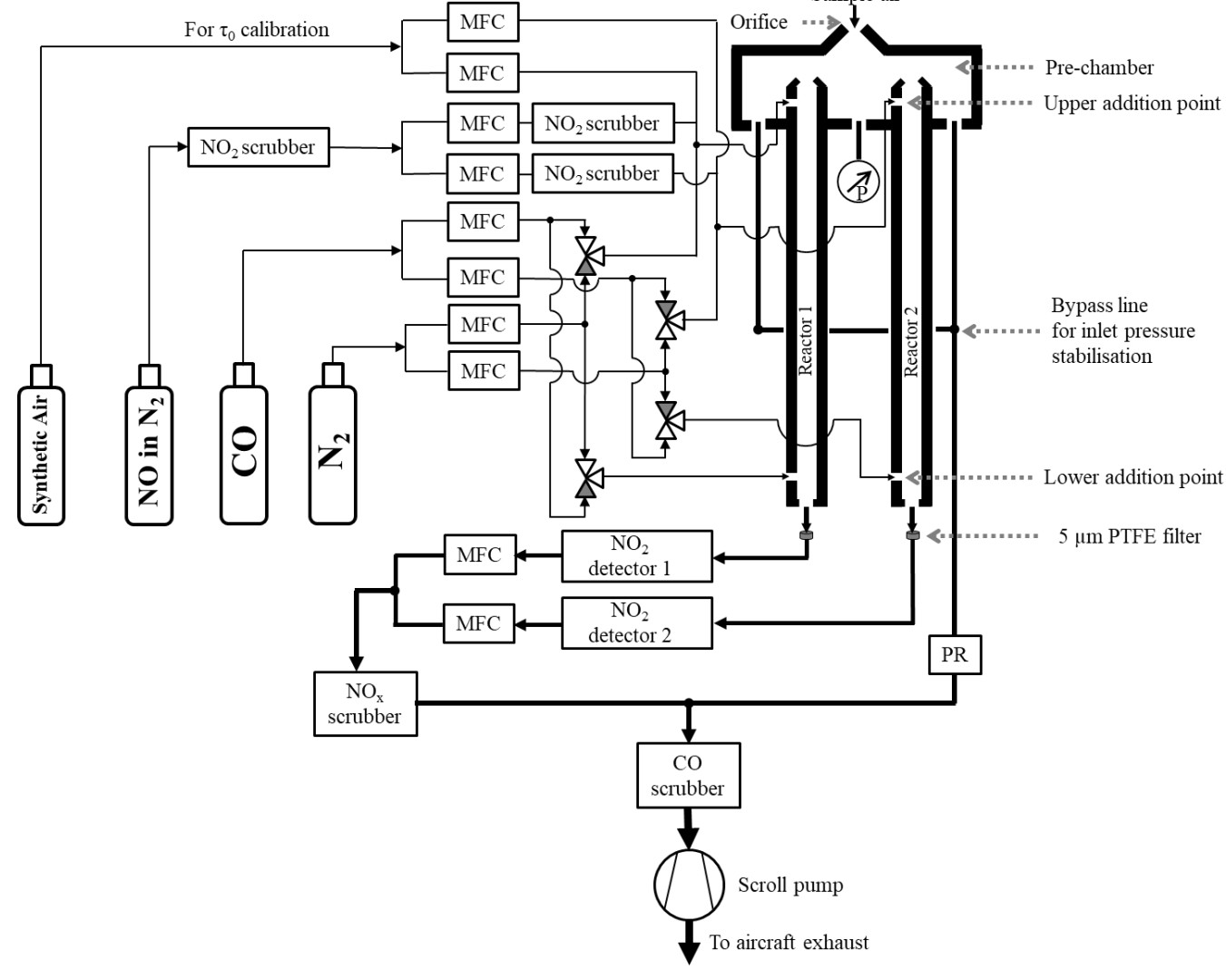

Figure 1: Schematic diagram of PeRCEAS instrument describing the gas flows, DUALER and detector subsystems.

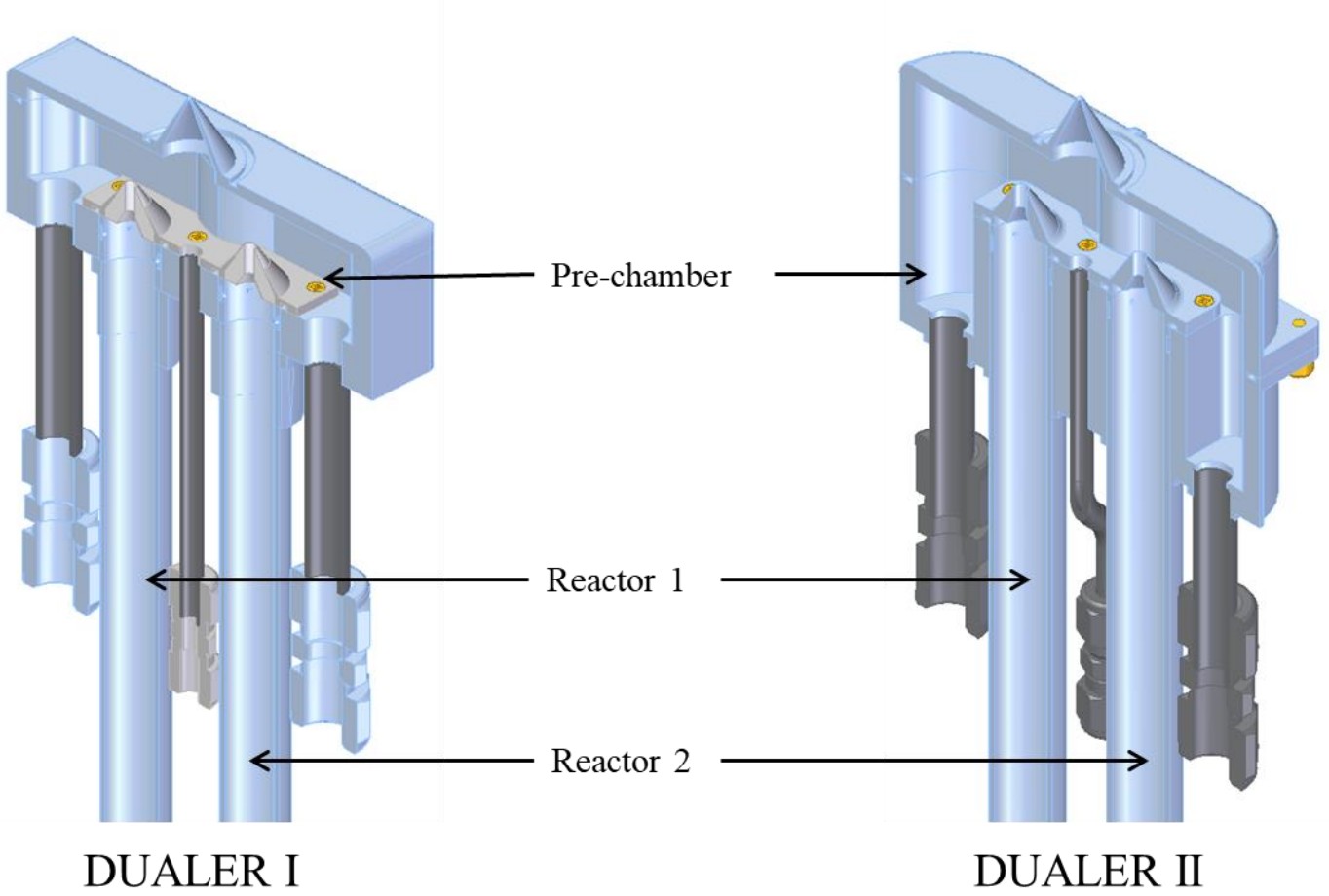

Figure 2: Graphical 3D representation of the upper part of the DUALER I and DUALER II inlets. The volume of the pre-chamber and the reactors of both detectors are: pre-chamber volume DUALER I = 75.25 cm$^3$; reactor volume DUALER I = 112 cm$^3$ and pre-chamber volume DUALER II = 119.57 cm$^3$; reactor volume DUALER I = 130.5 cm$^3$ respectively.

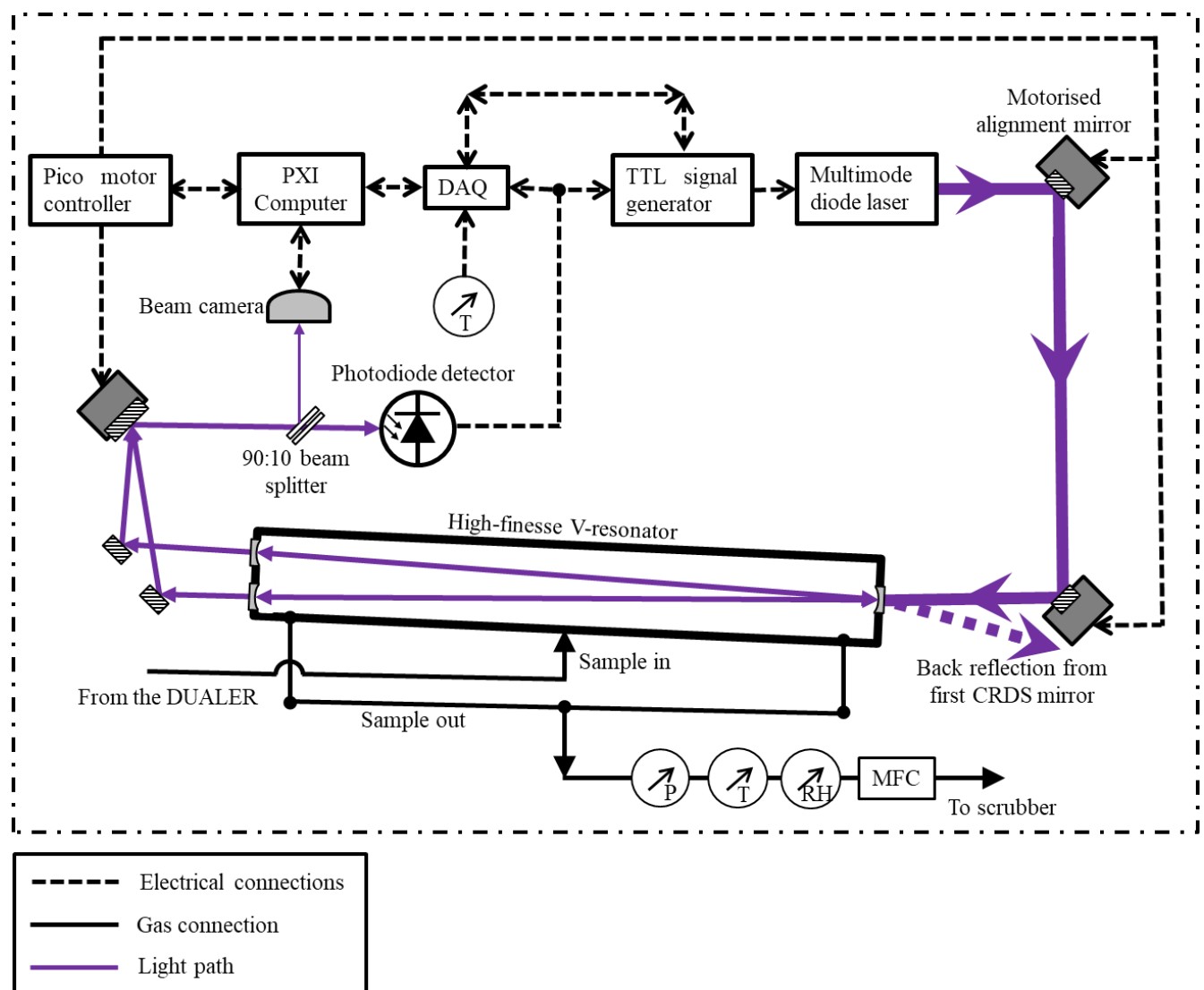

Figure 3: Schematic of the CRDS NO$_2$ detector

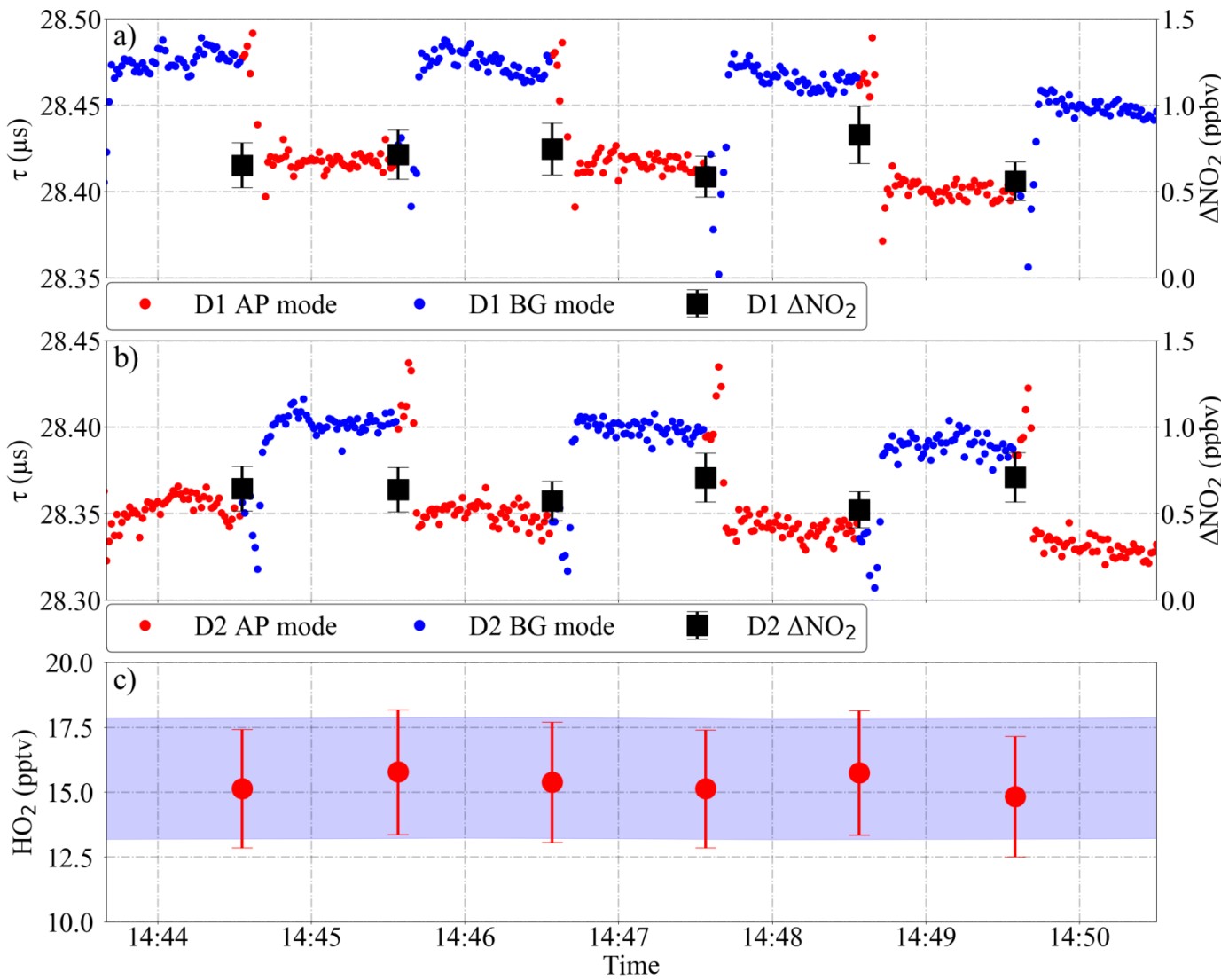

Figure 4: PeRCEAS measurement cycle during the laboratory measurement of 15 pptv of $HO_2$. a) and b) show the ring down time of detector one (D1) and two (D2) in both amplification (AP) and background (BG) modes and the retrieved $\Delta NO_2$. The $\Delta NO_2$ and the eCL of the respective reactors are used to retrieve the $HO_2$ mixing ratio in c). The blue shading in c) corresponds to the calculated $HO_2$ mixing ratio produced in the source ($2\sigma$ uncertainty).


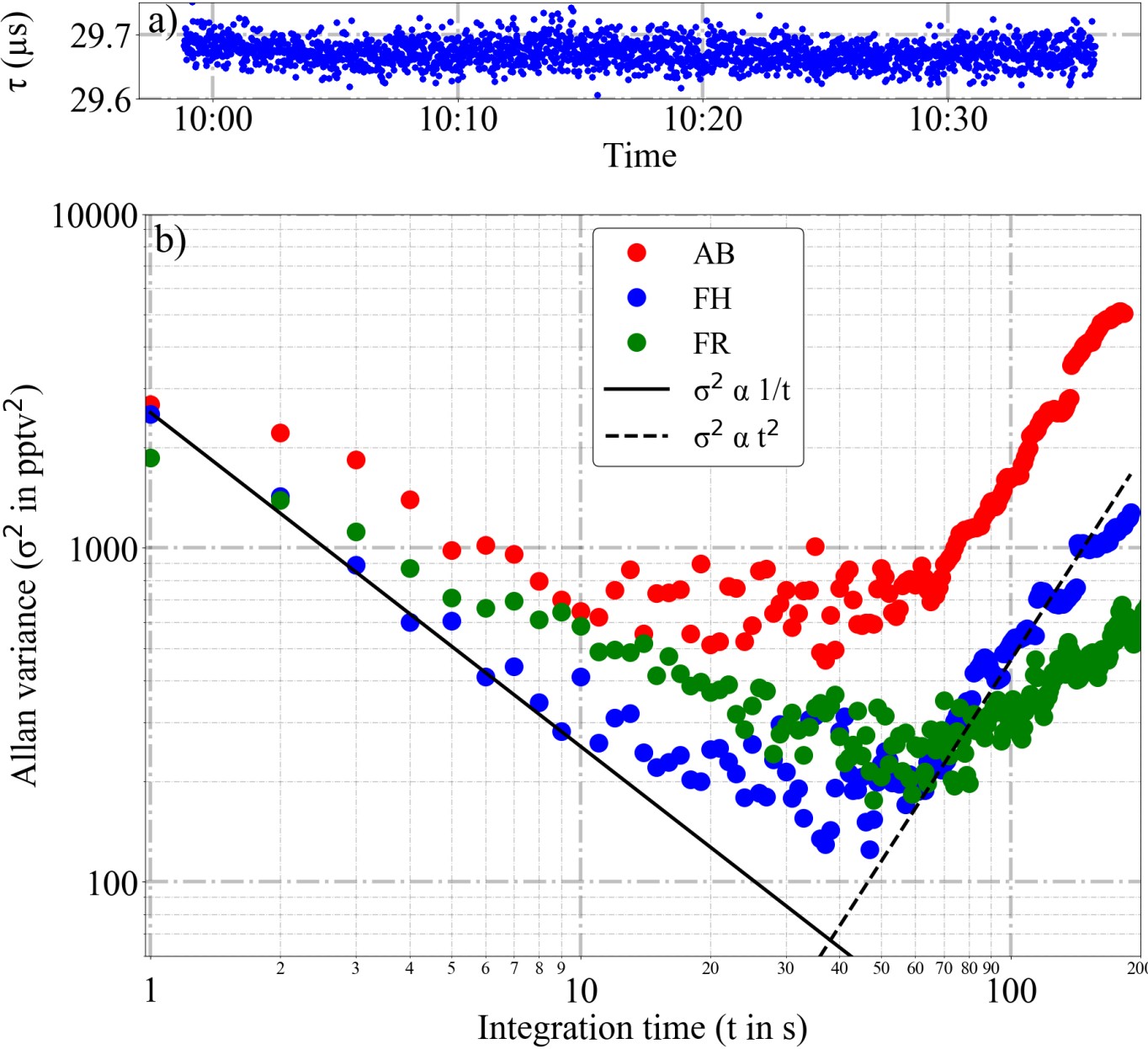

Figure 5: Analysis of the Allan variance of PeRCEAS measurements made in the laboratory: a) 40 minutes of data from detector FH used for the calculations, b) Allan variance of the $NO_2$ measurements for a mixture of 5.6 ppbv of $NO_2$ in air at 200 mbar and 23 °C sampled by the PeRCEAS detectors: AB, FH, and FR. The solid and dashed lines show respectively the theoretical behaviour of random noise (i.e. photon shot noise) and the noise attributed to longer time scale drifts.

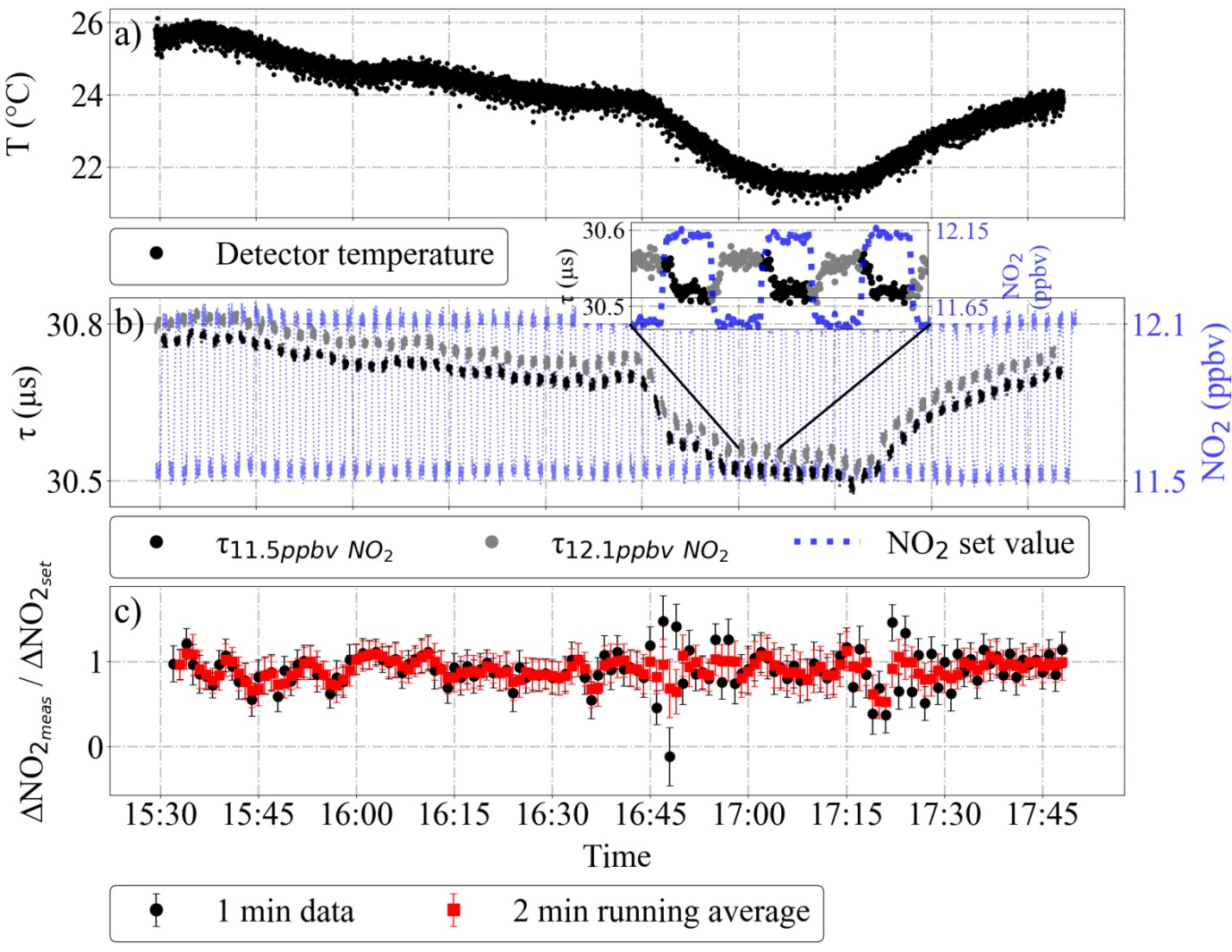

Figure 6: Effect of varying room temperature on ring down time $\tau$ and $\Delta NO_2$ accuracy: a) detector temperature b) $\tau$ for a modulated $NO_2$ flow and the corresponding $NO_2$ mixing ratios, and c) ratio of the measured to the set $\Delta NO_2$. The error bars in c) are estimates of the total uncertainty of the retrieved $\Delta NO_2$. The inset into b) is a magnification of three modulation cycles. The first 20 s of each signal after a change in the $NO_2$ mixing ratio are not used in the analysis (see text).

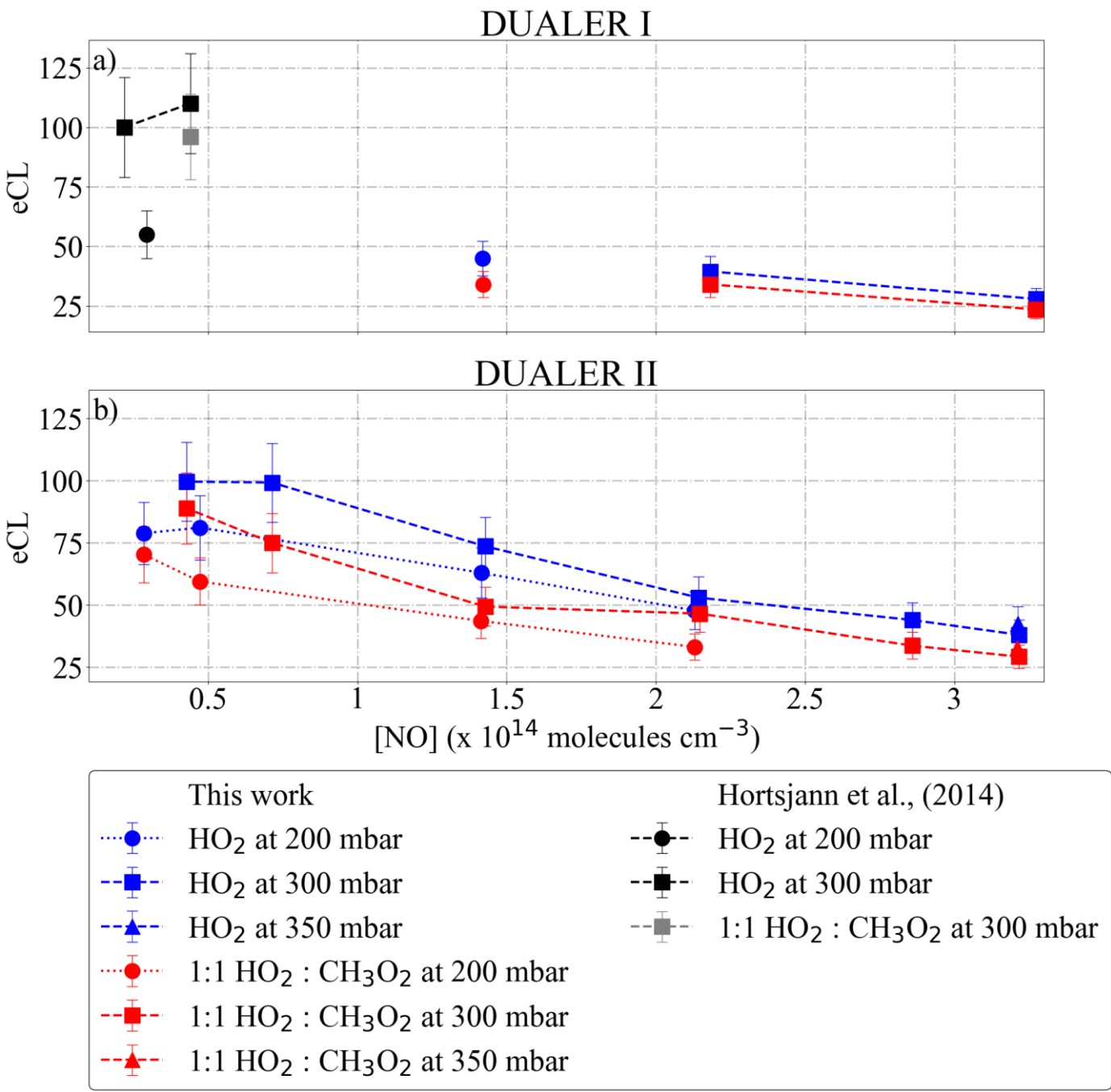

Figure 7: eCL versus [NO] measured a) DUALER I and b) DUALER II at inlet pressures between 200 and 350 mbar. The radical source flow tube is held at a pressure of 500 mbar. The values from Horstjann et al., 2014 are also depicted for comparison.


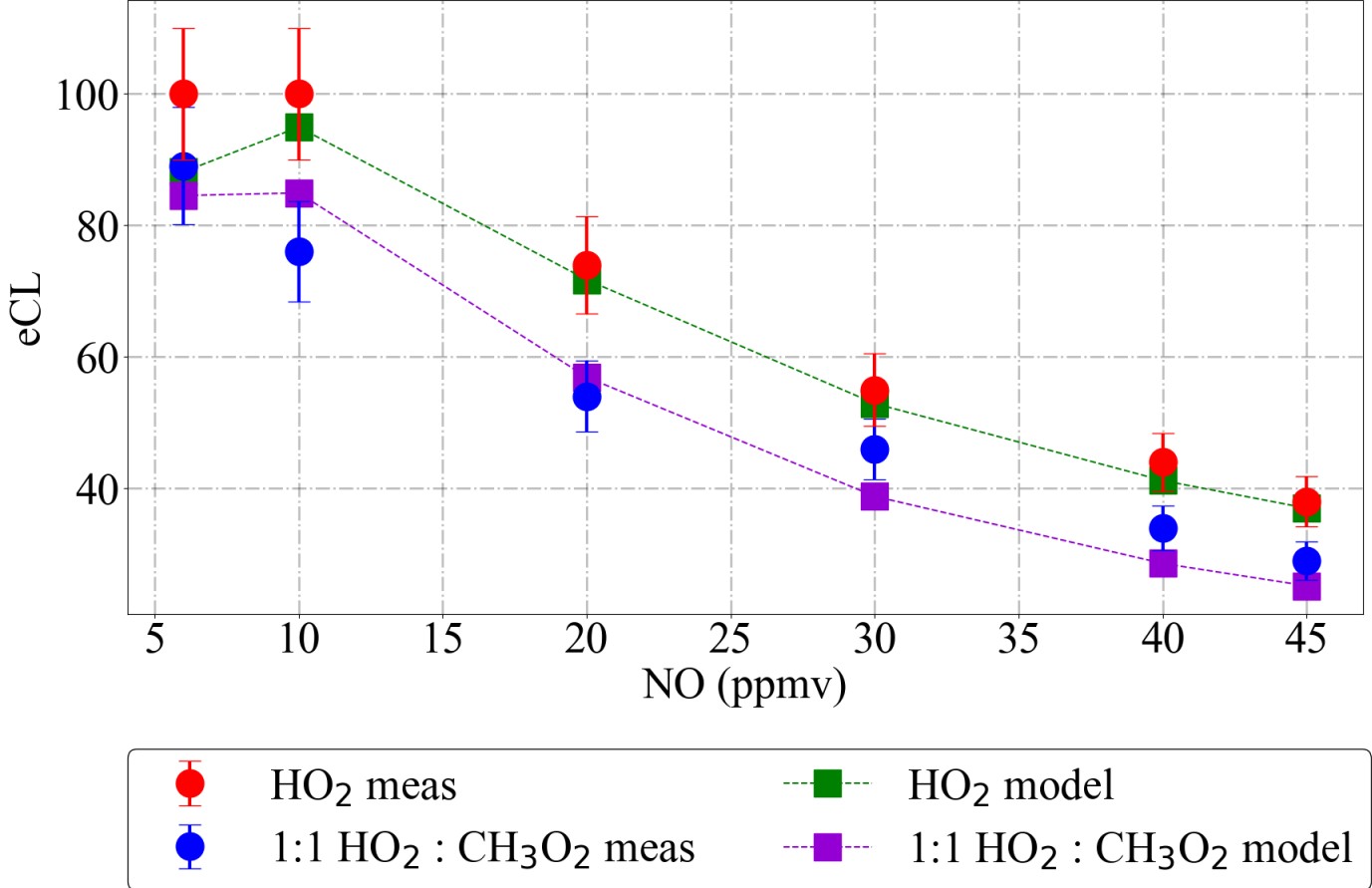

Figure 8: PeRCEAS eCL values retrieved experimentally at 300 mbar for DUALER II having different NO mixing ratios for HO$_2$ (blue triangles) and a 1:1 HO$_2$ : CH$_3$O$_2$ radical mixture (red squares). Simulated values of eCL obtained from the model for the same conditions are also depicted for comparison. The simulations use calculated values of $k_{w_{HO_2}} = 0.97$ s$^{-1}$ and $k_{w_{CH_3O_2}} = 0.74$ s$^{-1}$, and assume 64 % HO$_2$ and 54 % CH$_3$O$_2$ radical losses in the pre-chamber.


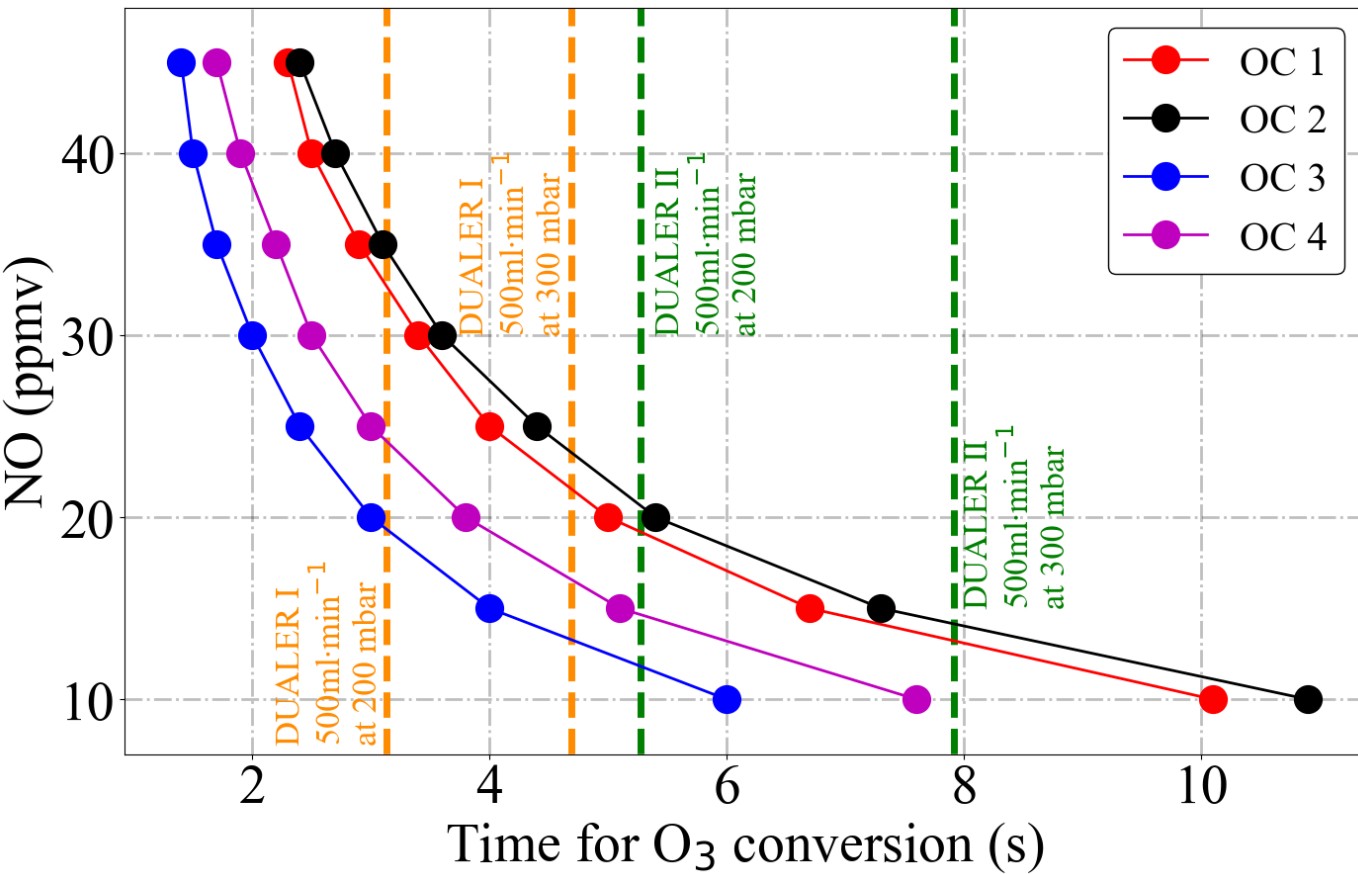

Figure 9: Time evolution of the $O_3$ decay for different NO mixing ratios added at the PeRCEAS reactor as simulated by a box model for 200 and 300 mbar. OC1: 100 ppbv $O_3$ at 200 mbar inlet pressure; OC2: 200 ppbv $O_3$ at 200 mbar inlet pressure; OC3: 100 ppbv $O_3$ at 200 mbar inlet pressure; OC4: 200 ppbv $O_3$ at 300 mbar inlet pressure. The sample residence times for 500 ml/min sample flow in the DUALER I and II are also depicted for reference.


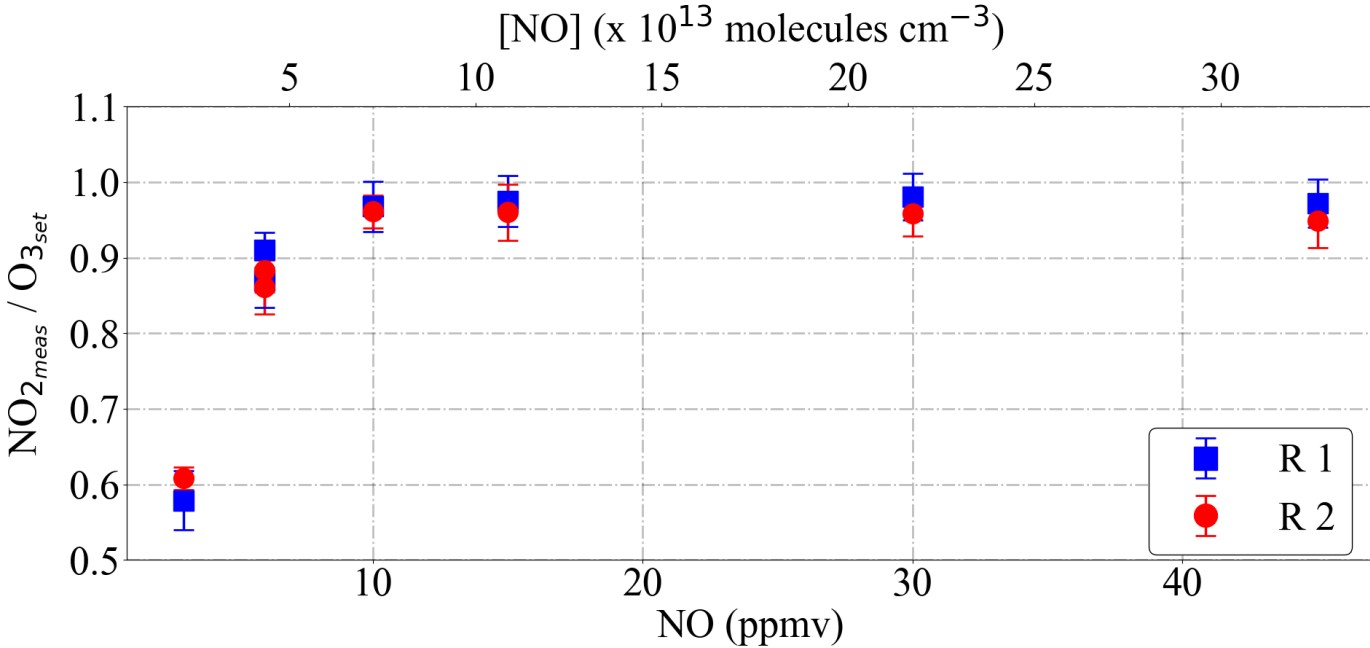

Figure 10: PeRCEAS measurement of $O_3$ mixing ratios up to 100 ppbv for different [NO] in the addition gas using DUALER II. NO is scaled in ppmv and molecules $cm^{-3}$. The $O_3$ conversion is completed when the ratio $NO_{2\ measured}$ / $O_{3\ set}$ at the calibrator reaches unity. R1: PeRCEAS reactor 1 (blue); R2: PeRCEAS reactor 2 (red).


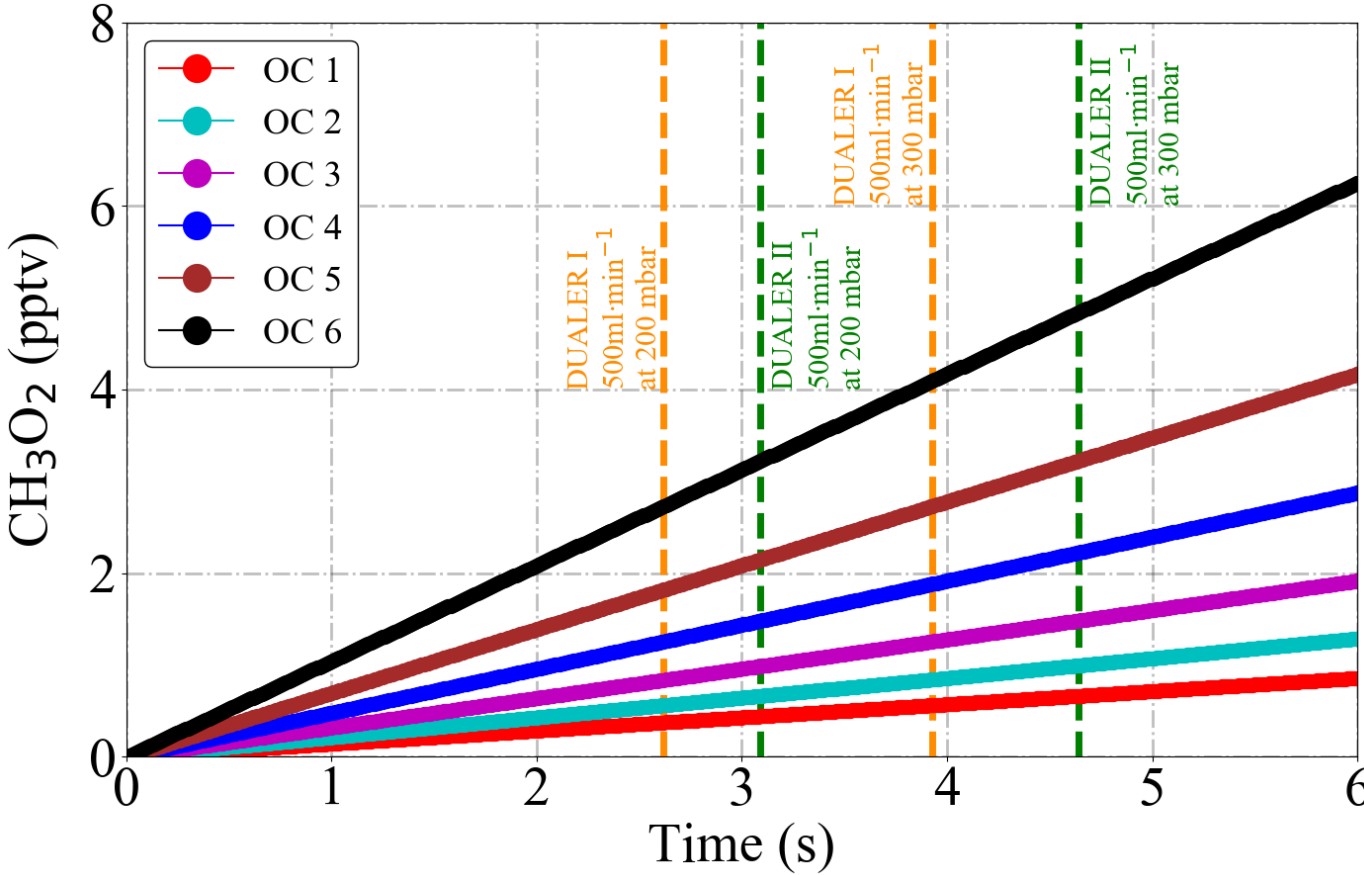

Figure 11: CH$_3$O$_2$ radical production from the thermal decomposition of 1 ppbv PAN as simulated by a box model between 288 K and 298 K at 200 and 300 mbar. OC1: 288 K and 300 mbar; OC2: 288 K and 200 mbar; OC3: 293 K and 300 mbar; OC4: 293 K and 200 mbar; OC5: 298 K and 300 mbar; OC6: 298 K and 200 mbar. The sample residence times for 500 ml/min sample flow in the DUALER I and II are also depicted for reference.


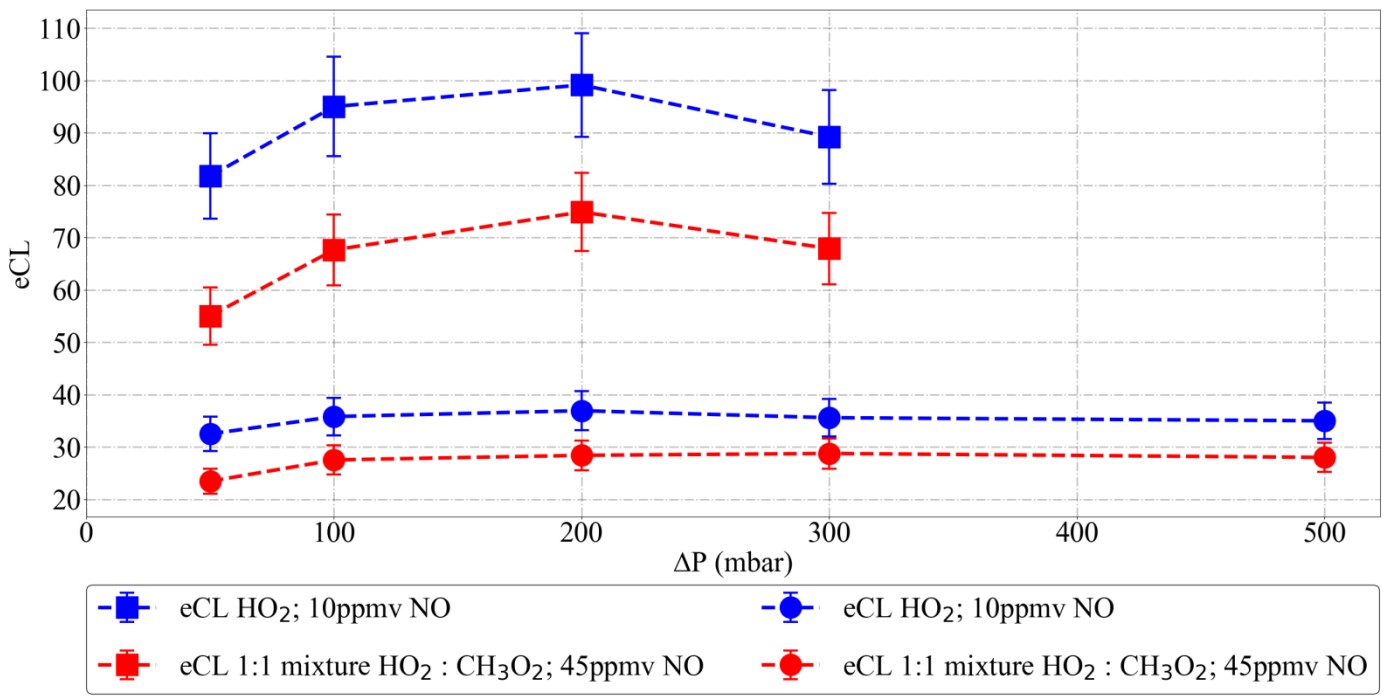

Figure 12: Dependency of eCL on $\Delta P$ ($\Delta P = P_{ambient} - P_{inlet}$) as determined for PeRCEAS under controlled laboratory conditions for 10 (squares) and 45 (circles) ppmv NO at 300 mbar inlet pressure. The error bars are $1\sigma$ deviation of eCL values obtained by identical calibrations at each $\Delta P$.

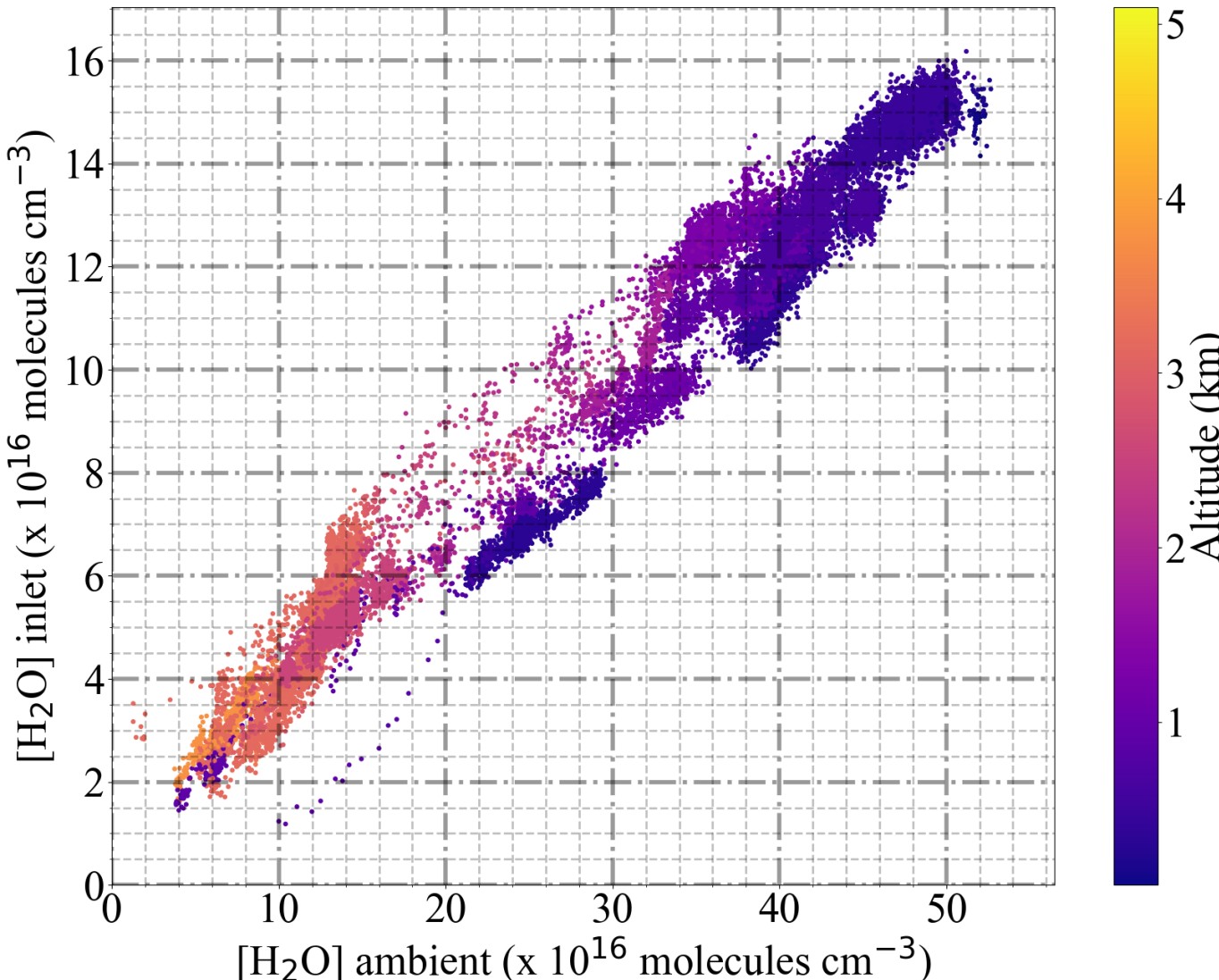

Figure 13: Comparison of the [$H_2O$] measured by the BAHAMAS instrument on board HALO and inside the DUALER inlet on the 17.03.2018 during the EMeRGe campaign in Asia. The colour scale indicates the altitude of the aircraft during the measurement.

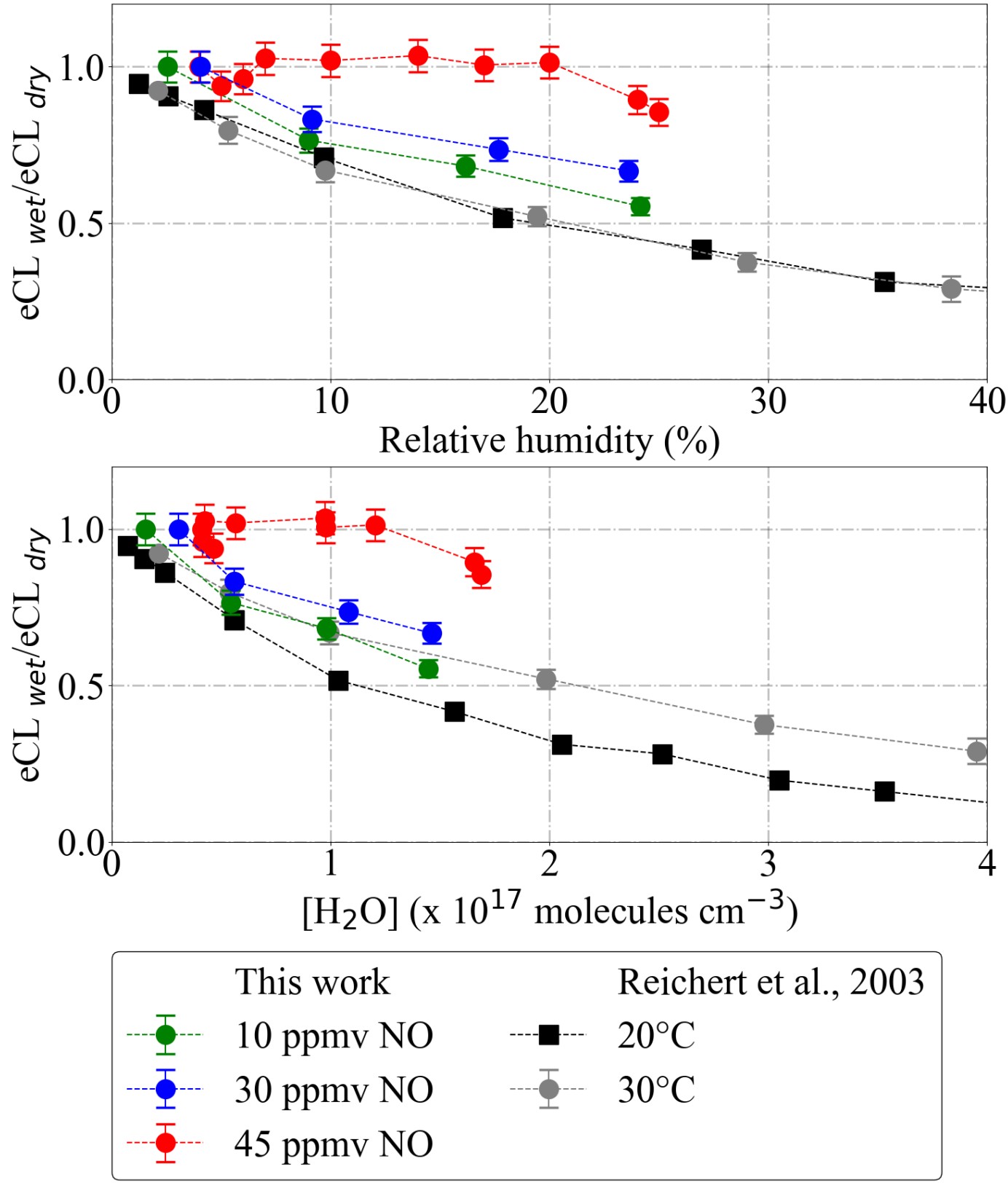

Figure 14: Dependency of PeRCEAS eCL a) on inlet humidity, and b) on [H$_2$O], at constant sampling flow, inlet pressure, ΔP, [CO] and [N$_2$], measured at 300 mbar for 10 ppmv (green), 30 ppmv (blue) and 45 ppmv (red) NO (respectively 7.29 x 10$^{13}$, 2.19 x 10$^{14}$ and 3.28 x 10$^{14}$ molecules cm$^{-3}$ [NO]) . The values from Reichert et al., 2013 obtained for 3.3 ppmv NO at standard pressure (8.12 x 10$^{13}$ molecules cm$^{-3}$ [NO]) are also plotted for comparison.

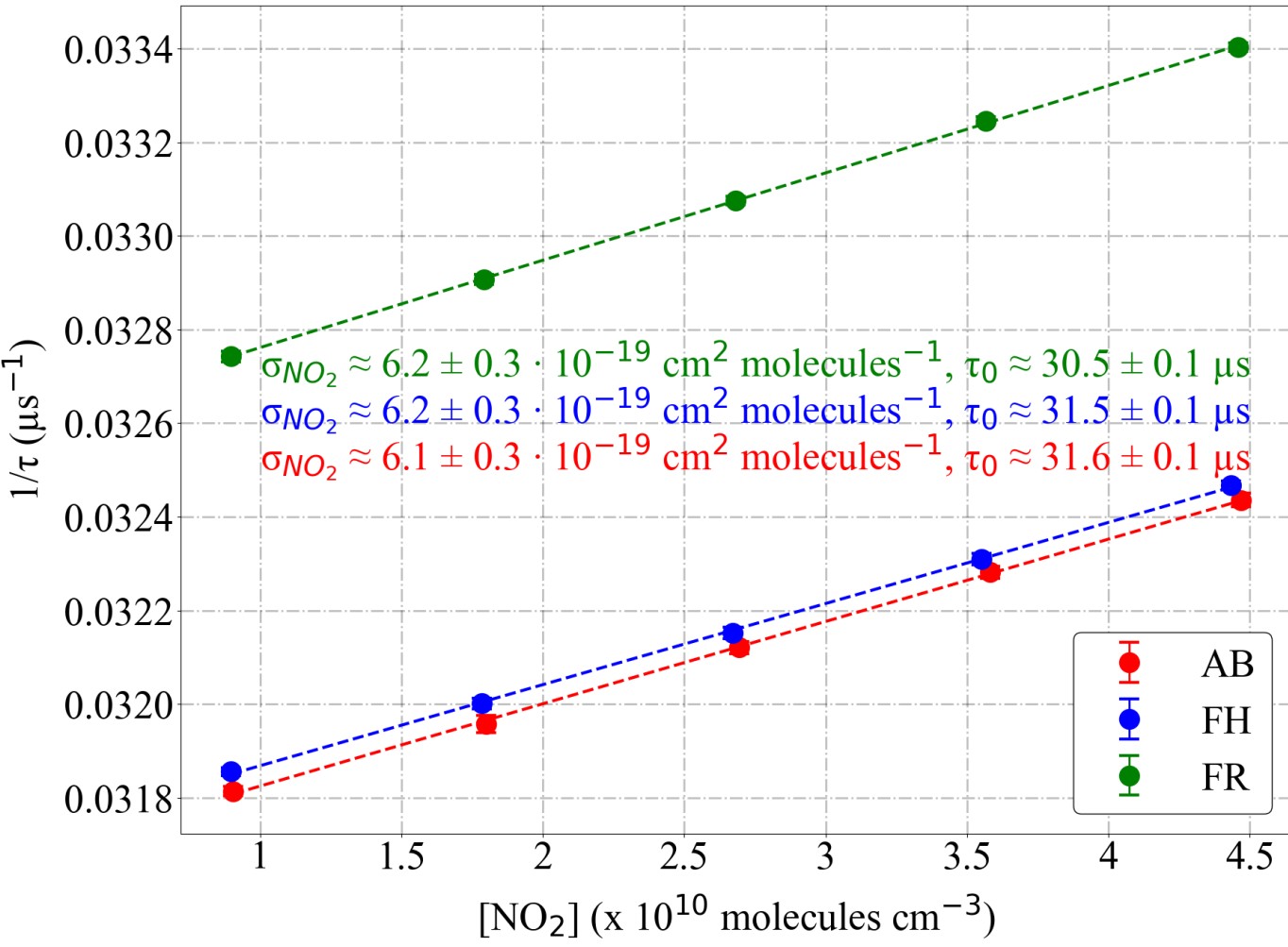

Figure 15: Determination of the effective absorption cross section, $\sigma_{NO_2}$ from $NO_2$ calibrations for the PeRCEAS detectors: AB (red), FH (blue) and FR (green) for measurements at 200 mbar. Linear fits are also shown by dashed lines.

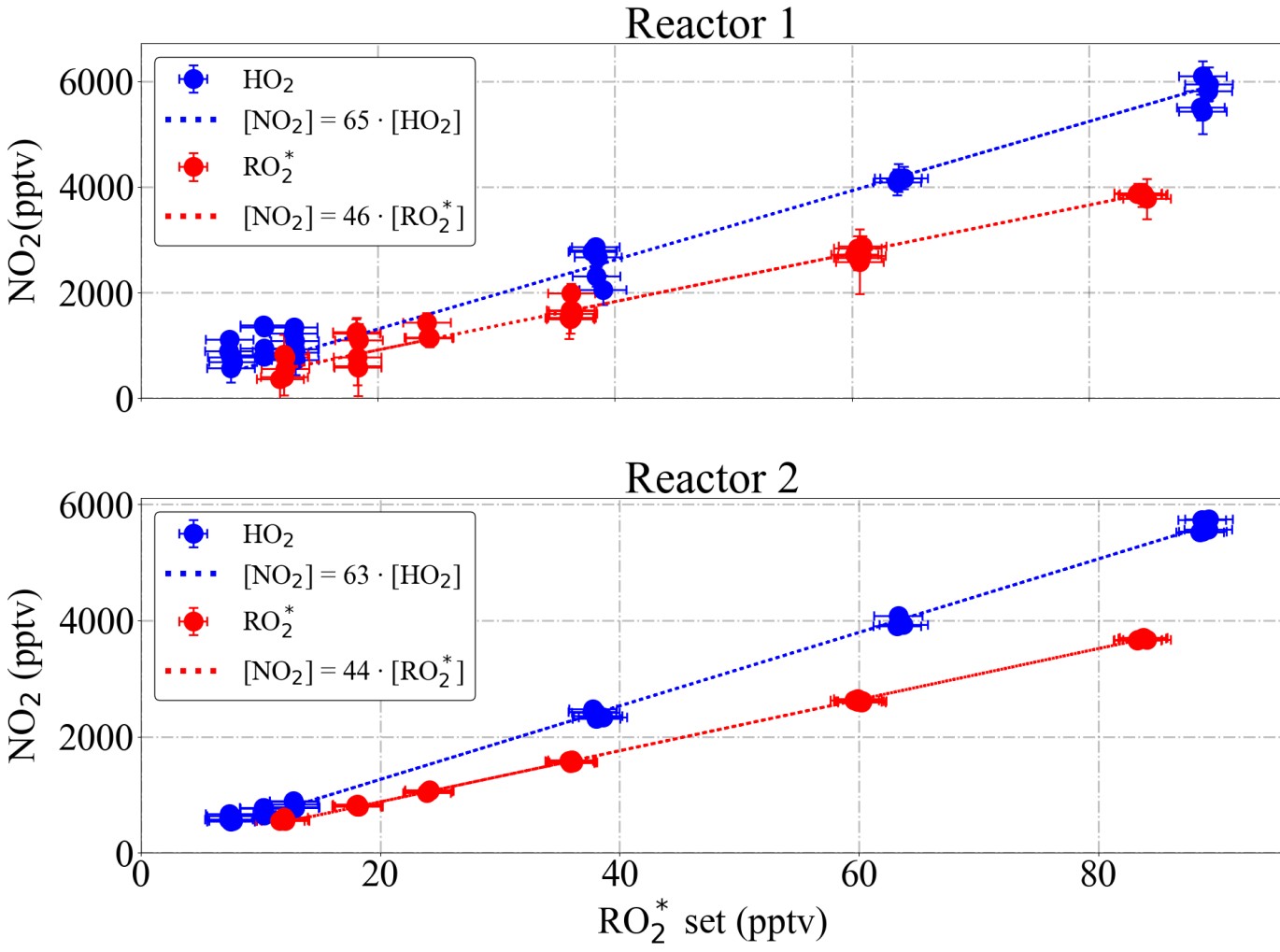

Figure 16: Experimental eCL determination of the DUALER II reactors from a series of 6 calibrations with generated mixing ratios of $HO_2$ (in blue) and a (1:1) $HO_2$: $CH_3O_2$ radical mixture (in red) at 200 mbar inlet pressure, 300 mbar $\Delta P$, and NO 30 ppmv within the inlet.

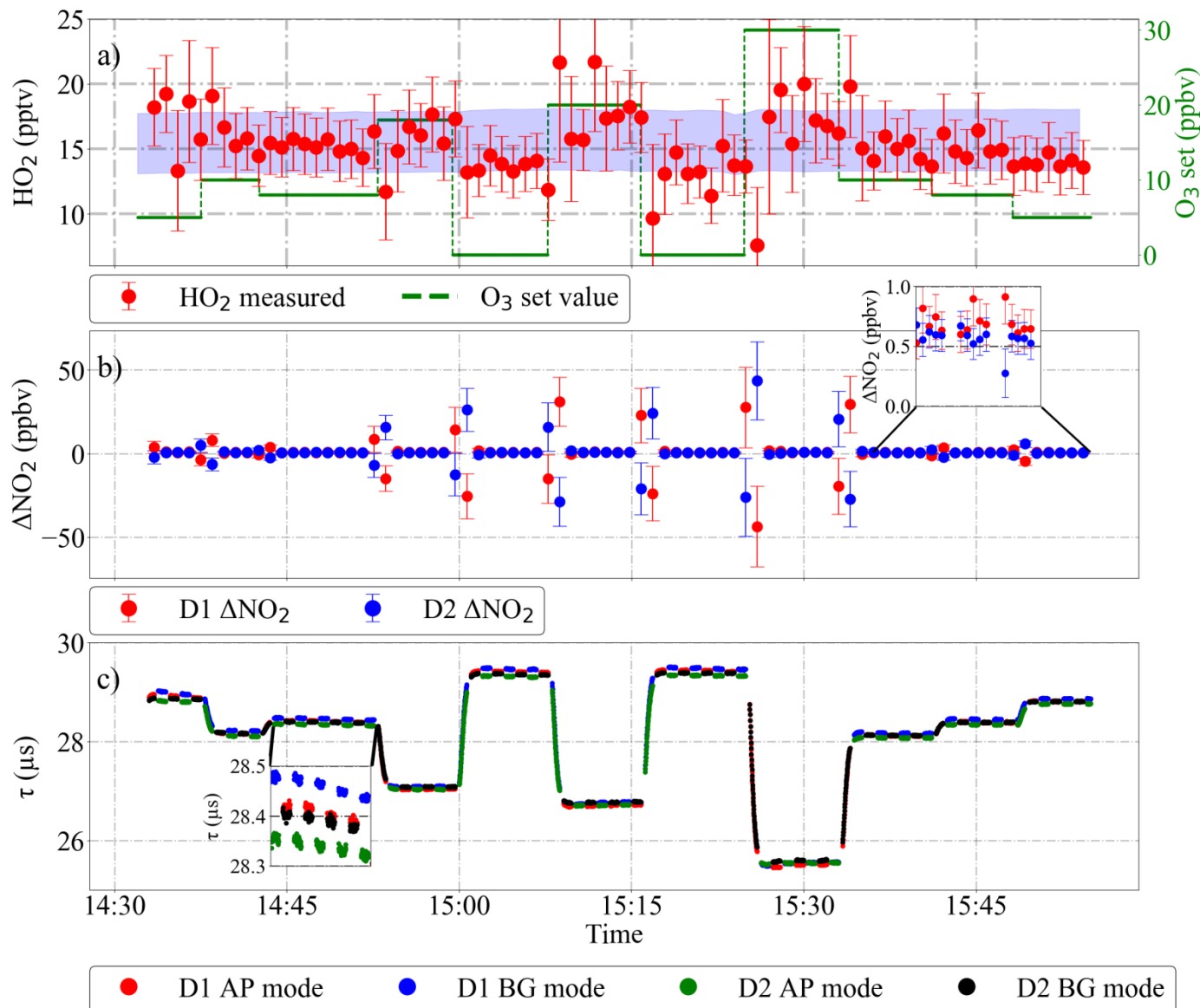

Figure 17: HO₂ retrieval under controlled changing of the O₃ background concentration using DUALER I: a) retrieved HO₂ and O₃ variation. The blue shaded area in a) shows the HO₂ produced in the radical source (15 % , i.e. 2σ, uncertainty); b) ΔNO₂ retrieved from detector 1 (red) and detector 2 (blue); c) ring down time from both detectors. D1: detector 1; D2: detector 2; AP: amplification mode, BG: background mode.

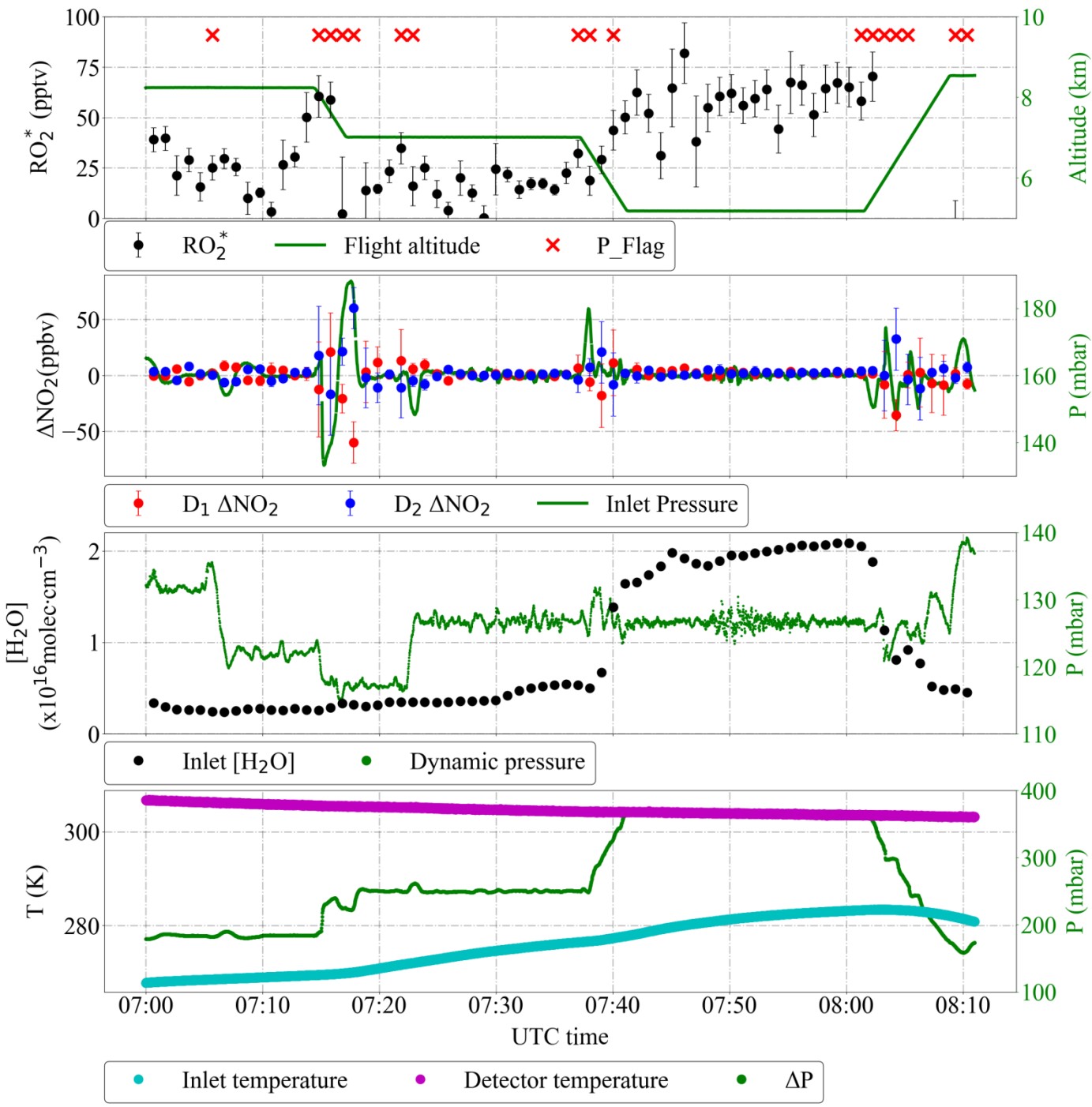

Figure 18: RO$_2^*$ PeRCEAS airborne measurement during the OMO flight on 25.08.2015. The DUALER I inlet was operated with 15 ppmv NO and at 160 mbar. Pressure variations with 1 min standard deviation >2 mbar are flagged (red crosses).


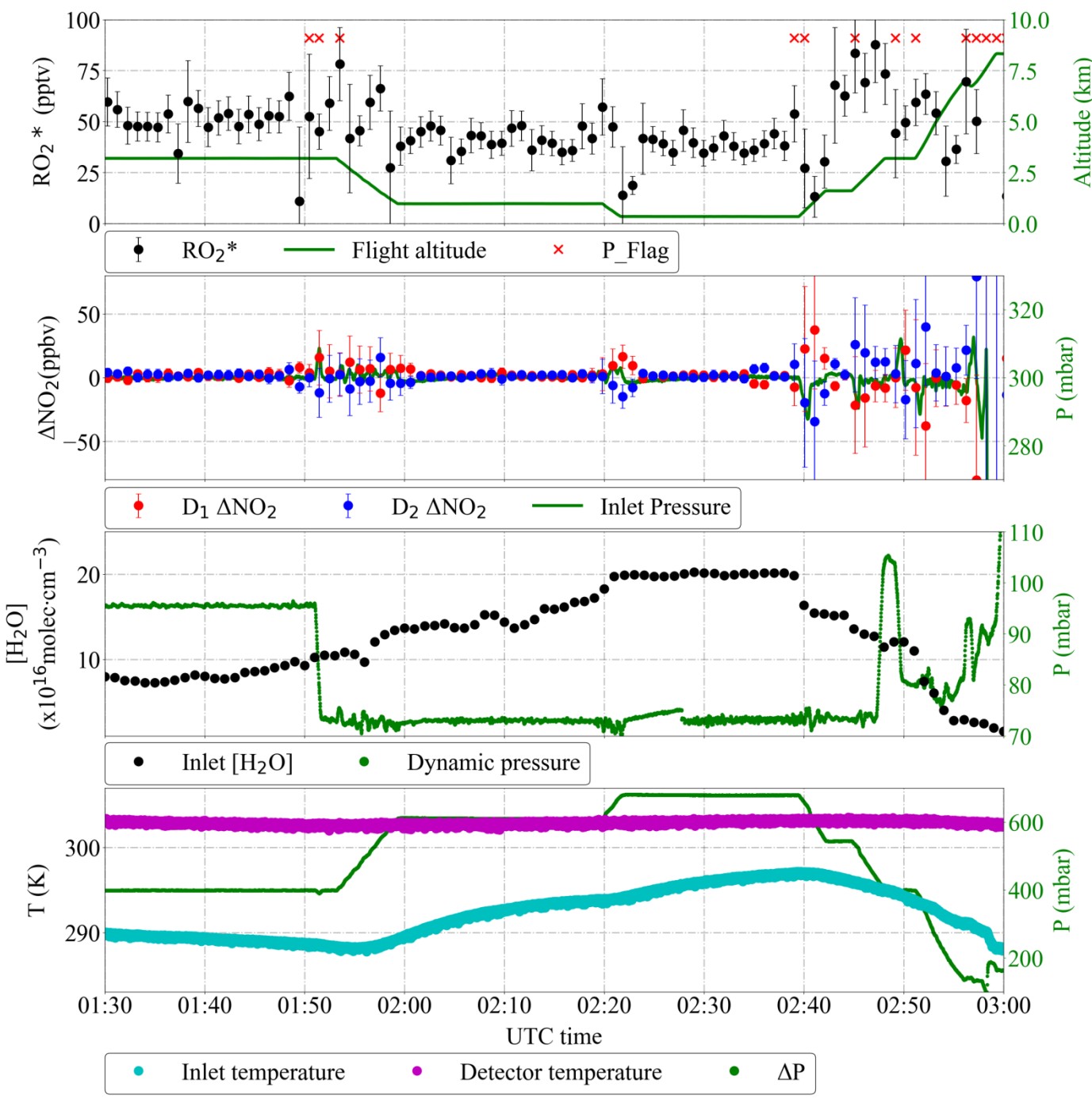


Figure 19: Detail of PeRCEAS measurements during the EMeRGe in Asia flight on 19.03.2018. The DUALER II inlet was operated with 45 ppmv NO and at 300 mbar. Pressure variations with 1 min standard deviation >2 mbar are flagged (red crosses).