# Peer review of "Airborne measurement of peroxy radicals using chemical amplification coupled with cavity ring down spectroscopy: the PeRCEAS instrument"

_Atmospheric Measurement Techniques, 2019_

## Referee Comment (RC1) · Anonymous Referee #1 · 19 Nov 2019

Review of "Airborne measurement of peroxy radicals using chemical amplification coupled with cavity ring down spectroscopy: the PeREAS instrument" by George et al.

This paper describes characterization of and sample data from an instrument based on chemical amplification and cavity ring down spectroscopy to measure the sum of peroxy radicals ($HO_2+RO_2$). Described are several instrument tests, laboratory evaluations and box modeling of the instrument performance and optimization. These include assessing the effects of environmental conditions and instrument parameters on its behavior.

This is a useful report of the status of this instrument and should be published. This reviewer has several comments and questions related to topics ranging from general philosophical issues to minor typographical errors.

**General comments and suggestions.**

The researchers appear to accept the basic characteristics of the inlet/reactor as given. The paper does not justify the selection of flow rates, inlet/reactor volume and composition, and thus the corresponding reaction time. This reviewer sees this approach as flawed. Perhaps this has been well thought out, and previously published, but not included in this paper. Suggest that the authors include some discussion of why these parameters were selected, and what the compromises and advantages in their selection are. One aspect is that higher NO levels used in the studies reported in this paper (30-45 ppmv) result in lower sensitivity to $CH_3O_2$ (and other $RO_2$) compared to previous values because of a faster rate of $CH_3O+NO+M$. High NO also converts a greater fraction of ambient ozone to $NO_2$, although complete conversion is not necessary. Most other chemical amplifiers used NO reagent mixing ratios of 2 to 6 ppmv.

It is not apparent why the response time of the system is so slow. For example, the Figure 5 caption states that 20 seconds is eliminated after change in $NO_2$. Given a reactor transit time of 3 seconds, this seems extreme. There may be delays, which can be accounted for in data analysis, that are different than transition to the correct value after perturbations. Part of the answer may be in Figure 19, where step functions in the $O_3$ concentration result in perturbations lasting about 40 seconds. Also, in Figure 20, the pressure variations last for a long time during and after altitude changes. There are also pressure fluctuations even when the aircraft is not changing altitude. This implies need for better pressure control. Perhaps the PID parameters of the pressure controller have not been adjusted properly. This is very important to get correct. Though improved, such fluctuations are still apparent in Figure 23. They add unnecessary noise to the measurements. Suggest adding more discussion of the pressure control system (manufacturer, model, adjustment procedures) and the response time of the system to step changes in $NO_2$ concentrations to allow the reader to better understand these issues.

Finally, why are the chain lengths reported in this paper so much lower that previous publications (Table 3)? In fact, values reported in this paper (28-38, Figure 18) are much lower than what is reported under "This Work" in Table 3 and would be the lowest values for the CO/NO chemical amplifier in the table (there are lower values for the ethane/NO chemical amplifier). This compromises the potential quality of the measurements. Perhaps this bears on the question about optimization of the instrument earlier. It seems that chain lengths of 100 or more are possible (at lower reagent NO mixing ratios). Some explanation in the paper is needed to explain this.

The English in this paper is really quite good, but there are few suggestions for improvement given in the specific comments below.

**Specific comments and suggestions.**

Page 1, line 16. "…for the airborne measurements in the…"

Page 1, line 18. "…instrumental channels successfully captures short term…".

Page 1, line 20. Not sure why the word "gradients" is used here. How about "…range of atmospheric pressures and temperatures expected…"?

Page 1, line 24. The phrase "…collectively known at $RO_2^*$…". Is it true that $HO_2+RO_2$ is "known" as $RO_2^*$? Is this the term accepted by the community? This reviewer suggests just using "$HO_2+RO_2$" instead.

Page 1, line 29. Suggest removing the summation symbol, since there is a plus sign used.

Page 1, line 32-33. Suggest "…photolyzed to ultimately produce…"

Page 2, line 34. Suggest "Overall, $HO_2+RO_2$ influences the…"

Page 2, line 41. Not sure what is meant by "those $RO_2$".

Page 2, line 43. Suggest "…compared to the total amount of $HO_2+RO_2$, with $HO_2$…"

Page 2, line 53. Should be "Kanaya".

Page 2, line 54. Suggest a different word than "largely". Suggest "The interference by some $RO_2$…"

Page 3, line 72. Add space "Peroxy Radical…".

Page 3, line 74. Suggest "…in a previous publication…".

Page 3, line 79. Suggest "…where $\Delta NO_2$ is the $NO_2$ formed…"

Page 3, line 84. Suggest "…spectroscopic measurement technique…"

Page 3, line 87. $NO_2$ comes from radical amplification and from the background ($O_3$ conversion). Suggest changing the sentence to reflect this.

Page 3, line 92-94. Suggest also saying that $c_0$ is the speed of light in a vacuum (stated later in the paper).

Page 3, line 92. Suggest "…are, respectively, the absorption…"

Page 3, line 95. Suggest "…used for ground-based measurements…"

Page 3, line 96-7. Suggest "…the particular constraints related to airborne measurement…".

Page 3, line 99. Suggest "In this study, the specifications…are described based on thorough laboratory…"

Page 4, line 107. Suggest (if correct) "…and located outside the HALO fuselage…"

Figures 1, 2, and 3. While photographs can be nice, schematic diagrams are more useful to see the path of sample, reagents, and signals throughout the system. Suggest limiting to only one or two photographs and add diagrams to show the details.

Page 4, line 117. Mixing and pressure regulation are mentioned here, but no detail is given. This is relevant to the general comment given earlier. Suggest adding more detail in the text and perhaps in Figures 1 and 2. Suggest discussing how DUALER I and II are different and describe why the changes do indeed result in improved performance.

Page 4, line 124. The term "piezo electric stack" is not a common term and needs more description.

Page 4, line 128. Suggest adding the manufacturer of the beam camera (MKS Ophir). This is very cool, by the way!

Page 4, line 132-3. Suggest "…other sensor data such as pressure, flow, temperature, and humidity." You don't need "etc." if using "like" or "such as".

Page 5, line 149. Would "occasionally" be better than "exceptionally"?

Page 5, line 150-152. The term that should be used is "Allan Variance" rather than "Allan Deviation". Perhaps the equation for its calculation should be given, since there is also modified Allan Variance that can be used. Also, give one or two references (e.g. Allan, 1966; Allan et al., 1991; Allan and Levine, 2016). Suggest "…was investigated using calculated Allan Variance in the measurement…". Also "…the optimum integration time for the three PeRCEAS detectors is between 20 s…". Do you use 20-30 second averaging in the data analysis? The timing of the instrument cycling should be shown, perhaps in Figure 1 or in a separate figure.

Page 5, line 155. The phrase "…over the modulation time…" need explanation. This might be apparent with addition of a figure showing the instrument cycle timing.

Page 5, line 161. Suggest "…signals generated that minutely varied the sampled $NO_2$…"

Page 5, line 164. "…time resolution using each background…". This will be more obvious with a graphical representation.

Page 5, line 167. "… molecules $cm^{-3}$ for typical measurement conditions".

Page 5, line 170. "It is inferred from laboratory calibrations that a 60 s modulation cycle is an optimum comprise between…". The term "modulation cycle" is not apparent here, but perhaps would be with a graphical representation of the instrument cycle. Also, add more discussion why fluctuations last so long (20 s).

Page 5, line 172. The detection for $NO_2$ should be performed with a background level of ozone in the sample, since this is how ambient measurements are performed. Was this is the case?

Page 5, line 173-4. It is not obvious that larger modulation times lower the representativeness of the averages. Do you mean variability in peroxy radicals or in the background? For the latter, the instrument is continually measuring the background, and it should be well accounted for. If you mean the peroxy radicals, while one-minute (or quicker) data are nice, longer averages can still be useful in adding understanding of tropospheric free radical behavior. Suggest rewording the last sentence of this paragraph.

Page 6, line 176. "Sample and reagent gas flows...".

Page 6, line 176-184. This discussion of reaction time should also include discussion of how the size of the reactor was selected, since it also affects the reaction time (reactor volume / total flow). Perhaps this would be a good place to discuss the approach to ensuring mixing of reagent gases with the ambient air sample. How was this done? Were fluid dynamical calculations performed? Were flow visualization approaches used? Related to this: how do you ensure that no components of the inlet system are leaking?

Page 6, line 182. "…lower explosion limit (LEL) in air of 12.5% v/v at room temperature…".

Page 6, line 189. Remove extra space between "air" and "and", and between "CO" and "in".

Page 6, line 190. Suggest "…between safety requirements, limiting…".

Page 6-7. Effective chain length. This might be good place to discuss experiments to determine the optimum NO concentration for the amplifier chemistry. Also, perhaps near the end of this section, discuss how the effective chain length values are used in the data analysis. In other words, have estimates of the $HO_2/RO_2$ ratio been made and used to apply the two eCL (for $HO_2$ and $HO_2+CH_3O_2$) values? If so, how is it done?

Page 6, line 197. "…to the conversion into $RO_2$.". Include that the approach is based on $O_2$ actinometry, as opposed to other approaches reported in the literature (such as $N_2O$ actinometry, calibrated NIST photodiodes).

Page 6, line 205. "…are changed stepwise every ten minutes from 8 pptv…". Also, note that too much reagent added to the calibrator has the potential to affect the inlet chemistry. This can be seen by a change in the background with change in radical concentration, which is not expected if the background is mostly due to ozone.

Page 6, line 206-7. "…is determined from the slope of the measured ΔNO$_2$ levels versus the calculated radical amounts. Example data is shown in…". Suggest rewording the end of the last sentence on this page, since the concentration of NO ***within the inlet*** is 30 ppmv. Perhaps "…and added reagent NO to achieve 30 ppmv within the inlet."

Page 7, line 208. Suggest "In Figure 7, the PeRCEAS eCL versus the inlet NO concentration…". In Figure 7, why aren't data shown for lower NO concentrations, such as used by your group in the past and by other researchers?

Page 7, line 212. "…concentration, eCL values increase with…".

Page 7, line 16-18. Suggest describing how wall losses were determined. Are they constant or are they affected by the cleanliness of the inlet? Suggest putting all the rate coefficients used for both reactor pressures into Table 1. It is interesting that a level of 3 ppbv O$_3$ was used, presumably because this is what comes out of the calibrator. Suggest also running the model with ambient-like levels of O$_3$.

Page 7, line 221. Inlet pre-chamber is not defined anywhere. This should be shown in the schematic diagram discussed earlier. If the radical losses in the model do not agree with what you think they are in the DUALER II inlet, suggest you perform experiments to determine what they are. The model of this simple chemistry should be much closer to the observations that a factor of two!

Page 7, line 222. Suggest "…shows measured eCL versus modeled CL for the…".

Page 7, line 223-4. "The CL$_{modeled}$/eCL$_{measured}$ ratio averages about 2 for HO$_2$…". Actually, the ratio is more than 2 for the 200 mB measurements and is about 2 for the 300 mB HO$_2$ measurements. Only the 300 mB HO$_2$+CH$_3$O$_2$ measurements are about 1.5. Does this mean that the inlet wall loss changes with reactor pressure? Were the rate coefficients in the model changed to reflect the reactor pressure? Perhaps the model wall loss values should be adjusted based on new laboratory measurements. The chemical amplifier chemistry is simple enough that a box model should be able to accurately reproduce laboratory data such as this. Add error bars to the points to represent total uncertainty in the measured and modeled values. Perhaps perform regressions of data.

Page 7, line 226. This reviewer does not like the term "titration" in this context, even though it is widely used in the community. Suggest using "conversion" instead.

Page 7, line 227. This reviewer disagrees that ozone in the sample has to be completely converted to NO$_2$. Why is this? It seems that partial conversion, as long as it is stable, would be fine.

Page 7, line 228. Figure 9 has a lot of information that could be presented in a more straightforward way. Suggest plotting the ozone lifetime (or three lifetimes) versus the reactor NO at the two pressures. This could be shown in one plot.

Page 7, line 229. There is no reason to require conversion of 100-200 ppbv of ozone to 1-2 pptv in the inlet. Conversions of 99% are more than sufficient. Suggest changing this paragraph as Figure 9 is changed.

Page 7, line 230. The wavelength 409 nm is mentioned, but everywhere else it indicates that the lasers operate at 408 nm.

Page 7, line 236. Suggest "…which are captured by…". This reviewer disagrees that the radicals and NO$_2$ from PAN-like compounds cancels and does not lead to interference. Yes, the NO$_2$ from the decomposition should be like ambient NO$_2$ and be corrected for by the background measurement.

But the radicals formed from the decomposition will amplify and appear like ambient radicals. This is an interference! Suggest rewording this paragraph. Some direct laboratory measurements of the interference would also be helpful. Figure 11 shows that the PAN interference is greater at lower reactor pressures. Why would this be the case. At reduced pressure, the decomposition is slower and the time is the reactor is shorter. Suggest checking the modeling.

Page 8, line 259-260. It is not clear what is meant by "based on the similarity of the eCL values". Suggest rewording this sentence, and perhaps this entire paragraph to make the message clearer.

Page 8, line 263. Suggest replacing "as shown exemplary" with "with an example shown".

Page 8, line 265. "…sample humidity do not lead to…".

Page 8, line 267. "…is subject to two types of errors which either are: a) intrinsically…"

Page 8, line 268. "…in the laboratory, or b) result…"

Page 8, line 273. Equation 3 is very similar to Equation 1. Suggest eliminating Equation 3 and referring back to Equation 1 in this discussion. Perhaps change Equation 1 slightly, if needed.

Page 9, line 279. "…Vandaele et al. [2002] with the normalized laser spectrum from the corresponding detector.". Also, "The values obtained have been…"

Page 9, line 284. "…depicts a sample comparison of spectra…".

Page 9, line 294. "The effective $\sigma_{NO_2}$ obtained agrees within…".

Page 9, line 297. The tau symbol disappeared. You have a lot of ambient data. Does $\tau_0$ vary significantly as the CRDS cell mirrors are exposed to ambient air?

Page 9, line 300. "…measurement requires accurate…"

Page 9, line 302-3. "…are the radical calibration…". "…which is estimated to be…".

Page 10, line 307. "The errors associated…".

Page 10, line 312. "…reactor 2, respectively, and…".

Page 10, line 315-6. Delete "during the airborne measurement of $RO_2^*$".

Page 10, line 317. "The noise in the $NO_2$ signal is enhanced by…".

Page 10, line 319. "…cabin temperature could increase…".

Page 10, line 320. "…stability of the CRDS signal and the accuracy of the supporting measurements.".

Page 10, line 323-4. Again, this reviewer does not agree with this statement. Since you are continually measuring the background and the amplified signal plus background, variations should be accounted for. Only changes happening faster than one second should have influence, unless there is something about the data analysis that is not obvious from the presentation in the paper. Suggest looking into why step changes in ozone should affect the signal for more than a few seconds. Also, step changes are extreme for ambient measurements. Even changes as the aircraft changes altitude are likely to be gradual unless a pollution layer is encountered.

Page 10, line 335. "…a standard deviation of the order of…".

Page 11, line 353. "…detector signals can be significantly affected…".

Page 11, line 360. "…airborne measurements and is difficult to implement in…".

Page 11, lines 364-370. Do you mean running the inlet at reduced pressure results in lower eCL values? Doesn't PeRCEAS continually measure the signal and background as mentioned in the second line? What do differences in detector accuracy (do you mean sensitivity) do to affect their uncertainties? Suggest adding a reference to the last sentence of this paragraph (about $RO_2$ interferences in LIF).

Page 12, line 374. "Figure 20 shows sample data of $RO_2^*$ measured…".

Page 12, line 382. Mention that the flagged values are shown in Figures 20 and 23. Also mention this in the figure captions.

Page 12, line 386. "…in more detail in Figure 23…".

Page 12, line 388. Not sure what is meant by "the signal is not affected by altitude changes", since there are jumps in $\Delta NO_2$ when the altitude changes. Suggest rewording to make the point clearer.

Page 12, line 402. "…over a 60 s integration…".

Page 12, line 404. While PeRCEAS may be suitable for measurements up to 12 km, no data were shown at this altitude. Suggest rewording this sentence.

Page 12, line 405. "…campaigns onboard HALO.".

References

Line 449. "…peroxy radicals by chemical amplification…".

Line 452. Two references are together. Need carriage return after "1993".

Tables

Suggest heading for "second addition point" to be changed to "reaction times", and "to detector" changed to "transfer times".

Page 3, Edwards et al., Inlet pressure should be 200 mB.

Figures

Most of the figures need larger symbols and bolder lines (4, 5, 6, 10, 12, 14, 17, 18, 19, 20, 23). In many of the plots, the legend is covered by data. Suggest enclosing legend in a box with a white background.

See comments earlier about Figures 1-3.

Figure 2 caption. "…Top view of the…". "..the laser beam is highlighted (purple) for…". "…exiting the cavity is depicted."

Figure 4. Change y-axis to Allan Variance. Describe what the lines depict (linear fits to data less that 10 seconds?). Can you sample for longer times than 70 seconds? Suggest doing the analysis out to 10 minute averages or more.

Figure 5. It is not obvious what this figure is trying to show. It appears to this reviewer that the point is temperature changes affect the $\tau_0$ of the detector, but the retrieved $\Delta NO_2$ is affected very little. Why not do this experiment with two detectors as done for radical measurements? This would be a more realistic representation of the actual measurement situation.

Figure 6. Are the equations determined from linear fits? Are they standard or bivariate fits?

Figure 7. Why are not data shown for lower values of NO? Suggest more work going from 0 to 3.5E14 NO with at least 10 points for each instrumental condition (pressure and radical type). Perhaps also show the same y-axis for both plots.

Figure 8. Since it is mentioned in the text, suggest adding 1:1.5 line. As discussed earlier, perhaps more modeling with more realistic wall loss rates needs to be done.

Figure 9. Changes to this figure suggested earlier (plot $O_3$ lifetime versus NO). If it is kept the same, suggest labeling each sub-figure and referring to those labels in the caption.

Figure 10. This shows that 60% of the $O_3$ is converted with NO of 3 ppmv, and 90% at 6 ppmv. This means that the instrument could be run with much lower NO levels.

Figure 11. It is stated that PAN interference is not a problem with PeRCEAS, but this plot shows that with reaction times of 3 seconds (compared to 2.6 to 3.1 seconds for the two DUALER inlets), up to 6 pptv of peroxy radicals can be produced from 1 ppbv of PAN. Is this representative of the conditions for which the instrument has been used? This figure could be changed to plot the fraction conversion of PAN ($CH_3O_2$ produced / PAN) versus temperature for the two DUALER reaction times.

Figure 13. There are places where the ambient water is below the inlet water. How can this be? This figure could be changed to plot inlet $H_2O$ versus ambient $H_2O$ with the points colored by altitude.

Figure 14. How can the lack of dependence on water vapor be explained, given that it is a purported to be related to one of the amplification chemistry reactions ($HO_2$+NO)? Perhaps modeling of these data would be instructive. Also, suggest showing data down to the lowest water values possible.

Figure 15. Can the laser emission be adjusted so all three detectors peak at the same wavelength? This would definitely help them to behave more similarly. Suggest changing the right-hand y-axis to go from 0 to10, and to average the cross-section data to a lower resolution, say 0.1 nm, to make the plot message clearer.

Figure 16. Not sure this figure is necessary, since the data are plotted in Figure 17.

Figure 19. change caption "..while changing $O_3$…", "…in the source with estimated 15%...". It is not clear why the perturbations to $\Delta NO_2$ last so long (up to 40 seconds) when the background should be measured on the 1 second time scale. Is this a data processing issue? Suggest checking why this is the case.

Figure 20. There are big swings in the inlet pressure even when the altitude is not being changed. Why is this?

Figures 21 and 22. These figures could be eliminated.

---

## Referee Comment (RC2) · Anonymous Referee #2 · 22 Nov 2019

**Review of "Airborne measurement of peroxy radicals using chemical amplification couple with cavity ring down spectroscopy: the PeRCEAS instrument" by Midhun George et al.**

This paper describes a configuration of a peroxy radical chemical amplification (PeRCA) inlet system and a cavity ring down spectrometer (CRDS) detector in a flight ready package for measurement of the sum of peroxy radicals ($HO_2$ + $RO_2$) in the troposphere. The authors describe a series of instrument development experiments and modelling exercises used for optimization of instrument parameters. Example data from a single flight is shown as an example of flight performance.

This paper is a worthwhile addition to the literature as an update to a previously developed instrument from this group. Considerable improvements have been made in the detection of NO2 from Horstjann et al 2014 ($LOD_{NO2}$ = 300 pptv) in the current configuration ($LOD_{NO2}$ = 60 pptv), averaging time (120 s to 60 s) and altitude pressure limitations of the inlet. The reviewer believes this paper should be published but recommends a more focused approach on the novel improvements to the instrument rather than discussing well established and previously published methods (ie CRDS calibration, PeRCA calibration, etc).

**General comments:**

Overall the reviewer believes 23 figures is too many for an instrument development paper of this nature. An instrument schematic could replace the first three figures (photos of the instrument, inlet and inside of aircraft) similar to Horstjann et al 2014 figure 1. The authors referred to an improved inlet design (DUALER vs DUALER II) by modifying the pre-chamber design and reducing wall interaction in the inlet. This modification seems significant and likely affects the instrument performance more than discussed in this paper. A figure comparing the two inlet designs or the changes in inlet design would be useful.

The general description of how the inlet operates (alternating measurement modes) is somewhat confusing and is evident in the Reviewer #1's comments. A time series figure of the operation of each channel would make this clearer (ie switching from amplification mode to background mode and showing how each channels mode switching is out of phase with each other). Furthermore, a detailed description of how the mixing ratio of NO was decided on (30 ppmv) would be useful, as it differs significantly from the DUALER I (6ppmv) inlet and other groups PeRCA inlets (0.9 to 7.7 ppmv).

Generally speaking the flight data section of the paper should be focused on the improved performance of the instrument rather than flight tracks and mixing ratio figures. A comparison of DUALER I and DUALER II flight data is recommended. Considering how to show improvements between DUALER II deployments is recommended.

**Specific comments:**

Page 4, line 120, "The optical cavity remains similar to that described in Horstjann et al…" It is useful to include mirror specifications (substrate, coating, reflectivity, diameter, etc) for a CRDS instrument, as they are critical part of theoretical instrument performance. Does the piezo optical alignment system run in a closed loop control with beam profile as a feedback parameter? If so describe this, as it seems novel.

Page 5, line 146, "mode and modulation times…" it is not clear what mode and modulation times refer to, might be useful to define them discretely

Page 5, line 160, "…detector temperature. For this different detector temperature gradients, $\Delta T$, where applied to modulated signals generated by varying the sampled $NO_2$ concentration…" it's not clear why the investigators modulated $NO_2$ while applying a temperature gradient to the detector. Would it not be easier to interpret if a constant mixing ratio gas was sampled while applying a temperature gradient? It is not clear from the text where this temperature gradient is and how it was applied. It would be useful to readers that are not familiar with optics, on why a temperature gradient of 7 degC would cause detector instability. It is also not clear from the text, what was done to address this flaw in the detector design, as the authors state earlier detector stability is paramount in overall instrument performance.

Page 6, line187, "…of the sampled O3 by NO to form NO2 also depends on the concentration of NO added to the sample flow and the time for reaction before reaching the detector.", This would be a good place to discuss how 30 ppmv NO was decided on for a reagent mixing ratio and discuss flow rate choices for both NO and CO.

Page 6, line 192, "3.2.1 Effective Chain Length…", This section seems to describe a well-established method documented in literature. The reviewer recommends shorting the description of the method and explain better the difference in DUALER I and DUALER II eCL.

Page 7, line 216, "The model was initialized with 9% CO, 3 ppb $O_3$, 50 pptv $HO_2$…" why was 3 ppb $O_3$ determined to be a representative mixing ratio for ozone? I may be misunderstanding the inlet chemistry, but it seems like missing 30 ppmv of NO would significantly affect the modeled CL. Assuming the box model initialization is correct, would it not be useful to vary the wall loss rate constants to match the eCL and determine if this wall loss is reasonable? It would also be useful to experimentally determine the wall loss of the inlet.

Page 7, line 222, "figure 8 shows eCL vs CL", the authors should include error bars on these data.

Page 7, line 228, "Figure 9 depicts the $O_3$ decay simulated for 100 to 200 ppb…" these figures are somewhat confusing to the reviewer. One could take the 99% conversion time for each NO mixing ratio curve and plot all 4 conditions (ie pressure and $O_3$ mixing ratio) on 1 figure for varying NO mixing ratio. Additionally, adding an inlet residence time reference line would be useful for helping the reader visualize what time limit you have on this reaction.

Page 7, line 234, "PAN and PPN thermal decomposition", the reviewer believes that experimental work is justified to confirm 'this source of radicals is considered to be negligible'. The box modelling done for CL prediction was shown to not capture the actual inlet system, so it's not clear why it would do a better job with modelling PAN and PPN. Figure 11 shows up to 10 pptv interference, this does not seem negligible to the reviewer.

Page 8, line 252, "Figure 12 shows the variation of the eCL for 45 ppm NO within a pressure range…", The reviewer does not understand why this experiment was done with 45 ppm NO when the decided upon mixing ratio of NO addition seems to be 30 ppm NO for the rest of the paper. If this is a typo, it should be corrected, if not the experiment should be done at the actual mixing ratio the instrument is operated at.

Page 9, line 284, "Figure 15 depicts exemplary a comparison of spectra…", the reviewer does not believe including $NO_2$ absorption cross section and detector spectra is a useful figure for the main text of this paper.  Remove or include in the SI.

Page 9, line 289, "In addition, the effective $\sigma_{NO2}$ can be calculated by sampling known mixtures…", the reviewer does not believe including a time series of calibration gas addition to instrument is a useful figure for the main text of this paper.  Remove or include in the SI.

Page 9, line 294, "The result of apply Eq. 4 to the PeRCEAS detectors at 200 mbar is depicted in Figure 17."  It is more common to plot $NO_2$ number density [molecules/cm$^3$] vs. α, as the slope has the physical meaning of the absorption cross section of $NO_2$.

Page 9, line 302, "The main source of uncertainty…" the authors previously mention detector drift due to temperature changes (figure 5), is this not a significant source of uncertainty as well?

Page 10, line 309, "Figure 18 shows the calculated eCL from 14 radical calibrations…." why were radical calibrations done with a NO mixing ratio of 45 ppm when the instrument is run at 30 ppm?

Page 10, line 330, "As can be seen in Figure 19…", it would be useful to plot actual $O_3$ mixing ratio rather than set point in the top panel of this figure.  It would also be useful to plot chamber pressure or channel pressure, as it seems like when the $O_3$ mixing ratio is changed the $NO_2$ signal displays a large amount of noise (50 ppb).  It is also unclear what D1, D2, SG and BG stand for in this figure.

Page 11, line 364, "The pressure regulation in PeRCA based airborne instruments results in lower eCL than ground based ones." This statement should be substantiated.  The reviewer sees a range of eCL for aircraft instruments from 45 to 322 and ground based instruments a range of 91 to 1010.  A description of why lower pressure or pressure regulation in general affects eCL would be useful.

Page 11, line 367, "the detection limit and uncertainty of PeRCA based instruments are strongly depending on the variation of $O_3$ and $NO_2$ in the sampled air mass…" The reviewer believes this statement should be substantiated with uncertainty and/or detection limit analysis to show the reader the magnitude of this affect.  If the changes in $O_3$ and $NO_2$ are measured (as often are in aircraft field campaigns) can a correction method not be proposed and evaluated?

Page 12, line 379, "As can be seen in figure 20…" this reviewer believes it would be useful to add the detector temperature (or deltaT used earlier) to this figure, as this was determined to be a large effect on $\Delta NO_2$ earlier in the manuscript.  If $NO_2$ and $O_3$ mixing ratio data is available from the flight, this would be useful to include as well.

Page 12, line 383, "Figure 22…" the reviewer does not believe this or figure 21 add considerable value to this paper.  Recommend removing or moving to SI.

Page 12, line 387, "…illustrate the improvement in the dynamical stability achieved in successive airborne deployments…" the reviewer finds it difficult to see the improvements made in the measurement by looking at time series data.  Suggest thinking of a different way of presenting this conclusion.  It would also be very useful to add a comparison to the previous generation of instrument somewhere in the paper.

---

## Author Comment (AC1) · 16 Mar 2020

General comments and suggestions

(1) Comments from Referees: The researchers appear to accept the basic characteristics of the inlet/reactor as given. The paper does not justify the selection of flow rates, inlet/reactor volume and composition, and thus the corresponding reaction time. This reviewer sees this approach as flawed. Perhaps this has been well thought out, and previously published, but not included in this paper. Suggest that the authors include some discussion of why these parameters were selected, and what the compromises and advantages in their selection are.

[Figure]

(2)Response to RC1: The objective of the present publication is to explain the dominant factors affecting the overall performance and accuracy of PeRCEAS for the determination of the hydroperoxyl, HO2, and organic peroxy radicals, RO2, which react with NO to form NO2, when deployed on the the HALO aircraft. The operating conditions of PeRCEAS are optimised for the specific sampling position used, cabin location of the instruments, safety requirements and the type of flight tracks and altitude profiles which were flown by HALO. Generally, these limitations are often different in different campaigns. For this reason, this manuscript does not aim at describing a unique universal set of PeRCEAS operating conditions. In practice the mechanical constraints and the safety requirements of the HALO (e.g. the size and weight of the pylon for the inlet, the amount of CO permitted on board) determine the volume and shape of the reactors and partly the range of flows of the gases and the residence time within the PeRCEAS. More detailed information about the inlets DUALER I and DUALER II and their differences are now provided in section 2 (see answers to specific comments).

(1) Comments from Referees: One aspect is that higher NO levels used in the studies reported in this paper (30-45 ppmv) result in lower sensitivity to CH3O2 (and other RO2) compared to previous values because of a faster rate of CH3O+NO+M. High NO also converts a greater fraction of ambient ozone to NO2, although complete conversion is not necessary. Most other chemical amplifiers used NO reagent mixing ratios of 2 to 6 ppmv.

(2) Response to RC1: The selection of the concentration rather than the mixing ratio of NO is indeed a critical issue for the PERCA approach. High concentrations of NO are required to guaranty the full titration of O3 in NO2 to capture fast variations of O3 within a measurement cycle. The total conversion of O3 to NO2 in the system enables the quantification of radical data at a 60s temporal resolution, as explained now in figure 4. In this way, the horizontal resolution of the PeRCEAS airborne measurements, which depends on the speed and altitude of HALO, is typically between 7 and 15 km. Longer modulation cycles than 120 s result in noisy and unrepresentative averages

for ambient measurements in air masses having significant short term variability of O3 and NO2. Provided the partial conversion is stable and identical for both reactor-detector lines of the PeRCEAS, we agree with RC1 that the partial conversion of O3 to NO2 is sufficient to determine RO2*. Malfunctioning of one of the detectors yields the RO2* to be determined at 120 s. In this case, if O3 is completely converted and the simultaneous O3 and NO2 measurements on board are of sufficient accuracy, a RO2* time resolution of 60 s may still be feasible.

The majority of PERCA measurements in the literature using NO concentration in the range 3-6 ppm were made at 1 atmosphere i.e. concentrations of (7.3-14.6) x 1013 molecule cm-3. At 300 mbar these correspond to the mixing ratio range 10-20 ppm. We expect that at concentrations of NO of 14.6 x 1013 molecule cm-3, i.e. a mixing ratio of 20 ppm, the ratio of eCL(CH3O2)/eCL(HO2) is 65 % and 40 % for 45 ppm NO.

As now explained in the section 3.3. and in the response to the corresponding specific question below, the water vapour dependence of eCL decreases significantly from 10 ppm to 45 ppm NO ([NO] 7.29 x 1013 to 3.28 x 1014 molecules cm−3 at 300 mbar). The results in figure 14 for 45 ppm NO indicate that variations in the sample humidity do not lead to additional uncertainty in the RO2* retrieval as the PeRCEAS eCL remains invariable within the experimental error up to [H2O] $\sim$ 1.4 x 1017 molec cm-3.

The final selection of [NO] will be a balance between having stable eCL with respect to water vapour concentrations and having smaller eCL for RO2 measurements. As now shown in Table 2 the eCL of RO2 for 300mbar and NO 45 ppm, assuming to be CH3O2 the dominant atmospheric RO2, would be 40% of the eCL for HO2. The sum of HO2 + 40% RO2 can then be compared with atmospheric model values to test our understanding of the production of HO2 an RO2 in air masses sampled in flight.

(1) Comments from Referees: It is not apparent why the response time of the system is so slow. For example, the Figure 5 caption states that 20 seconds is eliminated after change in NO2. Given a reactor transit time of 3 seconds, this seems extreme. There

may be delays, which can be accounted for in data analysis, that are different than transition to the correct value after perturbations.

(2) Response to RC1: The PeRCEAS operating conditions have now been explained more in detail in the text and supplementary information (see answers to specific comments). During calibration measurements, different $NO_2$ mixing ratios are generated by the dilution of a mixture of $NO_2$ in synthetic air from a commercially certified gas cylinder (Airliquid 10 ppmv $NO_2$ in $N_2$). As illustrated in the Figure RC1_I below, the flow controller used for adding different $NO_2$ mixing ratios to the detector requires approximately 10s second to reach the set value after a change. If this time is added to the time required for the probe to reach the detector (see Table 3; 5.27 s), at least 16 seconds of data should be ignored after a change in the $NO_2$ mixing ratio.

(1) Comments from Referees: Part of the answer may be in Figure 19, where step functions in the $O_3$ concentration result in perturbations lasting about 40 seconds.

(2)Response to RC1: The $O_3$ concentration and mixing ratios were changed by adjusting the speed of the flow passing through a Hg lamp, which photolyses $O_2$. When the $O_3$ concentration is changed, it takes a time for the ozone generator to stabilise the $O_3$ concentration in its flow and a time lag required for the probe to reach the detector. This is the reason for the changes in figure 19 (now figure 17).

(1) Comments from Referees: Also, in Figure 20, the pressure variations last for a long time during and after altitude changes. There are also pressure fluctuations even when the aircraft is not changing altitude. This implies need for better pressure control. Perhaps the PID parameters of the pressure controller have not been adjusted properly. This is very important to get correct. Though improved, such fluctuations are still apparent in Figure 23. They add unnecessary noise to the measurements. Suggest adding more discussion of the pressure control system (manufacturer, model, adjustment procedures) and the response time of the system to step changes in $NO_2$ concentrations to allow the reader to better understand these issues.

(2)Response to RC1: To avoid misunderstanding the figure 20 (new figure 18) has been improved. In this figure the dynamic pressure changes are depicted which are the cause of fluctuation in the inlet. The changes in the dynamic pressure arise from altitude changes and from changes of the aircraft velocity (i.e., including turning) and air turbulence. Under laboratory condition the time constant for the pressure system to stabilise after a perturbation in PeRCEAS is 15s. The error induced for the RO2* measurements is shown in figure 20 (new figure 18) to be small or negligible as a result of turbulence or HALO velocity changes. After identifying the measurements influenced by changes in the dynamic pressure, only RO2* which have pressure fluctuations of less than 2 mbar in 60 s, the modulation time, are primarily used for analysis. The relevant paragraph now reads as follows (Lines 481- 487)

(3) Author's changes in manuscript: "As can be seen in the figure 18, the dynamic pressure variations experienced by the aircraft influence the stability of the inlet pressure. These changes are attributed to altitude changes, air turbulence, and changes in aircraft velocity, including turning, of the aircraft. The effect of inlet pressure instabilities on the retrieved $\Delta NO2$ is not exactly identical for both detector signals. This leads to additional uncertainty in the RO2* determination when using the procedure discussed in section 4.3. For the data analysis, pressure spikes within 1 minute standard deviation higher than 2 mbar are identified and flagged. This approach enables data with large error due to dynamic pressure changes to be identified. Overall the error in the retrieved RO2* is around 20 % in the measurement period shown in figure 18."

(1) Comments from Referees: Finally, why are the chain lengths reported in this paper so much lower that previous publications (Table 3)? In fact, values reported in this paper (28-38, Figure 18) are much lower than what is reported under "This Work" in Table 3 and would be the lowest values for the CO/NO chemical amplifier in the table (there are lower values for the ethane/NO chemical amplifier). This compromises the potential quality of the measurements. Perhaps this bears on the question about optimization of the instrument earlier. It seems that chain lengths of 100 or more are

possible (at lower reagent NO mixing ratios). Some explanation in the paper is needed to explain this.

(2)Response to RC1: The text has been extensively rewritten to address this issue. We hope that we have removed any misunderstandings with respect to the values reported in the text and in the old table 3 (new table 4).

Specific comments and suggestions

(1) Comments from Referees: Page 1, line 16. "...for the airborne measurements in the..."

(2)Response to RC1: It has been corrected as suggested by the referee.

(1) Comments from Referees: Page 1, line 18. "...instrumental channels successfully captures short term...".

(2)Response to RC1: It has been corrected as suggested by the referee.

(1) Comments from Referees: Page 1, line 20. Not sure why the word "gradients" is used here. How about "...range of atmospheric pressures and temperatures expected..."?

(2)Response to RC1: It has been changed as suggested by the referee.

(1) Comments from Referees: Page 1, line 24. The phrase "...collectively known at RO2*...". Is it true that HO2+RO2 is "known" as RO2*? Is this the term accepted by the community? This reviewer suggests just using "HO2+RO2" instead.

(2)Response to RC1: RO2* is the term defined as RO2*= HO2+2+OH + and used specifically for the PeRCA measurements. The text has been accordingly modified.

(1) Comments from Referees: Page 1, line 29. Suggest removing the summation symbol, since there is a plus sign used.

(2)Response to RC1: It has been corrected as suggested by the referee.

(1) Comments from Referees: Page 1, line 32-33. Suggest ". . .photolyzed to ultimately produce. . ."

(2)Response to RC1: It has been changed as suggested by the referee.

(1) Comments from Referees: Page 2, line 34. Suggest "Overall, HO2+RO2 influences the. . ."

(2)Response to RC1: It has been changed as suggested by the referee.

(1) Comments from Referees: Page 2, line 41. Not sure what is meant by "those RO2".

(2)Response to RC1: The text has been extended for clarification. Lines 41-50:

(3) Author's changes in manuscript: "The chemical amplification technique (Cantrell and Stedman, 1982; Hastie et al., 1991) has been used to measure the sum of peroxy radicals. The Peroxy Radical Chemical Amplification (PeRCA) converts by addition of NO and CO, HO2 and most atmospherically significant RO2 to NO2. The OH formed in the reaction cell reacts with CO to reform HO2 in a chain reaction. Oxy, alkoxy, hydroxy and alkylperoxy radicals (OH + + HO2 + 2) are converted into NO2. As the RO and OH abundances in the troposphere are much lower than those of HO2 and RO2, PeRCA measures to a good approximation the sum of peroxy radicals collectively known as RO2*, (RO2* = HO2 + $\Sigma$ RO2 ; being R any organic chain), which convert NO to NO2. The rate coefficients of the HO2 and RO2 reactions with NO are very similar (Lightfoot et al, 1993). Large RO2 which do not react with NO to form NO2 are not detected, and are assumed to be negligibly small compared to the sum of HO2 + 2 concentrations. HO2 and CH3O2 are the dominant peroxy radicals present in an air mass in most conditions."

(1) Comments from Referees: Page 2, line 43. Suggest ". . .compared to the total amount of HO2+RO2, with HO2. . ."

(2)Response to RC1: This has been changed as suggested by the referee.

(1) Comments from Referees: Page 2, line 53. Should be "Kanaya".

(2)Response to RC1: This has been corrected as suggested by the referee.

(1) Comments from Referees: Page 2, line 54. Suggest a different word than "largely". Suggest "The interference by some RO2…"

(2)Response to RC1: This has been changed as suggested by the referee.

(1) Comments from Referees: Page 3, line 72. Add space "Peroxy Radical…".

(2)Response to RC1: This has been corrected as suggested by the referee.

(1) Comments from Referees: Page 3, line 74. Suggest "…in a previous publication…".

(2)Response to RC1: This has been changed as suggested by the referee.

(1) Comments from Referees: Page 3, line 79. Suggest "…where $\Delta NO2$ is the NO2 formed…"

(2)Response to RC1: This has been changed as suggested by the referee.

(1) Comments from Referees: Page 3, line 84. Suggest "…spectroscopic measurement technique…"

(2)Response to RC1: This has been changed as suggested by the referee.

(1) Comments from Referees: Page 3, line 87. NO2 comes from radical amplification and from the background (O3 conversion). Suggest changing the sentence to reflect this.

(2)Response to RC1: The sentence has been reworded. Line 102:

(3) Author's changes in manuscript: "In PeRCEAS the absorber of interest is NO2 which is formed in both the amplification and the background modes. "

(1) Comments from Referees: Page 3, line 92-94. Suggest also saying that c0 is the

speed of light in a vacuum (stated later in the paper).

(2)Response to RC1: The text has been extended as suggested by the referee.

(1) Comments from Referees: Page 3, line 92. Suggest ". . .are, respectively, the absorption. . ."

(2)Response to RC1: This has been changed as suggested by the referee.

(1) Comments from Referees: Page 3, line 95. Suggest ". . .used for ground-based measurements. . ."

(2)Response to RC1: This has been corrected as suggested by the referee.

(1) Comments from Referees: Page 3, line 96-7. Suggest ". . .the particular constraints related to airborne measurement. . .".

(2)Response to RC1: This has been changed as suggested by the referee.

(1) Comments from Referees: Page 3, line 99. Suggest "In this study, the specifications. . .are described based on thorough laboratory. . ."

(2)Response to RC1: The text has been changed as suggested by the referee.

(1) Comments from Referees: Page 4, line 107. Suggest (if correct) ". . .and located outside the HALO fuselage. . ."

(2)Response to RC1: The pylon is a part of the fuselage. The sentence has been reworded. Line 123:

(3) Author's changes in manuscript: ". . ..installed inside a pylon located on the outside of the HALO fuselage"

(1) Comments from Referees: Figures 1, 2, and 3. While photographs can be nice, schematic diagrams are more useful to see the path of sample, reagents, and signals throughout the system. Suggest limiting to only one or two photographs and add diagrams to show the details.

(2)Response to RC1: The photos have been replaced by schematic diagrams as suggested by the referee.

(1) Comments from Referees: Page 4, line 117. Mixing and pressure regulation are mentioned here, but no detail is given. This is relevant to the general comment given earlier. Suggest adding more detail in the text and perhaps in Figures 1 and 2. Suggest discussing how DUALER I and II are different and describe why the changes do indeed result in improved performance.

(2)Response to RC1: The text has been extended with the description of the DUALER operation and Figure 2 has been included to highlight differences between DUALER I and II. Lines 127 to 151:

(3) Author's changes in manuscript: "Briefly, sampled air enters PeRCEAS through the DUALER pre-chamber, which is at a lower pressure than that outside of the HALO, through an orifice in a truncated cone, i.e. a nozzle. From this pre-chamber the air is pumped simultaneously through the two flow reactors and a bypass line. At the upper addition point a mixture of CO or N2 and NO enters each reactor. At the lower addition point, a flow of N2 or CO enters each reactor. This enables the CO and N2 flows in the two reactors within the DUALER to be switched simultaneously but out of phase with one another from the upper to the lower addition point. At the addition points, the reagent gases enter the reactor through eight circular distributed 1 mm holes to facilitate the rapid mixing with the sampled air. During measurements, the pressure in the pre-chamber and both reactors is held constant. However, there is a small pressure fluctuation during the switching of flows between the upper and lower mixing point. The flow passing through each reactor enters a CRDS NO2 detector. Afterwards, the sample flows together with the air from the bypass line are scrubbed for CO and NO and exhausted by the pump.

The DUALER inlet comprises two PeRCA chemical reactors having alternating measurement modes, which are out of phase with one another. During the first part of the

measurement cycle, the first reactor and detector are in amplification mode, while simultaneously the second reactor and detector are in background mode. In the second part of the cycle, the CO addition point in both reactors is switched. Consequently, the first reactor and detector are then in background mode while the second reactor and detector are in amplification mode. In the analysis of the measurements, the amplification and background signals from both detectors are combined appropriately. This improves accuracy and temporal resolution of the resultant RO2* data set (see 3.1).

In the DUALER, a stable pressure in the pre-chamber is achieved by a pressure regulator, which controls the flow through the bypass line. As noted the flow rate through the reactors is held constant during measurements. Consequently, when the outside air pressure changes, the bypass flow rate from the pre-chamber is changed. The outer dimensions, shape, form and weight of the DUALER are constrained by the inlet pylon in use with the research aircraft HALO. After the first version of the DUALER (from now on called DUALER I) was flown, the inner dimensions of the pre-chamber were further optimised to reduce the wall losses and turbulence in the pre-chamber. For this, in the DUALER II the volume of the pre-chamber was increased by extending its vertical extent, the length of the truncated cone on top of the reactors was reduced in 3 mm, and the volume of the reactors was increased to 130.5 ml from the 112 ml in DUALER I. These changes resulted in a higher eCL and improved pressure stability in DUALER II as compared to DUALER I. Figure 2 shows the upper part of both DUALER I and DUALER II."

(1) Comments from Referees: Page 4, line 124. The term "piezo electric stack" is not a common term and needs more description.

(2)Response to RC1: A piezo electric stack features a longitudinal deformation when voltage is applied. A mirror mounted on piezo electric stack was used to achieve mode matching between the single mode laser and resonator in Hortsjann et al., 2014. This is not used in the current configuration of PeRCEAS.

The text has been accordingly modified (now lines 158-159).

(3) Author's changes in manuscript: "With this, the fine adjustment of the laser is simplified and improved, and the piezo electric stack used to achieve mode matching between the single mode laser and the optical cavity in Hortsjann et al., (2014) becomes unnecessary and is removed."

(1) Comments from Referees: Page 4, line 128. Suggest adding the manufacturer of the beam camera (MKS Ophir). This is very cool, by the way!

(2)Response to RC1: The text has been extended. Line 162:

(3) Author's changes in manuscript: "During alignment procedures and for test purposes, a beam camera (BM-USB-SP907-OSI, Ophir Spiricon Europe GmbH) monitors the beam profile and simplifies the identification of misalignments or loss of performance of the optical system."

(1) Comments from Referees: Page 4, line 132-3. Suggest "...other sensor data such as pressure, flow, temperature, and humidity." You don't need "etc." if using "like" or "such as".

(2)Response to RC1: This has been corrected as suggested by the referee.

(1) Comments from Referees: Page 5, line 149. Would "occasionally" be better than "exceptionally"?

(2)Response to RC1: This has been changed as suggested by the referee.

(1) Comments from Referees: Page 5, line 150-152. The term that should be used is "Allan Variance" rather than "Allan Deviation". Perhaps the equation for its calculation should be given, since there is also modified Allan Variance that can be used. Also, give one or two references (e.g. Allan, 1966; Allan et al., 1991; Allan and Levine, 2016). Suggest "...was investigated using calculated Allan Variance in the measurement...". Also "...the optimum integration time for the three PeRCEAS detectors is between 20

s. . .". Do you use 20-30 second averaging in the data analysis? The timing of the instrument cycling should be shown, perhaps in Figure 1 or in a separate figure.

(2)Response to RC1: The analysis of the Allan deviation was used to infer the detection limit of the measurement resulting from random noise. The plot of the Allan deviation has been replaced by another plot of the Allan variance as suggested by the referee. The text has also been extended for clarification. Lines 195-206:

(3) Author's changes in manuscript: " To optimize the mode time and thus also the modulation cycle, the Allan variance (Allan, 1966; Werle et. al., 1993) was analysed for PeRCEAS. Given a time series of N elements and a total measurement time tacq, tacq = facq·N, where facq is the frequency of acquisition, then the Allan variance is defined as:

$\sigma$_x^2($\tau$) = 1/2 âŇľãĂŰ (x_(i+1)-x_i )ãĂŮ^2 âŇł_$\tau$ (Eq.2)

where xi is the mean over a time interval of a length $\tau$, being $\tau$ = facq·m; and m the number of elements in a selected interval. The use of âŇľ...âŇł denotes the arithmetic mean. The square root of the Allan variance is the Allan deviation. For random noise, the Allan deviation at any given integration time determines the detection limit of the measurement.

The Allan variance plot for measurements of 5.6 ppbv NO2 at 200 mbar and 23 °C is shown in figure 5. As can be seen, the optimal averaging time for the three PeRCEAS detectors is in the range between 20 s and 50 s. The corresponding minimum (2$\sigma$) detectable mixing ratio is < 60 pptv (3.15 x 108 molecules cm−3 for these P and T conditions). Slow temperature drifts over longer averaging times impact on both the laser and the resonator characteristics. This behaviour is observed for averaging times longer than 60 s."

(1) Comments from Referees: Page 5, line 155. The phrase ". . .over the modulation time. . ." need explanation. This might be apparent with addition of a figure showing the

instrument cycle timing.

(2)Response to RC1: The text has been extended for clarification at the beginning of 3.1. (Lines 181-194), and Figure 4 has been added:

(3) Author's changes in manuscript: "The mode time is defined as the time selected for the measurement in either amplification or background mode. The modulation time is the time taken for a complete measurement cycle, which comprises the sum of one amplification and one background mode. The PeRCEAS measurement cycle is illustrated in Figure 4. The $\Delta NO_2$ for each detector is calculated from the ring down time of two consecutive modes using Eq.(1). If the mode time is adequately selected, the RO2* retrieved per measurement cycle is identical in both measurement lines, as the two reactors are operated out of phase with one another. The final RO2* data is calculated as the mean of the RO2* determined from the $\Delta NO_2$ and eCL of both detectors for a given measurement cycle. The time resolution of the RO2* measurement is then equal to the mode time. After switching modes, a small pressure pulse leads to an oscillation of the NO2 signal. Consequently, the first 20 s of each mode are not used in data analysis. The time lag arising from the time taken for the sample flow between the CRDS detector and the point of switching is typically less than 8 s (see Table 3).

Typically, 650 to 800 ring down times of the NO2 absorption are averaged per second and the measurement of NO2 is made at 1 Hz. Individual ring down times are occasionally saved for sensitivity studies. Modulation and mode times are selected empirically. The optimised values are a compromise between the time taken for the detector signal to stabilise after the CO/N2 flow is switched between the addition points, and the temporal variability of the chemical composition of the air probed"

(1) Comments from Referees: Page 5, line 161. Suggest "...signals generated that minutely varied the sampled NO2..."

(2)Response to RC1: The paragraph has been rewritten for clarification. Lines 211-217:
[Figure]

(3) Author's changes in manuscript: "Temperature changes of the CRDS affect: i) the diode laser emission, both its amplitude and wavelength; ii) the mode matching between laser and detector, and consequently the $\tau 0$. The effect of the variations in $\tau$, resulting from changes in room or HALO cabin air temperatures, on the accuracy and precision of the $\Delta NO2$ determination was investigated by a series of laboratory experiments. For this, modulated concentrations of NO2 in the flow were generated. This was achieved by alternating between two selected NO2 concentrations once per minute. The temperature of the CRDS detector, T, and $\tau$ were then measured. Detector temperature gradients over a time t, i.e., $\Delta T/\Delta t$, determined by the temperature within the CRDS housing close to the photodiode detector, were induced by controlled changes in the room temperature."

(1) Comments from Referees: Page 5, line 164. "…time resolution using each background…". This will be more obvious with a graphical representation.

(2)Response to RC1: Now the figure 4 depicts the PeRCEAS measurement cycle.

(1) Comments from Referees: Page 5, line 167. "… molecules cm-3 for typical measurement conditions".

(2)Response to RC1: This value refers to the value in molecules cm-3 corresponding to 150 pptv at the particular T and P of the measurement. The text has been changed for clarification. Line 220:

(3) Author's changes in manuscript: "….the experimental precision of the $\Delta NO2$ determination remains within ($2\sigma$) 150 pptv (= 7.3 x 108 molecules cm$-3$ at 200 mbar and 23°C). "

(1) Comments from Referees: Page 5, line 170. "It is inferred from laboratory calibrations that a 60 s modulation cycle is an optimum comprise between…". The term "modulation cycle" is not apparent here, but perhaps would be with a graphical representation of the instrument cycle. Also, add more discussion why fluctuations last so

long (20 s).

(2)Response to RC1: Figure 4 now depicts the measurement cycle. The text has also been extended as suggested by RC1 to clarify the modulation cycle (see lines 181-190:

(3) Author's changes in manuscript: "The mode time is defined as the time selected for the measurement in either amplification or background mode. The modulation time is the time taken for a complete measurement cycle, which comprises the sum of one amplification and one background mode. The PeRCEAS measurement cycle is illustrated in Figure 4. The $\Delta NO2$ for each detector is calculated from the ring down time of two consecutive modes using Eq.(1). If the mode time is adequately selected, the RO2* retrieved per measurement cycle is identical in both measurement lines, as the two reactors are operated out of phase with one another. The final RO2* data is calculated as the mean of the RO2* determined from the $\Delta NO2$ and eCL of both detectors for a given measurement cycle. The time resolution of the RO2* measurement is then equal to the mode time. After switching modes, a small pressure pulse leads to an oscillation of the NO2 signal. Consequently, the first 20 s of each mode are not used in data analysis. The time lag arising from the time taken for the sample flow between the CRDS detector and the point of switching is typically less than 8 s (see Table 3)."

(1) Comments from Referees: Page 5, line 172. The detection for NO2 should be performed with a background level of ozone in the sample, since this is how ambient measurements are performed. Was this is the case?

(2)Response to RC1: As described in the text, the detection limit for NO2 was determined by measuring a modulated signal generated by dilution of NO2 from commercial standard cylinders in synthetic air to get 11.5 and 12.1 ppbv NO2 as background and amplification signals respectively. This is equivalent to adding a background level of O3, since O3 is converted totally in NO2 before reaching the detector. In further complementary measurements it has not been observed any significant variation in the NO2 detection limit for variations up to 100 ppb in the O3 background produced by a

ozone generator.

(1) Comments from Referees: Page 5, line 173-4. It is not obvious that larger modulation times lower the representativeness of the averages. Do you mean variability in peroxy radicals or in the background? For the latter, the instrument is continually measuring the background, and it should be well accounted for. If you mean the peroxy radicals, while one-minute (or quicker) data are nice, longer averages can still be useful in adding understanding of tropospheric free radical behavior. Suggest rewording the last sentence of this paragraph.

(2)Response to RC1: In instruments based on PERCA modulated signals, the modulation time determines the resolution. In contrast to ground based measurements, as the airborne platform additionally moves horizontally and vertically, modulation times longer than 120 s can be critical to mirror the peroxy radical and background variabilities in the encountered air masses. This part of the text has been rewritten to address the criticism of the referee and moved to 3.2.2 for clarification (Lines 310-316):

(3) Author's changes in manuscript: "As explained in section 3.1. the simultaneous use of two detectors measuring out of phase results in the temporal resolution of the RO2* data being 60s. In this way, the horizontal resolution of the PeRCEAS airborne measurements, which depends on the speed and altitude of HALO, is typically between 7 and 15 km. Longer modulation cycles than 120 s result in noisy and unrepresentative averages for ambient measurements in air masses having significant short term variability of O3 and NO2. To keep the temporal resolution of the RO2* data to be equal to the mode time, the rapid and complete conversion of ambient O3 into NO2 within the PeRCEAS is required. For this, the NO concentration added at the inlet has to be sufficient for a complete titration of the sampled O3 before reaching the detector."

(1) Comments from Referees: Page 6, line 176. "Sample and reagent gas flows...".

(2)Response to RC1: It has been changed as suggested by the referee.

(1) Comments from Referees: Page 6, line 176-184. This discussion of reaction time should also include discussion of how the size of the reactor was selected, since it also affects the reaction time (reactor volume / total flow). Perhaps this would be a good place to discuss the approach to ensuring mixing of reagent gases with the ambient air sample. How was this done? Were fluid dynamical calculations performed? Were flow visualization approaches used? Related to this: how do you ensure that no components of the inlet system are leaking?

(2)Response to RC1: The length of the reactor is limited by the design of the pylon. Therefore the reaction time is constrained by the mechanical borders and the flow conditions selected. The description in section 2 has been extended for clarification.

The reagent gases are added to the reactor through eight 1mm holes distributed circularly around the reactor tube in order to maximise the mixing with the ambient air sample.

The tightness of the inlet is controlled regularly by applying an overpressure of 0.8 mbar after closing all openings of the inlet and using leak soap at the connections. CO leakages in the aircraft are a critical safety issue in the aircraft. Therefore the inlet tightness is checked before installation and the potential critical points of the instrument are tested by adding He overpressure and using a He detector.

(1) Comments from Referees: Page 6, line 182. "...lower explosion limit (LEL) in air of 12.5% v/v at room temperature...".

(2)Response to RC1: The text has been changed as suggested by the referee.

(1) Comments from Referees: Page 6, line 189. Remove extra space between "air" and "and", and between "CO" and "in".

(2)Response to RC1: This has been corrected as suggested by the referee.

(1) Comments from Referees: Page 6, line 190. Suggest "...between safety requirements, limiting...".

(2)Response to RC1: This has been corrected as suggested by the referee.

(1) Comments from Referees: Page 6-7. Effective chain length. This might be good place to discuss experiments to determine the optimum NO concentration for the amplifier chemistry. Also, perhaps near the end of this section, discuss how the effective chain length values are used in the data analysis. In other words, have estimates of the $HO_2/RO_2$ ratio been made and used to apply the two eCL (for $HO_2$ and $HO_2+CH_3O_2$) values? If so, how is it done?

(2)Response to RC1: The text of section 3.2.1: Effective chain length has been rewritten to address this criticism. Additional experiments were undertaken to determine the eCL as a function of [NO]. These are reported in figure 7 and also in table 2. The concentration or mixing ratios of $HO_2$ and $RO_2$ are not known in ambient air. Thus the ratio is also not known. The $RO_2^*$ values are reported to be the sum of $HO_2 + \alpha \cdot RO_2$. Using $CH_3O_2$ as a surrogate for all $RO_2$, the values of $\alpha$, which depends on [NO], have been determined by modeling and measurement in Table 2. The relevant paragraph in section 3.1.1 now reads as follows (Lines 302-306):

(3) Author's changes in manuscript: "Table 2 summarises the simulated PeRCEAS sensitivity for the $HO_2$ and $CH_3O_2$ detection for different NO mixing ratios in the reactor at 300 mbar. Up to 10 ppm NO ([NO] 7,29 x 10$^{13}$ molecules cm$^{-3}$) the difference in sensitivity remains within the PeRCEAS uncertainty. The ratio of the $eCL_{CH_3O_2}/eCL_{HO_2}$ is defined as $\alpha$. The estimated values of $\alpha$ from modelling and measurements are given in table 2. For the assessment of air masses the measurements of $HO_2 + \alpha \cdot RO_2$, where $\alpha RO_2 \approx \alpha \cdot CH_3O_2$, are compared with atmospheric model values of $HO_2 + \alpha \cdot RO_2$."

(1) Comments from Referees: Page 6, line 197. "...to the conversion into $RO_2$.". Include that the approach is based on $O_2$ actinometry, as opposed to other approaches reported in the literature (such as $N_2O$ actinometry, calibrated NIST photodiodes).

(2)Response to RC1: The text has been extended for clarification. Lines 247-258:

(3) Author's changes in manuscript: "The eCL of the DUALER reactors is determined in the laboratory by using a calibrated source of peroxy radicals. The latter uses the photolysis of water vapour at 184.9 nm (see Schultz et al., 1995). Briefly, a known water vapour - air mixture is photolysed by a low pressure mercury (Hg) lamp. A nitrous oxide (N2O) absorption filter attenuates the intensity of 184.9 nm radiation. This is achieved by varying the N2O/N2 ratio in the filter absorption zone. The photolysis of H2O makes an OH and H. In air, the H reacts with O2 in a termolecular reaction to make HO2. The photolysis of oxygen molecules yield oxygen atoms, O which react with O2 in a termolecular reaction to make O3 (see Reichert et al., 2003). CO is added to the gas mixture in the source to convert the OH into HO2 radicals. As a result, each absorbed photon by a water vapour molecule generates two HO2 molecules. Alternatively, the addition of a hydrocarbon, RH, leads to the conversion of OH to a RO2, and consequently to a 1:1 mixture of HO2 and RO2 for calibration. The concentration of HO2 or RO2, and O3 is thus proportional to the intensity of 184.9 nm electromagnetic radiation. As the absorption coefficient of N2O (Cantrell et al., 1997) does not change significantly around 185 nm ($\sigma$N2O=14.05×10-20 cm2 molecule-1 at 25 ◦C with a 0.02×10-20 cm2 molecule-1 K-1, temperature dependency), different HO2 and RO2 radical amounts can be produced for a constant H2O concentration.."

(1) Comments from Referees: Page 6, line 205. "...are changed stepwise every ten minutes from 8 pptv...". Also, note that too much reagent added to the calibrator has the potential to affect the inlet chemistry. This can be seen by a change in the background with change in radical concentration, which is not expected if the background is mostly due to ozone.

(2)Response to RC1: The radical calibration procedure does not require any change in reagents. Different radical mixing ratios are generated by attenuation of the light of the Hg/Ne UV lamp used for photolysis of H2O and O2 by using different concentrations of N2O as absorption filter, as described in previous publications (e.g, Reichert et al., 2003). In each step of the calibration both the amplification and the background signal

change due to the effect of the light attenuation in the photolysis of H2O and O2 leading to radicals and O3 respectively.

The text has been extended for clarification:

(3) Author's changes in manuscript: In line 255 "….The concentration of HO2 or RO2, and O3 is thus proportional to the intensity of 184.9 nm electromagnetic radiation, and the absorption coefficient of N2O (Cantrell et al., 1997) does not change significantly around 185 nm ($\sigma$N2O=14.05×10-20 cm2 molecule-1 at 25 âŮ̧C with a 0.02×10-20 cm2 molecule-1 K-1, temperature dependency), different HO2 and RO2 radical amounts can be produced for a constant H2O concentration."

(3) Author's changes in manuscript: In line 271: "The O3 generated by the radical source is converted in the DUALER to NO2 by its reaction with NO, which is in excess. Therefore the O3 entering the reactor during the radical calibration is detected as NO2 in the background and amplified signals.. "

(1) Comments from Referees: Page 6, line 206-7. "…is determined from the slope of the measured $\Delta$NO2 levels versus the calculated radical amounts. Example data is shown in…". Suggest rewording the end of the last sentence on this page, since the concentration of NO ***within the inlet*** is 30 ppmv. Perhaps "…and added reagent NO to achieve 30 ppmv within the inlet."

(2)Response to RC1: We agree with the comment of the referee but the original figure has been removed.

(1) Comments from Referees: Page 7, line 208. Suggest "In Figure 7, the PeRCEAS eCL versus the inlet NO concentration…". In Figure 7, why aren't data shown for lower NO concentrations, such as used by your group in the past and by other researchers?

(2)Response to RC1: This figure has been extended for eCL values at lower NO concentrations as suggested.

(1) Comments from Referees: Page 7, line 212. "…concentration, eCL values increase

with. . .".

(2)Response to RC1: The text has been modified as suggested.

(1) Comments from Referees: Page 7, line 16-18. Suggest describing how wall losses were determined. Are they constant or are they affected by the cleanliness of the inlet? Suggest putting all the rate coefficients used for both reactor pressures into Table 1. It is interesting that a level of 3 ppbv O3 was used, presumably because this is what comes out of the calibrator. Suggest also running the model with ambient-like levels of O3.

(2)Response to RC1: The wall losses are not determined experimentally. The result of eCL calibrations before and after the measurement campaigns indicate that the effect of cleanliness of the inlet in the eCL is within the experimental error.

The text has been extended (Lines 289-297) and new simulations have been performed for clarification. Table 1 includes now all rate coefficients taken from JPL- Publication 15-10 (Burkholder et al, 2015). The model is initialised with 3ppb O3 because this is the mixing ratio produced during the calibration. Sensitivity studies have shown no significant change in the simulated eCL up to 100ppb O3. This is also confirmed by the experimental values shown in figure 17.

(1) Comments from Referees: Page 7, line 221. Inlet pre-chamber is not defined anywhere. This should be shown in the schematic diagram discussed earlier. If the radical losses in the model do not agree with what you think they arein the DUALER II inlet, suggest you perform experiments to determine what they are. The model of this simple chemistry should be much closer to the observations that a factor of two!

(2)Response to RC1: The inlet pre-chamber is now described in detail in section 2. As mentioned in the previous response, the text has been extended for clarification (Lines 280-306) and new simulations have been performed to better reproduce the pre-chamber + reactor configuration in PeRCEAS.

(1) Comments from Referees: Page 7, line 222. Suggest "...shows measured eCL versus modeled CL for the...".

(2)Response to RC1: This figure has been replaced.

(1) Comments from Referees: Page 7, line 223-4. "The CLmodeled/eCLmeasured ratio averages about 2 for HO2...". Actually, the ratio is more than 2 for the 200 mB measurements and is about 2 for the 300 mB HO2 measurements. Only the 300 mB HO2+CH3O2 measurements are about 1.5. Does this mean that the inlet wall loss changes with reactor pressure? Were the rate coefficients in the model changed to reflect the reactor pressure? Perhaps the model wall loss values should be adjusted based on new laboratory measurements. The chemical amplifier chemistry is simple enough that a box model should be able to accurately reproduce laboratory data such as this. Add error bars to the points to represent total uncertainty in the measured and modeled values. Perhaps perform regressions of data.

(2)Response to RC1: As already mentioned, the text has been modified (lines 280-301) and new simulations of the pre-chamber losses have been included. The inlet wall losses are calculated using Eq.4. $k\_w=1.85((v^{(1/3)} D^{(2/3)})/(d^{(1/3)} L^{(1/3)}))(S/V)$ (Eq.4) S is the surface area in cm2, V the volume in cm3, L the length and d the diameter of the flow tube in cm, v the velocity of the gas in cm s-1, and D is the diffusion coefficient, which is calculated to be DHO2=0.21 and DCH3O2=0.14 in cm2 s-1. At different pressures the velocity of the gas in the reactor changes and therefore the kw.

The modeling data obtained for 300mbar is now shown in Figure 8. These agree reasonably with the corresponding experimental results.

(1) Comments from Referees: Page 7, line 226. This reviewer does not like the term "titration" in this context, even though it is widely used in the community. Suggest using "conversion" instead.

(2)Response to RC1: In the case of PeRCEAS the term titration is used correctly when the conditions are selected to achieve the complete conversion of O3 into NO2 before reaching the detector. The title of 3.2.2 has been replaced by "conversion of ambient O3 into NO2" and the text in lines 310-316 has been extended for clarification:

(3) Author's changes in manuscript: "As explained in section 3.1. the simultaneous use of two detectors measuring out of phase results in the temporal resolution of the RO2* data being 60s. In this way, the horizontal resolution of the PeRCEAS airborne measurements, which depends on the speed and altitude of HALO, is typically between 7 and 15 km. Longer modulation cycles than 120 s result in noisy and unrepresentative averages for ambient measurements in air masses having significant short term variability of O3 and NO2. To keep the temporal resolution of the RO2* data to be equal to the mode time, the rapid and complete conversion of ambient O3 into NO2 within the PeRCEAS is required. For this, the NO concentration added at the inlet has to be sufficient for a complete titration of the sampled O3 before reaching the detector. "

(1) Comments from Referees: Page 7, line 227. This reviewer disagrees that ozone in the sample has to be completely converted toNO2. Why is this? It seems that partial conversion, as long as it is stable, would be fine.

(2)Response to RC1: We agree with RC1 that the partial conversion of O3 in the sample, as long as stable and identical for both reactor-detector lines of the PeRCEAS is sufficient to determine RO2*. However the total conversion of O3 in the system enables the quantification of radical data at a 60s temporal resolution as explained in figure 4. Malfunctioning of one of the reactor-detector line yields the RO2* to be determined at 120 s. In this case, if O3 is completely converted and the simultaneous O3 and NO2 measurements on board are of sufficient accuracy, a RO2* time resolution of 60s may still be feasible.

(1) Comments from Referees: Page 7, line 228. Figure 9 has a lot of information that could be presented in a more straightforward way. Suggest plotting the ozone lifetime

(or three lifetimes) versus the reactor NO at the two pressures. This could be shown in one plot.

(2)Reply to RC1: The figure 9 has been changed as proposed by the referee.

(1) Comments from Referees: Page 7, line 229. There is no reason to require conversion of 100-200 ppbv of ozone to 1-2 pptv in the inlet. Conversions of 99% are more than sufficient. Suggest changing this paragraph as Figure 9 is changed.

(2)Reply to RC1: The text has been changed for clarification (Lines 310-322).

(1) Comments from Referees: Page 7, line 230. The wavelength 409 nm is mentioned, but everywhere else it indicates that the lasers operate at 408 nm.

(2)Reply to RC1: This is a typo and has been corrected.

(1) Comments from Referees: Page 7, line 236. Suggest "...which are captured by...". This reviewer disagrees that the radicals and NO2 from PAN-like compounds cancels and does not lead to interference. Yes, the NO2 from the decomposition should be like ambient NO2 and be corrected for by the background measurement. But the radicals formed from the decomposition will amplify and appear like ambient radicals. This is an interference! Suggest rewording this paragraph. Some direct laboratory measurements of the interference would also be helpful. Figure 11 shows that the PAN interference is greater at lower reactor pressures. Why would this be the case. At reduced pressure, the decomposition is slower and the time is the reactor is shorter. Suggest checking the modeling.

(2)Response to the RC1: The time of reaction of CH3CO2 produced by the decomposition of PAN with NO is the same in both amplification and background modes. The decomposition of PAN is therefore a potential interference, if PAN decomposes between the upper and lower gas addition points in the reactor working in amplification mode. Taking into account the residence times given in Table 3 and the reactor temperatures showed now in figures 18 and 19, for most operating conditions the potential interference will remain below 2 pptv and can be considered negligible (see also answer to RC2).

The chemistry involved in the formation of CH3O2 from the PAN decomposition has now been revised, and all the rates have been taken from the recommendations in JPL 15-10, with the equilibrium rate constant from Zhang et al., (2011).

There is no significant difference in the production of CH3O2 radicals with the pressure. The differences observed in the graph are the result of the conversion of molecules·cm-3 in mixing ratios.

The figure has been updated and the text has been modified for clarification. Lines 324-339:

(3) Author's changes in manuscript: "Peroxyacyl nitrates (RC(O)OONO2) such as peroxyacetylnitrate, PAN and peroxypropionyl nitrate can decompose thermally inside PeRCEAS. The extent of the decomposition to peroxy radicals and NO2 depends on the time and the temperature. If the thermal decomposition occurs at shorter time scales than the modulation time, they can be a significant interfering source of radicals which are chemically amplified and lead to additional NO2. In a rapidly changing background the RO2* determination might be affected according to the temperatures and sample residence times between the gas addition points in the DUALER (Table 3).

To evaluate this effect the production of peroxy radicals from the thermal decomposition of 1 ppb PAN at different temperatures and pressures has been simulated. The results obtained with a box model (Ianini, 2003) including the reactions:

CH3COO2NO2 → CH3COO2 + NO2 (R1)

CH3COO2 + NO → CH3 + CO2 + NO2 (R2)

CH3 + O2 + M → CH3O2 (R3)

are depicted in figure 11. The rate coefficients used are taken from Burkholder et al.,

(2015).

The [CH3O2] produced does not vary significantly at the pressures investigated. As the temperature of the PeRCEAS reactors during flight generally remain under 290 K, this source of radicals is considered to be negligible for most operating conditions. The thermal stability of the PAN analogues is similar to that of PAN but they are usually at much lower concentrations than PAN in the atmosphere and also assumed to be a negligible source of error."

(1) Comments from Referees: Page 8, line 259-260. It is not clear what is meant by "based on the similarity of the eCL values". Suggest rewording this sentence, and perhaps this entire paragraph to make the message clearer.

(2)Response to RC1: It was concluded that the eCL dependency observed was related to the relative humidity and not to the absolute [H2O] because the eCL measured by Reichert et al., (2003) at 20°C and 30°C did not differ within the experimental errors although corresponding to significantly different absolute water concentrations. The text has been modified for clarification. Lines 354-358:

(3) Author's changes in manuscript: "The effect of changes in the sampled air humidity on the eCL has been reported and studied by Mihele and Hastie, (1998) and Mihele et al., (1999). Reichert et al. (2003) investigated the dependency of the eCL for ground based measurements at 20 °C and 30 °C and standard pressure, i.e., keeping the relative humidity but almost doubling the absolute water concentration. The obtained eCL values did not differ within the experimental error and confirmed the dependency of eCL on the relative humidity. All these measurements were performed at a pressure of one atmosphere and for 3.3 ppmv NO ([NO] 8.12 x 1013 molecules cm−3)."

(1) Comments from Referees: Page 8, line 263. Suggest replacing "as shown exemplary" with "with an example shown".

(2)Response to RC1: The text has been modified.

(1) Comments from Referees: Page 8, line 265. "...sample humidity do not lead to...".

(2)Response to RC1: The text has been modified.

(1) Comments from Referees: Page 8, line 267. "...is subject to two types of errors which either are: a) intrinsically..."

(2)Response to RC1: The text has been modified as suggested.

(1) Comments from Referees: Page 8, line 268. "...in the laboratory, or b) result..."

(2)Response to RC1: The text has been modified as suggested.

(1) Comments from Referees: Page 8, line 273. Equation 3 is very similar to Equation 1. Suggest eliminating Equation 3 and referring back to Equation 1 in this discussion. Perhaps change Equation 1 slightly, if needed.

(2)Response to RC1: The text has been modified as suggested and Equation 3 has been eliminated.

(1) Comments from Referees: Page 9, line 279. "...Vandaele et al. [2002] with the normalized laser spectrum from the corresponding detector.". Also, "The values obtained have been..."

(2)Response to RC1: The text has been modified as suggested.

(1) Comments from Referees: Page 9, line 284. "...depicts a sample comparison of spectra...".

(2)Response to RC1: The figure has been moved to the supplementary information as suggested by RC2 and the text has been modified. Lines 387-388 :

(3) Author's changes in manuscript: "A sample comparison of spectra obtained for the three PeRCEAS detectors is included in the supplementary information (Figure SI-1)."

(1) Comments from Referees: Page 9, line 294. "The effective $\sigma$NO2 obtained agrees within...".

(2)Response to RC1: The sentence has been corrected.

(1) Comments from Referees: Page 9, line 297. The tau symbol disappeared. You have a lot of ambient data. Does $\tau 0$ vary significantly as the CRDS cell mirrors are exposed to ambient air?

(2)Response to RC1: The $\tau 0$ symbol disappeared during the processing of the document. This is a typo and has been corrected. The variation in the value of $\tau 0$ during the measurement depends on the composition of the air probed. As shown in figure 1, the sample air goes through a $5\mu$m filter before reaching the PeRCEAS detectors. This filter removes most of the ambient particles. As a consequence there is only a gradual change in $\tau 0$ over the 10 hours flight which is not critical for the measurement. The filter is replaced after each flight.

(1) Comments from Referees: Page 9, line 300. ". . .measurement requires accurate. . ."

(2)Response to RC1: The text has been modified as suggested.

(1) Comments from Referees: Page 9, line 302-3. ". . .are the radical calibration. . .". ". . .which is estimated to be. . .".

(2)Response to RC1: The source of uncertainty is actually the generation of radicals during the radical calibration.

(1) Comments from Referees: Page 10, line 307. "The errors associated. . .".

(2)Response to RC1: The text has been modified as suggested.

(1) Comments from Referees: Page 10, line 312. ". . .reactor 2, respectively, and. . .".

(2)Response to RC1: The text has been modified as suggested.

(1) Comments from Referees: Page 10, line 315-6. Delete "during the airborne measurement of RO2*".

(2)Response to RC1: The text has been modified as suggested.

[Figure]

(2)Page 10, line 317. "The noise in the NO2 signal is enhanced by…".

Response to RC1: The text has been modified as suggested.

(1) Comments from Referees: Page 10, line 319. "…cabin temperature could increase…".

(2)Response to RC1: The cabin temperature varies depending on the characteristics of the flight. The sentence has been reworded.

(1) Comments from Referees: Page 10, line 320. "…stability of the CRDS signal and the accuracy of the supporting measurements."

(2)Response to RC1: The text refers to measurements taken before flying to check the overall performance of the instrument. Therefore they are rather "reference" than "supporting" measurements.

(1) Comments from Referees: Page 10, line 323-4. Again, this reviewer does not agree with this statement. Since you are continually measuring the background and the amplified signal plus background, variations should be accounted for. Only changes happening faster than one second should have influence, unless there is something about the data analysis that is not obvious from the presentation in the paper. Suggest looking into why step changes in ozone should affect the signal for more than a few seconds. Also, step changes are extreme for ambient measurements. Even changes as the aircraft changes altitude are likely to be gradual unless a pollution layer is encountered.

(2)Response to RC1: The PeRCEAS operating conditions has been clarified in text and in answers to previous questions of RC1. These clarifications have already addressed this issue.

(1) Comments from Referees: Page 10, line 335. "…a standard deviation of the order of…".

(2)Response to RC1: The sentence has been corrected.

(1) Comments from Referees: Page 11, line 353. "...detector signals can be significantly affected...".

(2)Response to RC1: The text has been changed as suggested.

(1) Comments from Referees: Page 11, line 360. "...airborne measurements and is difficult to implement in...".

(2)Response to RC1: The text has been changed as suggested.

(1) Comments from Referees: Page 11, lines 364-370. Do you mean running the inlet at reduced pressure results in lower eCL values? Doesn't PeRCEAS continually measure the signal and background as mentioned in the second line? What do differences in detector accuracy (do you mean sensitivity) do to affect their uncertainties? Suggest adding a reference to the last sentence of this paragraph (about RO2 interferences in LIF).

(2)Response to RC1: As now better described in section 2 the pressure in the inlet is controlled in the pre-chamber during the flight. This pre-chamber is however associated with radical wall losses which reduce the eCL as discussed in 3.2.1. The losses in the pre-chamber depend on the residence time in the pre-chamber. The latter depends on the volume of the pre-chamber and the total flow rates.

The sentence has been extended for clarification. Lines 469-470:

(3) Author's changes in manuscript: "The pressure regulation in PeRCA based airborne instruments results in lower eCL than ground based ones. This is attributed to radical losses in the pre-chamber prior to the addition of reagent gases for the radical chemical amplification."

(2)Response to RC1:The reference Fuchs et al. (2011) is already cited in the introduction. However, it has additionally been included here as suggested by the referee.

(1) Comments from Referees: Page 12, line 374. "Figure 20 shows sample data of

RO2* measured. . .".

(2)Response to RC1: The text has been changed as suggested. The figure 20 is now Figure 18.

(1) Comments from Referees:Page 12, line 382. Mention that the flagged values are shown in Figures 20 and 23. Also mention this in the figure captions.

(2)Response to RC1: The flags are now mentioned in the text and in the corresponding figure captions (now figures 18 and 19) as suggested by the referee.

(1) Comments from Referees: Page 12, line 386. ". . .in more detail in Figure 23. . .".

(2)Response to RC1: The text has been modified (Lines 483-487). The corresponding figure is now Figure 19.

(1) Comments from Referees: Page 12, line 388. Not sure what is meant by "the signal is not affected by altitude changes", since there are jumps in $\Delta NO2$ when the altitude changes. Suggest rewording to make the point clearer.

(2)Response to RC1: Although $\Delta NO2$ changes by altitude changes the value of RO2* does not change significantly. The text has been accordingly rewritten. Lines 489-492:

(3) Author's changes in manuscript: "As can be seen in the figure, pressure fluctuations due to dynamic pressure changes have been reduced by up to 80 % in the improved PeRCEAS. Although the measured $\Delta NO2$ is affected by altitude changes, the value of the retrieved RO2* does not change significantly except for the maximum climbing rate directly after take-off.".

(1) Comments from Referees: Page 12, line 402. ". . .over a 60 s integration. . .".

(2)Response to RC1: The sentence has been corrected.

(1) Comments from Referees: Page 12, line 404. While PeRCEAS may be suitable for measurements up to 12 km, no data were shown at this altitude. Suggest rewording

this sentence.

(2)Response to RC1: The sentence has been reworded. Line 506:

(3) Author's changes in manuscript: "The performance of the PeRCEAS instrument has been proven to be suitable for airborne measurements during different campaigns onboard HALO."

(1) Comments from Referees: Page 12, line 405. "...campaigns onboard HALO.".

(2) Response to RC1: The sentence has been changed as suggested.

(1) Comments from Referees: References Line 449. "...peroxy radicals by chemical amplification...". Line 452. Two references are together. Need carriage return after "1993".

(2)Response to RC1: The references have been corrected.

Tables

(1) Comments from Referees: Suggest heading for "second addition point" to be changed to "reaction times", and "to detector" changed to "transfer times".

(2)Response to RC1: The headings have been changed to "reactor residence time" and "total residence time".

(1) Comments from Referees: Page 3, Edwards et al., Inlet pressure should be 200 mB.

(2)Response to RC1: This has been corrected.

Figures

(1) Comments from Referees: Most of the figures need larger symbols and bolder lines (4, 5, 6, 10, 12, 14, 17, 18, 19, 20, 23). In many of the plots, the legend is covered by data. Suggest enclosing legend in a box with a white background.

(2)Response to RC1: This has been corrected.

(1) Comments from Referees:See comments earlier about Figures 1-3.

(2)Response to RC1: These figures have been removed and replaced as suggested.

(1) Comments from Referees: Figure 2 caption. "…Top view of the…". "..the laser beam is highlighted (purple) for…". "…exiting the cavity is depicted."

(2)Response to RC1: This figure has been removed and replaced as suggested.

(1) Comments from Referees: Figure 4. Change y-axis to Allan Variance. Describe what the lines depict (linear fits to data less than 10 seconds?).

(2)Response to RC1: This figure has been removed and replaced as suggested. A subplot has been added to show the detector stability for 40 minutes measurement by detector FH as an example. These data are used for the Allan variance study of the corresponding detector.

(1) Comments from Referees: Figure 5. It is not obvious what this figure is trying to show. It appears to this reviewer that the point is temperature changes affect the $\tau 0$ of the detector, but the retrieved $\Delta NO2$ is affected very little. Why not do this experiment with two detectors as done for radical measurements? This would be a more realistic representation of the actual measurement situation.

(2)Response to RC1: The objective of this experiment is to illustrate the effect of temperature drifts in the $\Delta NO2$ at the detector. Since both NO2 detectors are identical in the operating conditions, it is sufficient to show the results obtained with one detector. The text has been extended for clarification

(1) Comments from Referees: Figure 6. Are the equations determined from linear fits? Are they standard or bivariate fits?

(2)Response to RC1: The equations are determined from linear and bivariate fits, considering the errors in x and y axis. This plot is now placed in the supplementary information. (1) Comments from Referees: Figure 7. Why are not data shown for lower values of NO? Suggest more work going from 0 to 3.5E14 NO with at least 10 points for each instrumental condition (pressure and radical type). Perhaps also show the same y-axis for both plots.

(2)Response to RC1: The data series have been extended as suggested.

(1) Comments from Referees: Figure 8. Since it is mentioned in the text, suggest adding 1:1.5 line. As discussed earlier, perhaps more modeling with more realistic wall loss rates needs to be done.

(2)Response to RC1: The text has been extended and the results of additional modeling have been depicted in a new figure.

(1) Comments from Referees: Figure 9. Changes to this figure suggested earlier (plot O3 lifetime versus NO). If it is kept the same, suggest labeling each sub-figure and referring to those labels in the caption.

(2)Response to RC1: The figure has been changed as suggested.

(1) Comments from Referees: Figure 10. This shows that 60% of the O3 is converted with NO of 3 ppmv, and 90% at 6 ppmv. This means that the instrument could be run with much lower NO levels.

(2)Response to RC1: This issue has been extensively discussed in previous answers.

(1) Comments from Referees: Figure 11. It is stated that PAN interference is not a problem with PeRCEAS, but this plot shows that with reaction times of 3 seconds (compared to 2.6 to 3.1 seconds for the two DUALER inlets), up to 6 pptv of peroxy radicals can be produced from 1 ppbv of PAN. Is this representative of the conditions for which the instrument has been used? This figure could be changed to plot the fraction conversion of PAN (CH3O2 produced / PAN) versus temperature for the two DUALER reaction times.

(2)Response to RC1: As mentioned before, the simulations have been revised and the plots updated. The ratio CH3O2 produced/PAN is <10-3 for the range of temperatures expected in the reactors, which are in the outer part of the fuselage. The maximum interference expected remains below 1-2 pptv for the transition times given in Table 2.

(1) Comments from Referees: Figure 13. There are places where the ambient water is below the inlet water. How can this be? This figure could be changed to plot inlet H2O versus ambient H2O with the points colored by altitude.

(2)Response to RC1: The figure has been changed as suggested by the referee. The relative humidity sensor at the inlet has been re-calibrated. After applying the calibration correction all the above mentioned values of inlet humidity lower than the ambient humidity have disappeared. The plot has been updated.

(1) Comments from Referees: Figure 14. How can the lack of dependence on water vapor be explained, given that it is a purported to be related to one of the amplification chemistry reactions (HO2+NO)? Perhaps modeling of these data would be instructive. Also, suggest showing data down to the lowest water values possible.

(2)Response to RC1: The dependency on water vapour only becomes significant at higher water number concentrations for NO 45 ppm at 300 mbar as shown in figure 14. New measurements at 10 and 30 ppmv NO at 300 mbar show a clear dependency with [H2O] which has to be taken into account in the analysis of ambient air measurements at these operating conditions. The water vapour dependence of eCL decreases significantly from 10 ppm to 45 ppm. For water vapour concentrations in the range of 4 -14 x 1016 molecule cm-3 of relevance for HALO flights, this results in the water vapour impact on the eCL being negligible. This is shown more explicitly in the new figure 14 in the manuscript.

As described in Hastie et al., (1991) and Reichert et al., (2003), the chain length (CL) of a PERCA reactor can be expressed using a resistance model as follows:

[Figure]

1/CL= (1/CL_(HO_2 ) + 1/CL_OH )

1/CL = 1/PNO2 ((2 removal rates/2 propagating rates) + (removal rates/propagating rates))

And if only the predominant processes are considered, as:

1/CL = 1/PNO2 ((kwHO2 /k1) + ((kwOH + k2 [NO])/k3 [CO]))

P_(NO_2 ) is the probability of radical conversion into NO2, k_(w_(HO_2 ) )and k_(w_OH ) are the wall losses of HO2 and OH respectively, k1 is the rate constant for NO2 production from HO2 + NO reaction, k2 is the rate constant for HONO formation from OH + NO reaction, k3 is the rate constant for HO2 formation from OH + CO reaction. In the presence of H2O, HO2 water clusters are formed as postulated by Reichert et al., (2003).

Our interpretation of these novel results is that as the NO is increased from mixing ratios of 10 to 45 ppm at 300 mbar, the CLOH is reduced, because the rate of the chain termination termolecular reaction of OH with NO making HONO increases, and the CLHO2is increased, because the rate of the propagation reaction between NO and HO2 increases. As a result the CL begins to be dominated by the CLOH, which is independent of water vapour.

The new series of measurements are now included in Figure 14 and the text has been extended for clarification (Lines 359-371):

(3) Author's changes in manuscript: "In this work radical mixtures were sampled at 25 °C for relative humidity between 2 % and 25 %. This leads to a ca. 20 times increase in the absolute [H2O]. These conditions cover the [H2O] expected for a larger T range (-20 − 30 °C) during airborne measurements in the free troposphere at 200 and 300 mbar inlet pressures. Figure 13 shows the [H2O] in the air probed versus the [H2O] in the inlet for real measurements on board of the HALO aircraft. The results in figure 14 for 45 ppm NO ([NO] 3.28 x 1014 molecules cm−3 at 300 mbar) indicate

that variations in the sample humidity do not lead to additional uncertainty in the RO2*
retrieval as the PeRCEAS eCL remains invariable within the experimental error up to
[H2O] $\sim$ 1.4 x 1017 molecules cm-3. In contrast, for 10 ppm and 30 ppm NO in the
reactor ([NO] 7.29 x 1013 and 2.19 x 1014 molecules cm$-$3 at 300 mbar) the eCL
shows a clear dependency with the ambient [HO2]. The comparison with the eCL
values obtained by Reichert et al (2003) at 1 atmosphere indicate a eCL dependency
on [H2O], temperature and pressure having a different pattern for 45 ppm NO in the
reactor. This is explained by invoking the competition in the amplification chain length,
CL, between HO2 and OH removal rates, as explained in Hastie et al., (1991) and
Reichert et al., (2003). At [NO] $\sim$ 3.28 x 1014 molecules cm$-$3 the CL begins to be
dominated by the rate of the termination termolecular reaction of OH with NO, which is
independent of water vapour. This eCL dependency has to be taken considered in the
analysis of ambient air RO2* measurements."

The [H2O] values showed are the lowest water values experimentally possible with the
used set up. (1) Comments from Referees: Figure 15. Can the laser emission be
adjusted so all three detectors peak at the same wavelength? This would definitely
help them to behave more similarly. Suggest changing the right-hand y-axis to go from
0 to10, and to average the cross-section data to a lower resolution, say 0.1 nm, to
make the plot message clearer.

(2)Response to RC1: The laser emissions cannot be changed. They are multimode
diode lasers with fixed emission at a particular measurement condition. The plot has
been modified as suggested and moved to the SI as suggested by RC2.

(1) Comments from Referees:Figure 16. Not sure this figure is necessary, since the
data are plotted in Figure 17.

(2)Response to RC1: This figure has been moved to the SI.

(1) Comments from Referees:Figure 19. change caption "..while changing O3...",
"...in the source with estimated 15%...". It is not clear why the perturbations to ∆NO2

last so long (up to 40 seconds) when the background should be measured on the 1 second time scale. Is this a data processing issue? Suggest checking why this is the case.

(2)Response to RC1: The $\Delta NO2$ from each detector is calculated from individual ring down times. Consequently, variations in the background affect the calculated $\Delta NO2$. This effect is cancelled out when averaging the $\Delta NO2$ of both detectors to retrieve RO2* which consequently remains unaffected. The variation of O3 from the calibrator is not instantaneous but takes around 1 minute to become stable. This is the reason why ca. three RO2* values are affected by each change in O3.

(1) Comments from Referees:Figure 20. There are big swings in the inlet pressure even when the altitude is not being changed. Why is this?

(2)Response to RC1: The figure has been modified for clarification. The pressure spikes are related to dynamical pressure changes in the inlet, e.g. caused by changes in velocity of the aircraft and air turbulences.

(1) Comments from Referees:Figures 21 and 22. These figures could be eliminated

(2)Response to RC1: These figures have been removed as suggested by both referees.

---

## Author Comment (AC2) · 17 Mar 2020

The comment was uploaded in the form of a supplement:
https://www.atmos-meas-tech-discuss.net/amt-2019-359/amt-2019-359-AC2-supplement.pdf

[Figure]

[Figure]

**Fig. 1.** Figure RC1_I: PeRCEAS response to a step change in sampled NO2 mixing ratio

---

## Author Comment (AC3) · 17 Mar 2020

General comments:

(1) Comments from Referees: Overall the reviewer believes 23 figures is too many for an instrument development paper of this nature. An instrument schematic could replace the first three figures (photos of the instrument, inlet and inside of aircraft) similar to Horstjann et al 2014 figure 1.

(2)Response to RC2: The first three figures have been replaced by instrument schematics as suggested by the referee.

[Figure]

(1) Comments from Referees: The authors referred to an improved inlet design (DUALER vs DUALER II) by modifying the pre-chamber design and reducing wall interaction in the inlet. This modification seems significant and likely affects the instrument performance more than discussed in this paper. A figure comparing the two inlet designs or the changes in inlet design would be useful.

(2)Response to RC2: A new figure (Figure 2) showing the changes to the pre-chamber and reactors in the DUALER has been included as suggested by the referee. In addition the text has been rewritten (Lines 143-151):

(3) Author's changes in manuscript: "In the DUALER, a stable pressure in the pre-chamber is achieved by a pressure regulator, which controls the flow through the by-pass line. As noted the flow rate through the reactors is held constant during measurements. Consequently, when the outside air pressure changes, the bypass flow rate from the pre-chamber is changed. The outer dimensions, shape, form and weight of the DUALER are constrained by the inlet pylon in use with the research aircraft HALO. After the first version of the DUALER (from now on called DUALER I) was flown, the inner dimensions of the pre-chamber were further optimised to reduce the wall losses and turbulence in the pre-chamber. For this, in the DUALER II the volume of the pre-chamber was increased by extending its vertical extent, the length of the truncated cone on top of the reactors was reduced in 3 mm, and the volume of the reactors was increased to 130.5 ml from the 112 ml in DUALER I. These changes resulted in a higher eCL and improved pressure stability in DUALER II as compared to DUALER I. Figure 2 shows the upper part of both DUALER I and DUALER II."

(1) Comments from Referees: The general description of how the inlet operates (alternating measurement modes) is somewhat confusing and is evident in the Reviewer #1's comments. A time series figure of the operation of each channel would make this clearer (ie switching from amplification mode to background mode and showing how each channels mode switching is out of phase with each other).

(2)Response to RC2: A new figure (Figure 4) has been added showing the ring down times of the two detector, the retrieved $\Delta NO2$ and HO2 for a laboratory calibration of HO2. The text has been rewritten (Lines 127-142):

(3) Author's changes in manuscript: "Briefly, sampled air enters PeRCEAS through the DUALER pre-chamber, which is at a lower pressure than that outside of the HALO, through an orifice in a truncated cone, i.e. a nozzle. From this pre-chamber the air is pumped simultaneously through the two flow reactors and a bypass line. At the upper addition point a mixture of CO or N2 and NO enters each reactor. At the lower addition point, a flow of N2 or CO enters each reactor. This enables the CO and N2 flows in the two reactors within the DUALER to be switched simultaneously but out of phase with one another from the upper to the lower addition point. At the addition points, the reagent gases enter the reactor through eight circular distributed 1 mm holes to facilitate the rapid mixing with the sampled air. During measurements, the pressure in the pre-chamber and both reactors is held constant. However, there is a small pressure fluctuation during the switching of flows between the upper and lower mixing point. The flow passing through each reactor enters a CRDS NO2 detector. Afterwards, the sample flows together with the air from the bypass line are scrubbed for CO and NOx and, exhausted by the pump.

The DUALER inlet comprises two PeRCA chemical reactors having alternating measurement modes, which are out of phase with one another. During the first part of the measurement cycle, the first reactor and detector are in amplification mode, while simultaneously the second reactor and detector are in background mode. In the second part of the cycle, the CO addition point in both reactors is switched. Consequently, the first reactor and detector are then in background mode while the second reactor and detector are in amplification mode. In the analysis of the measurements, the amplification and background signals from both detectors are combined appropriately. This improves accuracy and temporal resolution of the resultant RO2* data set (see 3.1). "

(1) Comments from Referees: Furthermore, a detailed description of how the mixing
ratio of NO was decided on (30 ppmv) would be useful, as it differs significantly from the DUALER I (6ppmv) inlet and other groups PeRCA inlets (0.9 to 7.7 ppmv).

(2)Response to RC2: The objective of the present publication is to explain the dominant factors affecting the overall performance and accuracy of PeRCEAS for the determination of the hydroperoxyl, HO2, and organic peroxy radicals, RO2, which react with NO to form NO2, when deployed on the the HALO aircraft. The operating conditions of PeRCEAS are optimised for the specific sampling position used, cabin location of the instruments, safety requirements and the type of flight tracks and altitude profiles which were flown by HALO. Generally, these limitations are often different in different campaigns. For this reason, this manuscript does not aim at describing a unique universal set of PeRCEAS operating conditions.

(1) Comments from Referees: Generally speaking the flight data section of the paper should be focused on the improved performance of the instrument rather than flight tracks and mixing ratio figures. A comparison of DUALER I and DUALER II flight data is recommended. Considering how to show improvements between DUALER II deployments is recommended

(2)Response to RC2: The performance of DUALER I and DUALER II are now compared in section 5. In addition some key improvements of performance of DUALER II compared with DUALER I are highlighted. Lines 488-493:

(3) Author's changes in manuscript: "Two hours of measurements from the flight on the 19.03.2018 are shown in Figure 19 as an example of the third airborne deployment of PeRCEAS within the EMeRGe campaign in Asia. As can be seen in the figure, pressure fluctuations due to dynamic pressure changes have been reduced by up to 80 % in the improved PeRCEAS. Although the measured $\Delta$NO2 is affected by altitude changes, the value of the retrieved RO2* does not change significantly except for the maximum climbing rate directly after take-off. Furthermore, the beam camera and the motorised mirror mounts enable the identification and immediate correction of

small misalignments. This improves significantly the instrumental performance while simplifying maintenance."

  Specific comments:

(1) Comments from Referees: Page 4, line 120, "The optical cavity remains similar to that described in Horstjann et al. . ." It is useful to include mirror specifications (substrate, coating, reflectivity, diameter, etc) for a CRDS instrument, as they are critical part of theoretical instrument performance. Does the piezo optical alignment system run in a closed loop control with beam profile as a feedback parameter? If so describe this, as it seems novel.

(2)Response to RC2: The piezo optical alignment system was used in the previous PeRCEAS configuration using a single mode laser as reported by Horstjann et al., (2014). The actual PeRCEAS detector described here, has a multimode diode laser. The alignment is done manually using motorized mirror mounts while the beam camera confirms the TEM00 mode. The beam profile is checked manually in a regular basis but is not a feedback parameter for the alignment of the mirrors.

The text has been extended. Lines 153-155:

(3) Author's changes in manuscript: "The optical cavity remains similar to that described in Horstjann et al., (2014), i.e., a V-resonator of ca. 100 cm3 volume formed between glued highly reflective mirrors (reflectivity, R = 99.995 %, diameter, d = 0.5", radius of curvature, roc = 100 cm, AT Films, USA) on the side of a Teflon coated aluminium cuboid."

(1) Comments from Referees: Page 5, line 146, "mode and modulation times. . ." it is not clear what mode and modulation times refer to, might be useful to define them discretely

(2)Response to RC2: The text has been extended for clarification at the beginning of 3.1. (Lines 181-190) and Figure 4 has been added:

(3) Author's changes in manuscript: "The mode time is defined as the time selected for the measurement in either amplification or background mode. The modulation time is the time taken for a complete measurement cycle, which comprises the sum of one amplification and one background mode. The PeRCEAS measurement cycle is illustrated in Figure 4. The $\Delta NO2$ for each detector is calculated from the ring down time of two consecutive modes using Eq.(1). If the mode time is adequately selected, the RO2* retrieved per measurement cycle is identical in both measurement lines, as the two reactors are operated out of phase with one another. The final RO2* data is calculated as the mean of the RO2* determined from the $\Delta NO2$ and eCL of both detectors for a given measurement cycle. The time resolution of the RO2* measurement is then equal to the mode time. After switching modes, a small pressure pulse leads to an oscillation of the NO2 signal. Consequently, the first 20 s of each mode are not used in data analysis. The time lag arising from the time taken for the sample flow between the CRDS detector and the point of switching is typically less than 8 s (see Table 3)."

(1) Comments from Referees: Page 5, line 160, "…detector temperature. For this different detector temperature gradients, $\Delta T$, where applied to modulated signals generated by varying the sampled NO2 concentration…" it's not clear why the investigators modulated NO2 while applying a temperature gradient to the detector. Would it not be easier to interpret if a constant mixing ratio gas was sampled while applying a temperature gradient? It is not clear from the text where this temperature gradient is and how it was applied. It would be useful to readers that are not familiar with optics, on why a temperature gradient of 7 degC would cause detector instability. It is also not clear from the text, what was done to address this flaw in the detector design, as the authors state earlier detector stability is paramount in overall instrument performance.

(2)Response to RC2: A modulated signal was applied in order to study the effect on similar signals to those measured by PeRCEAS during operation i.e., a NO2 between amplification and background modes. The text has been extended for clarification. Lines 211-221:

(3) Author's changes in manuscript: "Temperature changes of the detector affect: i) the diode laser emission, both its amplitude and wavelength, and ii) the mode matching between laser and detector, and consequently the $\tau 0$. The effect of the variations in $\tau$, resulting from changes in room or HALO cabin air temperatures, on the accuracy and precision of the $\Delta NO2$ determination was investigated by a series of laboratory experiments. For this, modulated concentrations of NO2 in the flow were generated. This was achieved by alternating between two selected NO2 concentrations once per minute. The temperature of the CRDS detector, T, and $\tau$ were then measured. Detector temperature gradients over a time t, i.e., $\Delta T/\Delta t$, determined by the temperature within the detector housing close to the photodiode, were induced by controlled changes in the room temperature.

Figure 6 shows the effect of introducing temperature perturbations in a modulated NO2 signal between 11.5 and 12.1 ppbv measured at 200 mbar and 23 °C. As can be seen in the figure, a temperature perturbation affects both precision and accuracy of the retrieved $\Delta NO2$. For temperature gradients up to $\Delta T/\Delta t \approx 7$ °C h-1 the experimental precision of the $\Delta NO2$ determination remains within ($2\sigma$) 150 pptv (= 7.3 x 108 molecules cm$-3$ at 200 mbar and 23°C). "

Concerning T stability as states in 4.2, Lines 406-407:

"Generally, in-flight variations in the HALO cabin temperature affect minimally the accuracy of the RO2* determination. "

(1) Comments from Referees: Page 6, line187, "...of the sampled O3 by NO to form NO2 also depends on the concentration of NO added to the sample flow and the time for reaction before reaching the detector.". This would be a good place to discuss how 30 ppmv NO was decided on for a reagent mixing ratio and discuss flow rate choices for both NO and CO.

(2)Response to RC2: The purpose of this section is to present the effect of changing gas concentration and flows. A balance between different and competing effects, leads

to a selection of the optimal operating conditions for a specific measurement campaign. 30 ppm NO has just been selected for some of the testing series in this work as an example for suitable reagent mixing ratio. The paper has been revised and additional measurement series have been included to illustrate some of the effects investigated.

(1) Comments from Referees: Page 6, line 192, "3.2.1 Effective Chain Length. . .", This section seems to describe a well established method documented in literature. The reviewer recommends shorting the description of the method and explain better the difference in DUALER I and DUALER II eCL.

(2)Response to RC2: This part of the text has been modified taking into account the comments of both RC1 and RC2.

(1) Comments from Referees: Page 7, line 216, "The model was initialized with 9% CO, 3 ppb O3, 50 pptv HO2. . ." why was 3 ppb O3 determined to be a representative mixing ratio for ozone? I may be misunderstanding the inlet chemistry, but it seems like missing 30 ppmv of NO would significantly affect the modeled CL. Assuming the box model initialization is correct, would it not be useful to vary the wall loss rate constants to match the eCL and determine if this wall loss is reasonable? It would also be useful to experimentally determine the wall loss of the inlet.

(2)Response to RC2: The model is initialised with 3 ppbv O3 because this is the mixing ratio produced during the calibration. Sensitivity studies have shown no significant change in the simulated eCL up to 100ppb O3. This is also confirmed by the experimental values shown in figure 17.

Following the recommendation of RC2, the simulations have been extended varying the wall losses in the pre-chamber to match the experimental values, as now explained in the text (Lines 280-306). PeRCEAS is operated at low pressures and the pressure regulation during operation prevents the direct experimental determination of the wall losses in the pre-chamber. However these are estimated by comparison with the model used.

(1) Comments from Referees: Page 7, line 222, "figure 8 shows eCL vs CL", the authors should include error bars on these data.

(2)Response to RC2: The error bars on the experimental data are now included and the figure has been extended with more experimental and model data.

(1) Comments from Referees: Page 7, line 228, "Figure 9 depicts the O3 decay simulated for 100 to 200 ppb..." these figures are somewhat confusing to the reviewer. One could take the 99% conversion time for each NO mixing ratio curve and plot all 4 conditions (ie pressure and O3 mixing ratio) on 1 figure for varying NO mixing ratio. Additionally, adding an inlet residence time reference line would be useful for helping the reader visualize what time limit you have on this reaction.

(2)Response to RC2: The figures have been modified as suggested by the referee.

(1) Comments from Referees: Page 7, line 234, "PAN and PPN thermal decomposition", the reviewer believes that experimental work is justified to confirm 'this source of radicals is considered to be negligible'.The box modelling done for CL prediction was shown to not capture the actual inlet system, so it's not clear why it would do a better job with modelling PAN and PPN. Figure 11 shows up to 10 pptv interference, this does not seem negligible to the reviewer.

(2)Response to RC2: The discrepancies between measured eCL and simulated CL for the reactor are attributed to errors in the estimate of the wall losses in the inlet prechamber and not to the overall performance of the box model. The chemistry involved in the formation of $CH_3O_2$ from the PAN decomposition has now been revised, and all the rates have been taken from the recommendations in JPL 15-10, with the equilibrium rate constant from Zhang et al (2011).

The results do not change for NO mixing ratios in reactor within 10 and 45 ppm.

During the flight the PeRCEAS reactors are located inside the pylon on the upper part of the HALO fuselage. As a consequence, the temperature in the reactors, which

is measured, remains below 290K (Figure 20 and 21). Consequently the maximum interference expected for the transition times given in Table 3 will be below 2 pptv, which is within the measurement error. The figure (now Figure 11) has been updated and the text has been extended for clarification (Lines 324-339).

(1) Comments from Referees: Page 8, line 252, "Figure 12 shows the variation of the eCL for 45 ppm NO within a pressure range...", The reviewer does not understand why this experiment was done with 45 ppm NO when the decided upon mixing ratio of NO addition seems to be 30 ppm NO for the rest of the paper. If this is a typo, it should be corrected, if not the experiment should be done at the actual mixing ratio the instrument is operated at.

(2)Response to RC2: The figure has been extended with similar measurements for 10 ppm NO. The results at 10 ppm NO confirm the recommendation of P= 100 mbar as the minimum operating pressure. The text (Lines 348-353) and the figure (now Figure 12) have been updated.

(1) Comments from Referees: Page 9, line 284, "Figure 15 depicts exemplary a comparison of spectra...", the reviewer does not believe including NO2 absorption cross section and detector spectra is a useful figure for the main text of this paper. Remove or include in the SI.

(2)Response to RC2: The figure has now been included in the supplementary information. Line 387:

(3) Author's changes in manuscript: "A sample comparison of spectra obtained for the three PeRCEAS detectors is included in the supplementary information (Figure SI-1)."

(1) Comments from Referees: Page 9, line 289, "In addition, the effective $\sigma$NO2 can be calculated by sampling known mixtures...", the reviewer does not believe including a time series of calibration gas addition to instrument is a useful figure for the main text of this paper. Remove or include in the SI.

(2)Response to RC2: The figure has now been included in the supplementary information. Lines 392-396:

(3) Author's changes in manuscript: "The result of applying Eq. (5) to the PeRCEAS detectors is depicted in Figure 15. The detectors sampled known mixtures of NO2 from commercial gas cylinders in synthetic air at 200 mbar as shown in the supplementary information (Figure SI-2)."

(1) Comments from Referees: Page 9, line 294, "The result of apply Eq. 4 to the PeRCEAS detectors at 200 mbar is depicted in Figure 17." It is more common to plot NO2 number density [molecules/cm3] vs. $\alpha$, as the slope has the physical meaning of the absorption cross section of NO2.

(2)Response to RC2: In the case of CRDS  is not measured but must be calculated by using Eq.1. In contrast, the variables plotted in the Figure 15 (originally Figure 17) are taken directly from the measurement and do not require a prior knowledge of 0. In addition the y intercept of the plot is the $1/\tau 0$ for each detector.

(1) Comments from Referees: Page 9, line 302, "The main source of uncertainty..." the authors previously mention detector drift due to temperature changes (figure 5), is this not a significant source of uncertainty as well?

(2)Response to RC2: Temperature drifts can be minimised during laboratory measurements. Concerning in-flight measurements, it is mentioned in section 4.3 that the temperature in the HALO cabin remains reasonably constant and therefore is not a significant source of uncertainty during the airborne measurement. An example can be seen in Figures 18 and 19. A sentence has been added to clarify this point. Line 406:

(3) Author's changes in manuscript: "Generally, in-flight variations in the HALO cabin temperature affect minimally the accuracy of the RO2* determination."

(1) Comments from Referees: Page 10, line 309, "Figure 18 shows the calculated eCL

from 14 radical calibrations. . . ." why were radical calibrations done with a NO mixing ratio of 45 ppm when the instrument is run at 30 ppm?

(2)Response to RC2: In this publication there is no intention to define one unique set of operating conditions for the PeRCEAS on HALO. Several concentrations and mixing ratios for NO were investigated. At 300 mbar the longest series of measurements was selected in the figure 18 to show the best statistics in the reproducibility and stability of eCL over time. These measurements were carried out to investigate the dependence of eCL on NO. Figure 18 (now Figure 16 in the revised version) is been replaced to show the data obtained at 300 mbar and 30 ppmv as suggested by the referee. The estimates of the corresponding eCL values for 10 ppm, 30 ppm and 45 ppm NO are now listed in Table 4.

(1) Comments from Referees: Page 10, line 330, "As can be seen in Figure 19. . .", it would be useful to plot actual O3 mixing ratio rather than set point in the top panel of this figure. It would also be useful to plot chamber pressure or channel pressure, as it seems like when the O3 mixing ratio is changed the NO2 signal displays a large amount of noise (50 ppb). It is also unclear what D1, D2, SG and BG stand for in this figure.

(2)Response to RC2: The original figure 19 is now Figure 17. The O3 mixing ratio is set at the calibrated O3 generator and cannot be measured during the PeRCEAS measurement, as it is converted in NO2.

The variation observed in the NO2 (the NO2 signal is not plotted) is not related to any change in the pressure but to the change in the O3 concentration in the sample, which takes approximately 1 minute to stabilise after changing the set point in the ozone generator.

The meaning of D1, D2, AP and BG are now explained in the figure caption. SG has been replaced by AP for clarification.

(1) Comments from Referees: Page 11, line 364, "The pressure regulation in PeRCA based airborne instruments results in lower eCL than ground based ones." This statement should be substantiated. The reviewer sees a range of eCL for aircraft instruments from 45 to 322 and ground based instruments a range of 91 to 1010. A description of why lower pressure or pressure regulation in general affects eCL would be useful.

(2)Response to RC2: As now described in section 2 the pressure in the inlet in PeRCEAS is controlled in the pre-chamber during the flight. This pre-chamber prior to the reactors causes radical losses which do not have the ground based instruments without pressure chamber and pressure regulation. The sentence has been extended for clarification. Lines 469-470:

(3) Author's changes in manuscript: "The pressure regulation in PeRCA based airborne instruments results in lower eCL than ground based ones. This is attributed to radical losses in the pre-chamber prior to the addition of reagent gases for the radical chemical amplification."

(1) Comments from Referees: Page 11, line 367, "the detection limit and uncertainty of PeRCA based instruments are strongly depending on the variation of O3 and NO2 in the sampled air mass. . ." The reviewer believes this statement should be substantiated with uncertainty and/or detection limit analysis to show the reader the magnitude of this affect. If the changes in O3 and NO2 are measured (as often are in aircraft field campaigns) can a correction method not be proposed and evaluated?

(2)Response to RC2: The effect of O3 changes in the RO2* determination has been shown and quantified in figure 17. Simultaneous measurements of O3 and NO2 on-board can be used as a reference for the correction of the background variations in individual cases, but a method that relies on other measurements to correct the background will have to deal with different instrumental resolutions and sources of errors of the instruments involved.

(1) Comments from Referees: Page 12, line 379, "As can be seen in figure 20..." this reviewer believes it would be useful to add the detector temperature (or deltaT used earlier) to this figure, as this was determined to be a large effect on ∆NO2 earlier in the manuscript. If NO2 and O3 mixing ratio data is available from the flight, this would be useful to include as well. (2)Response to RC2: The figure 20 (now figure 18) has been modified to include the inlet and detector temperatures as suggested by the referee.

(1) Comments from Referees: Page 12, line 383, "Figure 22..." the reviewer does not believe this or figure 21 add considerable value to this paper. Recommend removing or moving to SI.

(2)Response to RC2: This figure has been removed as suggested by the referee.

(1) Comments from Referees: Page 12, line 387, "...illustrate the improvement in the dynamical stability achieved in successive airborne deployments..." the reviewer finds it difficult to see the improvements made in the measurement by looking at time series data. Suggest thinking of a different way of presenting this conclusion. It would also be very useful to add a comparison to the previous generation of instrument somewhere in the paper.

(2)Response to RC2: The figures 18 and 19 have been modified to make clearer the improvement made as suggested by the referee.

Please also note the supplement to this comment:
https://www.atmos-meas-tech-discuss.net/amt-2019-359/amt-2019-359-AC3-supplement.pdf

**Supplement:**

[Figure]

Figure SI-1: Emission spectrum of the lasers used in the PeRCEAS detectors: AB, FH, and FR. The high resolution $\sigma_{NO_2}$ at 294 K from Vandaele et al (2002) is also depicted for comparison.

[Figure]

Figure SI-2: PeRCEAS measurement of known $NO_2$ mixing ratios using the FH detector at 200 mbar and 500 ml/min sample flow. The mixing ratios of $NO_2$ in synthetic air set for the experiment are indicated in blue."

---

## Author Comment (AC4) · 17 Mar 2020

The dependency on water vapour only becomes significant at higher water number concentrations for NO 45 ppm at 300 mbar as shown in figure 14. New measurements at 10 and 30 ppmv NO at 300 mbar show a clear dependency with [$H_2O$] which has to be taken into account in the analysis of ambient air measurements at these operating conditions. The water vapour dependence of eCL decreases significantly from 10 ppm to 45 ppm. For water vapour concentrations in the range of 4 -14 x $10^{16}$ molecule cm$^{-3}$ of relevance for HALO flights, this results in the water vapour impact on the eCL being negligible. This is shown more explicitly in the new figure 14 in the manuscript.

As described in Hastie et al., (1991) and Reichert et al., (2003), the chain length (CL) of a PERCA reactor can be expressed using a resistance model as follows:

$$\frac{1}{CL} = \left( \frac{1}{CL_{HO_2}} + \frac{1}{CL_{OH}} \right)$$

$$\frac{1}{CL} \approx \frac{1}{P_{NO_2}} \cdot \left( \frac{\sum HO_2 \text{ removal rates}}{\sum HO_2 \text{ propagating rates}} + \frac{\sum OH \text{ removal rates}}{\sum OH \text{ propagating rates}} \right)$$

And if only the predominant processes are considered, as:

$$\frac{1}{CL} \approx \frac{1}{P_{NO_2}} \cdot \left( \frac{k_{w_{HO_2}} \cdot k_{HNO_3}[NO][M]}{k_{NO+HO_2}} + \frac{k_{w_{OH}} \cdot k_{HONO} \cdot [NO]}{k_{CO+OH} \cdot [CO]} \right)$$

$P_{NO_2}$ is the probability of radical conversion into $NO_2$, $k_{w_{HO_2}}$ and $k_{w_{OH}}$ are the wall losses of $HO_2$ and OH respectively, $k_{NO+HO_2}$ is the rate constant for $NO_2$ production from $HO_2$ + NO reaction, $k_{HONO}$ is the rate constant for HONO formation from OH + NO reaction, $k_{CO+OH}$ is the rate constant for $HO_2$ formation from OH + CO reaction. In the presence of $H_2O$, $HO_2$ water clusters are formed as postulated by Reichert et al., (2003).

Our interpretation of these novel results is that as the NO is increased from mixing ratios of 10 to 45 ppm at 300 mbar, the $CL_{OH}$ is reduced, because the rate of the chain termination termolecular reaction of OH with NO making HONO increases, and the $CL_{HO_2}$ is increased, because the rate of the propagation reaction between NO and $HO_2$ increases. As a result the CL begins to be dominated by the $CL_{OH}$, which is independent of water vapour.

The new series of measurements are now included in Figure 14 and the text has been extended for clarification (Lines 359-371):

"In this work radical mixtures were sampled at 25 °C for relative humidity between 2 % and 25 %. This leads to a ca. 20 times increase in the absolute [$H_2O$]. These conditions cover the [$H_2O$] expected for a larger T range (-20 – 30 °C) during airborne measurements in the free troposphere at 200 and 300 mbar inlet pressures. Figure 13 shows the [$H_2O$] in the air probed versus the [$H_2O$] in the inlet for real measurements on board of the HALO aircraft. The results in figure 14 for 45 ppm NO ([NO] 3.28 x $10^{14}$ molecules cm$^{-3}$ at 300 mbar) indicate that variations in the sample humidity do not lead to additional uncertainty in the $RO_2^*$ retrieval as the PeRCEAS

eCL remains invariable within the experimental error up to $[H_2O] \sim 1.4 \times 10^{17}$ molecules $cm^{-3}$. In contrast, for 10 ppm and 30 ppm NO in the reactor ([NO] $7.29 \times 10^{13}$ and $2.19 \times 10^{14}$ molecules $cm^{-3}$ at 300 mbar) the eCL shows a clear dependency with the ambient $[HO_2]$. The comparison with the eCL values obtained by Reichert et al (2003) at 1 atmosphere indicate a eCL dependency on $[H_2O]$, temperature and pressure having a different pattern for 45 ppm NO in the reactor. This is explained by invoking the competition in the amplification chain length, CL, between $HO_2$ and OH removal rates, as explained in Hastie et al., (1991) and Reichert et al., (2003). At $[NO] \sim 3.28 \times 10^{14}$ molecules $cm^{-3}$ the CL begins to be dominated by the rate of the termination termolecular reaction of OH with NO, which is independent of water vapour. This eCL dependency has to be taken considered in the analysis of ambient air $RO_2^*$ measurements."

---

## Author Comment (AC6) · 20 Mar 2020

The authors would like to thank the referees for their valuable comments. The comments helped us to improve the quality of the manuscript. The response to the comments has been included as a supplement pdf file. We hope all the issues raised have been addressed properly.

We also thank Dr. Anna Novelli for being the associate editor of this manuscript.

**Response to RC1**

**General comments and suggestions**

**Comments from Referee**: The researchers appear to accept the basic characteristics of the inlet/reactor as given. The paper does not justify the selection of flow rates, inlet/reactor volume and composition, and thus the corresponding reaction time. This reviewer sees this approach as flawed. Perhaps this has been well thought out, and previously published, but not included in this paper. Suggest that the authors include some discussion of why these parameters were selected, and what the compromises and advantages in their selection are.

**Response to RC1:** The objective of the present publication is to explain the dominant factors affecting the overall performance and accuracy of PeRCEAS for the determination of the hydroperoxyl, $HO_2$, and organic peroxy radicals, $RO_2$, which react with NO to form $NO_2$, when deployed on the the HALO aircraft. The operating conditions of PeRCEAS are optimised for the specific sampling position used, cabin location of the instruments, safety requirements and the type of flight tracks and altitude profiles which were flown by HALO. Generally, these limitations are often different in different campaigns. For this reason, this manuscript does not aim at describing a unique universal set of PeRCEAS operating conditions. In practice the mechanical constraints and the safety requirements of the HALO (e.g. the size and weight of the pylon for the inlet, the amount of CO permitted on board) determine the volume and shape of the reactors and partly the range of flows of the gases and the residence time within the PeRCEAS. More detailed information about the inlets DUALER I and DUALER II and their differences are now provided in section 2 (see answers to specific comments).

**Comments from Referee**: One aspect is that higher NO levels used in the studies reported in this paper (30-45 ppmv) result in lower sensitivity to $CH_3O_2$ (and other $RO_2$) compared to previous values because of a faster rate of $CH_3O+NO+M$. High NO also converts a greater fraction of ambient ozone to $NO_2$, although complete conversion is not necessary. Most other chemical amplifiers used NO reagent mixing ratios of 2 to 6 ppmv.

**Response to RC1:** The selection of the concentration rather than the mixing ratio of NO is indeed a critical issue for the PERCA approach. High concentrations of NO are required to guaranty the full titration of $O_3$ in $NO_2$ to capture fast variations of $O_3$ within a measurement cycle. The total conversion of $O_3$ to $NO_2$ in the system enables the quantification of radical data at a 60s temporal resolution, as explained now in figure 4. In this way, the horizontal resolution of the PeRCEAS airborne measurements, which depends on the speed and altitude of HALO, is typically between 7 and 15 km. Longer modulation cycles than 120 s result in noisy and unrepresentative averages for ambient measurements in air masses having significant short term variability of $O_3$ and $NO_2$. Provided the partial conversion is stable and identical for both reactor-detector lines of the PeRCEAS, we agree with RC1 that the partial conversion of $O_3$ to $NO_2$ is sufficient to determine $RO_2^*$. Malfunctioning of one of the detectors yields the $RO_2^*$ to be

determined at 120 s. In this case, if $O_3$ is completely converted and the simultaneous $O_3$ and $NO_2$ measurements on board are of sufficient accuracy, a $RO_2^*$ time resolution of 60 s may still be feasible.

40

The majority of PERCA measurements in the literature using NO concentration in the range 3-6 ppm were made at 1 atmosphere i.e. concentrations of $(7.3\text{-}14.6) \times 10^{13}$ molecule $cm^{-3}$. At 300 mbar these correspond to the mixing ratio range 10-20 ppm. We expect that at concentrations of NO of $14.6 \times 10^{13}$ molecule $cm^{-3}$, i.e. a mixing ratio of 20 ppm, the ratio of $eCL(CH_3O_2)/eCL(HO_2)$ is 65 % and 40 % for 45 ppm NO.

45 As now explained in the section 3.3. and in the response to the corresponding specific question below, the water vapour dependence of eCL decreases significantly from 10 ppm to 45 ppm NO ([NO] $7.29 \times 10^{13}$ to $3.28 \times 10^{14}$ molecules $cm^{-3}$ at 300 mbar). The results in figure 14 for 45 ppm NO indicate that variations in the sample humidity do not lead to additional uncertainty in the $RO_2^*$ retrieval as the PeRCEAS eCL remains invariable within the experimental error up to $[H_2O] \sim 1.4 \times 10^{17}$ molec $cm^{-3}$.

50 The final selection of [NO] will be a balance between having stable eCL with respect to water vapour concentrations and having smaller eCL for $RO_2$ measurements. As now shown in Table 2 the eCL of $RO_2$ for 300mbar and NO 45 ppm, assuming to be $CH_3O_2$ the dominant atmospheric $RO_2$, would be 40% of the eCL for $HO_2$. The sum of $HO_2$ + 40% $RO_2$ can then be compared with atmospheric model values to test our understanding of the production of $HO_2$ an $RO_2$ in air masses sampled in flight.

55

**Comments from Referee**: It is not apparent why the response time of the system is so slow. For example, the Figure 5 caption states that 20 seconds is eliminated after change in $NO_2$. Given a reactor transit time of 3 seconds, this seems extreme. There may be delays, which can be accounted for in data analysis, that are different than transition to the correct value after perturbations.

60 **Response to RC1:** The PeRCEAS operating conditions have now been explained more in detail in the text and supplementary information (see answers to specific comments). During calibration measurements, different $NO_2$ mixing ratios are generated by the dilution of a mixture of $NO_2$ in synthetic air from a commercially certified gas cylinder (Airliquid 10 ppmv $NO_2$ in $N_2$). As illustrated in the Figure RC1_I below, the flow controller used for adding different $NO_2$ mixing ratios to the detector requires approximately 10s second to reach the set value after a 65 change. If this time is added to the time required for the probe to reach the detector (see Table 3; 5.27 s), at least 16 seconds of data should be ignored after a change in the $NO_2$ mixing ratio.

[Figure]

Figure RC1_I: PeRCEAS response to a step change in sampled $NO_2$ mixing ratio

70    Comments from Referee:        Part of the answer may be in Figure 19, where step functions in the $O_3$ concentration result in perturbations lasting about 40 seconds.

**Response to RC1:**      The $O_3$ concentration and mixing ratios were changed by adjusting the speed of the flow passing through a Hg lamp, which photolyses $O_2$.  When the $O_3$ concentration is changed, it takes a time for the ozone generator to stabilise the $O_3$ concentration in its flow and a time lag required for the probe to reach the
75    detector. This is the reason for the changes in figure 19 (now figure 17).

**Comments from Referee**:      Also, in Figure 20, the pressure variations last for a long time during and after altitude changes. There are also pressure fluctuations even when the aircraft is not changing altitude. This implies need for better pressure control. Perhaps the PID parameters of the pressure controller have not been adjusted
80    properly. This is very important to get correct. Though improved, such fluctuations are still apparent in Figure 23. They add unnecessary noise to the measurements. Suggest adding more discussion of the pressure control system (manufacturer, model, adjustment procedures) and the response time of the system to step changes in $NO_2$ concentrations to allow the reader to better understand these issues.

**Response to RC1:**      To avoid misunderstanding the figure 20 (new figure 18) has been improved. In this figure
85    the dynamic pressure changes are depicted which are the cause of fluctuation in the inlet. The changes in the dynamic pressure arise from altitude changes and from changes of the aircraft velocity (i.e., including turning) and air turbulence. Under laboratory condition the time constant for the pressure system to stabilise after a perturbation in PeRCEAS is 15s. The error induced for the $RO_2^*$ measurements is shown in figure 20 (new figure 18) to be small or negligible as a result of turbulence or HALO velocity changes. After identifying the measurements influenced by
90    changes in the dynamic pressure, only $RO_2^*$ which have pressure fluctuations of less than 2 mbar in 60 s, the modulation time, are primarily used for analysis. The relevant paragraph now reads as follows (Lines 481- 487)

**Author's changes in manuscript**:      "As can be seen in the figure 18, the dynamic pressure variations experienced by the aircraft influence the stability of the inlet pressure. These changes are attributed to altitude changes, air turbulence, and changes
95    in aircraft velocity, including turning, of the aircraft. The effect of inlet pressure instabilities on the retrieved $\Delta NO_2$ is not exactly identical for both detector signals. This leads to additional uncertainty in the $RO_2^*$ determination when using the procedure discussed in section 4.3. For the data analysis, pressure spikes within 1 minute standard deviation higher than 2 mbar are identified and flagged. This approach enables data with large error due to dynamic pressure changes to be identified. Overall the error in the retrieved $RO_2^*$ is  around 20 % in the measurement period shown in figure 18."

100

**Comments from Referee**:      Finally, why are the chain lengths reported in this paper so much lower that previous publications (Table 3)? In fact, values reported in this paper (28-38, Figure 18) are much lower than what is reported under "This Work" in Table 3 and would be the lowest values for the CO/NO chemical amplifier in the table (there are lower values for the ethane/NO chemical amplifier). This compromises the potential quality of the measurements.
105    Perhaps this bears on the question about optimization of the instrument earlier. It seems that chain lengths of 100 or more are possible (at lower reagent NO mixing ratios). Some explanation in the paper is needed to explain this.

**Response to RC1:** The text has been extensively rewritten to address this issue. We hope that we have removed any misunderstandings with respect to the values reported in the text and in the old table 3 (new table 4).

110

**Specific comments and suggestions**

**Comments from Referee**: Page 1, line 16. "…for the airborne measurements in the…"

**Response to RC1:** It has been corrected as suggested by the referee.

115

**Comments from Referee**: Page 1, line 18. "…instrumental channels successfully captures short term…".

**Response to RC1:** It has been corrected as suggested by the referee.

**Comments from Referee**: Page 1, line 20. Not sure why the word "gradients" is used here. How about
120 "…range of atmospheric pressures and temperatures expected…"?

**Response to RC1:** It has been changed as suggested by the referee.

**Comments from Referee**: Page 1, line 24. The phrase "…collectively known at $RO_2^*$…". Is it true that $HO_2 + RO_2$ is "known" as $RO_2^*$? Is this the term accepted by the community? This reviewer suggests just using
125 "$HO_2 + RO_2$" instead.

**Response to RC1:** $RO_2^*$ is the term defined as $RO_2^* = HO_2 + \sum RO_2 + OH + \sum RO$ and used specifically for the PeRCA measurements. The text has been accordingly modified.

**Comments from Referee**: Page 1, line 29. Suggest removing the summation symbol, since there is a plus sign
130 used.

**Response to RC1:** It has been corrected as suggested by the referee.

**Comments from Referee**: Page 1, line 32-33. Suggest "…photolyzed to ultimately produce…"

**Response to RC1:** It has been changed as suggested by the referee.

135

**Comments from Referee**: Page 2, line 34. Suggest "Overall, $HO_2 + RO_2$ influences the…"

**Response to RC1:**     It has been changed as suggested by the referee.

 **Comments from Referee**:     Page 2, line 41. Not sure what is meant by "those RO₂".

140   **Response to RC1:**     The text has been extended for clarification. Lines 41-50:

 **Author's changes in manuscript**:     "The chemical amplification technique (Cantrell and Stedman, 1982; Hastie et al., 1991) has been used to measure the sum of peroxy radicals. The Peroxy Radical Chemical Amplification (PeRCA) converts by addition of NO and CO, $HO_2$ and most atmospherically significant $RO_2$ to $NO_2$. The OH formed in the reaction cell reacts with

145   CO to reform $HO_2$ in a chain reaction. Oxy, alkoxy, hydroxy and alkylperoxy radicals $(OH + \sum RO + HO_2 + \sum RO_2)$ are converted into $NO_2$. As the RO and OH abundances in the troposphere are much lower than those of $HO_2$ and $RO_2$, PeRCA measures to a good approximation the sum of peroxy radicals collectively known as $RO_2^*$, $(RO_2^* = HO_2 + \Sigma RO_2$ ; being R any organic chain), which convert NO to $NO_2$. The rate coefficients of the $HO_2$ and $RO_2$ reactions with NO are very similar (Lightfoot et al, 1993). Large $RO_2$ which do not react with NO to form $NO_2$ are not detected, and are assumed to be negligibly

150   small compared to the sum of $HO_2 + \sum RO_2$ concentrations. $HO_2$ and $CH_3O_2$ are the dominant peroxy radicals present in an air mass in most conditions."

 **Comments from Referee**:     Page 2, line 43. Suggest "…compared to the total amount of HO₂+RO₂, with HO₂…"

155   **Response to RC1:**     This  has been changed as suggested by the referee.

 **Comments from Referee**:     Page 2, line 53. Should be "Kanaya".

 **Response to RC1:**     This has been corrected as suggested by the referee.

160   **Comments from Referee**:     Page 2, line 54. Suggest a different word than "largely". Suggest "The interference by some RO₂…"

 **Response to RC1:**     This has been changed as suggested by the referee.

 **Comments from Referee**:     Page 3, line 72. Add space "Peroxy Radical…".

165   **Response to RC1:** This has been corrected as suggested by the referee.

 **Comments from Referee**:     Page 3, line 74. Suggest "…in a previous publication…".

 **Response to RC1:** This has been changed as suggested by the referee.

170    **Comments from Referee**:    Page 3, line 79. Suggest "…where $\Delta NO_2$ is the $NO_2$ formed…"

**Response to RC1:** This has been changed as suggested by the referee.

**Comments from Referee**:    Page 3, line 84. Suggest "…spectroscopic measurement technique…"

**Response to RC1:** This has been changed as suggested by the referee.

175

**Comments from Referee**:    Page 3, line 87. $NO_2$ comes from radical amplification and from the background ($O_3$ conversion). Suggest changing the sentence to reflect this.

**Response to RC1:** The sentence has been reworded. Line 102:

180    **Author's changes in manuscript**:    "In PeRCEAS the absorber of interest is $NO_2$ which is formed in both the amplification and the background modes. "

**Comments from Referee**:    Page 3, line 92-94. Suggest also saying that $c_0$ is the speed of light in a vacuum (stated later in the paper).

185    **Response to RC1:** The text has been extended as suggested by the referee.

**Comments from Referee**:    Page 3, line 92. Suggest "…are, respectively, the absorption…"

**Response to RC1:** This has been changed as suggested by the referee.

190    **Comments from Referee**:    Page 3, line 95. Suggest "…used for ground-based measurements…"

**Response to RC1:** This has been corrected as suggested by the referee.

**Comments from Referee**:    Page 3, line 96-7. Suggest "…the particular constraints related to airborne measurement…".

195    **Response to RC1:** This has been changed as suggested by the referee.

**Comments from Referee**:    Page 3, line 99. Suggest "In this study, the specifications…are described based on thorough laboratory…"

**Response to RC1:** The text has been changed as suggested by the referee.

200

**Comments from Referee**: Page 4, line 107. Suggest (if correct) "…and located outside the HALO fuselage…"

**Response to RC1:** The pylon is a part of the fuselage. The sentence has been reworded. Line 123:

 **Author's changes in manuscript**: "….installed inside a pylon located on the outside of the HALO fuselage"

205  **Comments from Referee**: Figures 1, 2, and 3. While photographs can be nice, schematic diagrams are more useful to see the path of sample, reagents, and signals throughout the system. Suggest limiting to only one or two photographs and add diagrams to show the details.

**Response to RC1:** The photos have been replaced by schematic diagrams as suggested by the referee.

210  **Comments from Referee**: Page 4, line 117. Mixing and pressure regulation are mentioned here, but no detail is given. This is relevant to the general comment given earlier. Suggest adding more detail in the text and perhaps in Figures 1 and 2. Suggest discussing how DUALER I and II are different and describe why the changes do indeed result in improved performance.

**Response to RC1:** The text has been extended with the description of the DUALER operation and Figure 2 has been
215  included to highlight differences between DUALER I and II. Lines 127 to 151:

 **Author's changes in manuscript**: "Briefly, sampled air enters PeRCEAS through the DUALER pre-chamber, which is at a lower pressure than that outside of the HALO, through an orifice in a truncated cone, i.e. a nozzle. From this pre-chamber the air is pumped simultaneously through the two flow reactors and a bypass line. At the upper addition point a mixture of CO or $N_2$ and NO enters each reactor. At the lower addition point, a flow of $N_2$ or CO enters each reactor. This enables the CO and $N_2$
220  flows in the two reactors within the DUALER to be switched simultaneously but out of phase with one another from the upper to the lower addition point. At the addition points, the reagent gases enter the reactor through eight circular distributed 1 mm holes to facilitate the rapid mixing with the sampled air. During measurements, the pressure in the pre-chamber and both reactors is held constant. However, there is a small pressure fluctuation during the switching of flows between the upper and lower mixing point. The flow passing through each reactor enters a CRDS $NO_2$ detector. Afterwards, the sample flows together with the air
225  from the bypass line are scrubbed for CO and NO and exhausted by the pump.

The DUALER inlet comprises two PeRCA chemical reactors having alternating measurement modes, which are out of phase with one another. During the first part of the measurement cycle, the first reactor and detector are in amplification mode, while simultaneously the second reactor and detector are in background mode. In the second part of the cycle, the CO addition point in both reactors is switched. Consequently, the first reactor and detector are then in background mode while the second reactor and
230  detector are in amplification mode. In the analysis of the measurements, the amplification and background signals from both detectors are combined appropriately. This improves accuracy and temporal resolution of the resultant $RO_2^*$ data set (see 3.1).

In the DUALER, a stable pressure in the pre-chamber is achieved by a pressure regulator, which controls the flow through the bypass line. As noted the flow rate through the reactors is held constant during measurements. Consequently, when the outside air pressure changes, the bypass flow rate from the pre-chamber is changed. The outer dimensions, shape, form and weight of the
235  DUALER are constrained by the inlet pylon in use with the research aircraft HALO. After the first version of the DUALER (from now on called DUALER I) was flown, the inner dimensions of the pre-chamber were further optimised to reduce the wall

losses and turbulence in the pre-chamber. For this, in the DUALER II the volume of the pre-chamber was increased by extending its vertical extent, the length of the truncated cone on top of the reactors was reduced in 3 mm, and the volume of the reactors was increased to 130.5 ml from the 112 ml in DUALER I. These changes resulted in a higher eCL and improved pressure stability in DUALER II as compared to DUALER I. Figure 2 shows the upper part of both DUALER I and DUALER II."

**Comments from Referee**: Page 4, line 124. The term "piezo electric stack" is not a common term and needs more description.

**Response to RC1:** A piezo electric stack features a longitudinal deformation when voltage is applied. A mirror mounted on piezo electric stack was used to achieve mode matching between the single mode laser and resonator in Hortsjann et al., 2014. This is not used in the current configuration of PeRCEAS.

The text has been accordingly modified (now lines 158-159).

**Author's changes in manuscript**: "With this, the fine adjustment of the laser is simplified and improved, and the piezo electric stack used to achieve mode matching between the single mode laser and the optical cavity in Hortsjann et al., (2014) becomes unnecessary and is removed."

**Comments from Referee**: Page 4, line 128. Suggest adding the manufacturer of the beam camera (MKS Ophir). This is very cool, by the way!

**Response to RC1:** The text has been extended. Line 162:

**Author's changes in manuscript**: "During alignment procedures and for test purposes, a beam camera (BM-USB-SP907-OSI, Ophir Spiricon Europe GmbH) monitors the beam profile and simplifies the identification of misalignments or loss of performance of the optical system."

**Comments from Referee**: Page 4, line 132-3. Suggest "…other sensor data such as pressure, flow, temperature, and humidity." You don't need "etc." if using "like" or "such as".

**Response to RC1:** This has been corrected as suggested by the referee.

**Comments from Referee**: Page 5, line 149. Would "occasionally" be better than "exceptionally"?

**Response to RC1:** This has been changed as suggested by the referee.

**Comments from Referee**: Page 5, line 150-152. The term that should be used is "Allan Variance" rather than "Allan Deviation". Perhaps the equation for its calculation should be given, since there is also modified Allan Variance that can be used. Also, give one or two references (e.g. Allan, 1966; Allan et al., 1991; Allan and Levine, 2016). Suggest "…was investigated using calculated Allan Variance in the measurement…". Also "…the optimum

270 integration time for the three PeRCEAS detectors is between 20 s…". Do you use 20-30 second averaging in the data analysis? The timing of the instrument cycling should be shown, perhaps in Figure 1 or in a separate figure.

**Response to RC1:** The analysis of the Allan deviation was used to infer the detection limit of the measurement resulting from random noise. The plot of the Allan deviation has been replaced by another plot of the Allan variance as suggested by the referee. The text has also been extended for clarification. Lines 195-206:

275

**Author's changes in manuscript:** " To optimize the mode time and thus also the modulation cycle, the Allan variance (Allan, 1966; Werle et. al., 1993) was analysed for PeRCEAS. Given a time series of N elements and a total measurement time $t_{acq}$, $t_{acq} = f_{acq} \cdot N$, where $f_{acq}$ is the frequency of acquisition, then the Allan variance is defined as:

$$\sigma_x^2(\tau) = \frac{1}{2} \langle (x_{i+1} - x_i)^2 \rangle_\tau \qquad (Eq.2)$$

280 where $x_i$ is the mean over a time interval of a length $\tau$, being $\tau = f_{acq} \cdot m$; and m the number of elements in a selected interval. The use of $\langle \dots \rangle$ denotes the arithmetic mean. The square root of the Allan variance is the Allan deviation. For random noise, the Allan deviation at any given integration time determines the detection limit of the measurement.

The Allan variance plot for measurements of 5.6 ppbv $NO_2$ at 200 mbar and 23 °C is shown in figure 5. As can be seen, the optimal averaging time for the three PeRCEAS detectors is in the range between 20 s and 50 s. The corresponding minimum ($2\sigma$)
285 detectable mixing ratio is < 60 pptv ($3.15 \times 10^8$ molecules $cm^{-3}$ for these P and T conditions). Slow temperature drifts over longer averaging times impact on both the laser and the resonator characteristics. This behaviour is observed for averaging times longer than 60 s."

**Comments from Referee:** Page 5, line 155. The phrase "…over the modulation time…" need explanation. This
290 might be apparent with addition of a figure showing the instrument cycle timing.

**Response to RC1:** The text has been extended for clarification at the beginning of 3.1. (Lines 181-194), and Figure 4 has been added:

**Author's changes in manuscript:** "The mode time is defined as the time selected for the measurement in either amplification or background mode. The modulation time is the time taken for a complete measurement cycle, which comprises
295 the sum of one amplification and one background mode. The PeRCEAS measurement cycle is illustrated in Figure 4. The $\Delta NO_2$ for each detector is calculated from the ring down time of two consecutive modes using Eq.. If the mode time is adequately selected, the $RO_2^*$ retrieved per measurement cycle is identical in both measurement lines, as the two reactors are operated out of phase with one another. The final $RO_2^*$ data is calculated as the mean of the $RO_2^*$ determined from the $\Delta NO_2$ and eCL of both detectors for a given measurement cycle. The time resolution of the $RO_2^*$ measurement is then equal to the mode time. After
300 switching modes, a small pressure pulse leads to an oscillation of the $NO_2$ signal. Consequently, the first 20 s of each mode are not used in data analysis. The time lag arising from the time taken for the sample flow between the CRDS detector and the point of switching is typically less than 8 s (see Table 3).

Typically, 650 to 800 ring down times of the $NO_2$ absorption are averaged per second and the measurement of $NO_2$ is made at 1 Hz. Individual ring down times are occasionally saved for sensitivity studies. Modulation and mode times are selected empirically. The optimised values are a compromise between the time taken for the detector signal to stabilise after the $CO/N_2$ flow is switched between the addition points, and the temporal variability of the chemical composition of the air probed"

**Comments from Referee**: Page 5, line 161. Suggest "…signals generated that minutely varied the sampled $NO_2$…"

**Response to RC1:** The paragraph has been rewritten for clarification. Lines 211-217:

**Author's changes in manuscript**: "Temperature changes of the CRDS affect: i) the diode laser emission, both its amplitude and wavelength; ii) the mode matching between laser and detector, and consequently the $\tau_0$. The effect of the variations in $\tau$, resulting from changes in room or HALO cabin air temperatures, on the accuracy and precision of the $\Delta NO_2$ determination was investigated by a series of laboratory experiments. For this, modulated concentrations of $NO_2$ in the flow were generated. This was achieved by alternating between two selected $NO_2$ concentrations once per minute. The temperature of the CRDS detector, T, and $\tau$ were then measured. Detector temperature gradients over a time t, i.e., $\Delta T/\Delta t$, determined by the temperature within the CRDS housing close to the photodiode detector, were induced by controlled changes in the room temperature."

**Comments from Referee**: Page 5, line 164. "…time resolution using each background…". This will be more obvious with a graphical representation.

**Response to RC1:** Now the figure 4 depicts the PeRCEAS measurement cycle.

**Comments from Referee**: Page 5, line 167. "… molecules $cm^{-3}$ for typical measurement conditions".

**Response to RC1:** This value refers to the value in molecules $cm^{-3}$ corresponding to 150 pptv at the particular T and P of the measurement. The text has been changed for clarification. Line 220:

**Author's changes in manuscript**: "….the experimental precision of the $\Delta NO_2$ determination remains within ($2\sigma$) 150 pptv (= $7.3 \times 10^8$ molecules $cm^{-3}$ at 200 mbar and 23°C). "

**Comments from Referee**: Page 5, line 170. "It is inferred from laboratory calibrations that a 60 s modulation cycle is an optimum comprise between…". The term "modulation cycle" is not apparent here, but perhaps would be with a graphical representation of the instrument cycle. Also, add more discussion why fluctuations last so long (20 s).

**Response to RC1:** Figure 4 now depicts the measurement cycle. The text has also been extended as suggested by RC1 to clarify the modulation cycle (see lines 181-190:

**Author's changes in manuscript**: "The mode time is defined as the time selected for the measurement in either amplification or background mode. The modulation time is the time taken for a complete measurement cycle, which comprises the sum of one amplification and one background mode. The PeRCEAS measurement cycle is illustrated in Figure 4. The $\Delta NO_2$

for each detector is calculated from the ring down time of two consecutive modes using Eq.. If the mode time is adequately selected, the $RO_2^*$ retrieved per measurement cycle is identical in both measurement lines, as the two reactors are operated out of phase with one another. The final $RO_2^*$ data is calculated as the mean of the $RO_2^*$ determined from the $\Delta NO_2$ and eCL of both detectors for a given measurement cycle. The time resolution of the $RO_2^*$ measurement is then equal to the mode time. After switching modes, a small pressure pulse leads to an oscillation of the $NO_2$ signal. Consequently, the first 20 s of each mode are not used in data analysis. The time lag arising from the time taken for the sample flow between the CRDS detector and the point of switching is typically less than 8 s (see Table 3)."

**Comments from Referee**:     Page 5, line 172. The detection for $NO_2$ should be performed with a background level of ozone in the sample, since this is how ambient measurements are performed. Was this is the case?

**Response to RC1:** As described in the text, the detection limit for $NO_2$ was determined by measuring a modulated signal generated by dilution of $NO_2$ from commercial standard cylinders in synthetic air to get 11.5 and 12.1 ppbv $NO_2$ as background and amplification signals respectively. This is equivalent to adding a background level of $O_3$, since $O_3$ is converted totally in $NO_2$ before reaching the detector. In further complementary measurements it has not been observed any significant variation in the $NO_2$ detection limit for variations up to 100 ppb in the $O_3$ background produced by an ozone generator.

**Comments from Referee**:     Page 5, line 173-4. It is not obvious that larger modulation times lower the representativeness of the averages. Do you mean variability in peroxy radicals or in the background? For the latter, the instrument is continually measuring the background, and it should be well accounted for. If you mean the peroxy radicals, while one-minute (or quicker) data are nice, longer averages can still be useful in adding understanding of tropospheric free radical behavior. Suggest rewording the last sentence of this paragraph.

**Response to RC1:**  In instruments based on PERCA modulated signals, the modulation time determines the resolution. In contrast to ground based measurements, as the airborne platform additionally moves horizontally and vertically, modulation times longer than 120 s can be critical to mirror the peroxy radical and background variabilities in the encountered air masses.

This part of the text has been rewritten to address the criticism of the referee and moved to 3.2.2 for clarification (Lines 310-316):

**Author's changes in manuscript**: "As explained in section 3.1. the simultaneous use of two detectors measuring out of phase results in the temporal resolution of the $RO_2^*$ data being 60s. In this way, the horizontal resolution of the PeRCEAS airborne measurements, which depends on the speed and altitude of HALO, is typically between 7 and 15 km. Longer modulation cycles than 120 s result in noisy and  unrepresentative averages for ambient measurements in air masses having significant short term variability of $O_3$ and $NO_2$. To keep the temporal resolution of the $RO_2^*$ data to be equal to the mode time, the rapid and complete conversion of ambient $O_3$ into $NO_2$ within the PeRCEAS is required. For this, the NO concentration added at the inlet has to be sufficient for a complete titration of the sampled $O_3$ before reaching the detector."

**Comments from Referee**:     Page 6, line 176. "Sample and reagent gas flows...".

**Response to RC1:** It has been changed as suggested by the referee.

**Comments from Referee**:      Page 6, line 176-184. This discussion of reaction time should also include discussion of how the size of the reactor was selected, since it also affects the reaction time (reactor volume / total flow). Perhaps this would be a good place to discuss the approach to ensuring mixing of reagent gases with the ambient air sample. How was this done? Were fluid dynamical calculations performed? Were flow visualization approaches used? Related to this: how do you ensure that no components of the inlet system are leaking?

**Response to RC1:** The length of the reactor is limited by the design of the pylon. Therefore the reaction time is constrained by the mechanical borders and the flow conditions selected. The description in section 2 has been extended for clarification.

The reagent gases are added to the reactor through eight 1mm holes distributed circularly around the reactor tube in order to maximise the mixing with the ambient air sample.

The tightness of the inlet is controlled regularly by applying an overpressure of 0.8 mbar after closing all openings of the inlet and using leak soap at the connections. CO leakages in the aircraft are a critical safety issue in the aircraft. Therefore the inlet tightness is checked before installation and the potential critical points of the instrument are tested by adding He overpressure and using a He detector.

**Comments from Referee**:      Page 6, line 182. "…lower explosion limit (LEL) in air of 12.5% v/v at room temperature…".

**Response to RC1:** The text has been changed as suggested by the referee.

**Comments from Referee**:      Page 6, line 189. Remove extra space between "air" and "and", and between "CO" and "in".

**Response to RC1:** This has been corrected as suggested by the referee.

**Comments from Referee**:      Page 6, line 190. Suggest "…between safety requirements, limiting…".

**Response to RC1:** This has been corrected as suggested by the referee.

**Comments from Referee**:      Page 6-7. Effective chain length. This might be good place to discuss experiments to determine the optimum NO concentration for the amplifier chemistry. Also, perhaps near the end of this section, discuss how the effective chain length values are used in the data analysis. In other words, have estimates of the $HO_2/RO_2$ ratio been made and used to apply the two eCL (for $HO_2$ and $HO_2+CH_3O_2$) values? If so, how is it done?

**Response to RC1:** The text of section 3.2.1: Effective chain length has been rewritten to address this criticism. Additional experiments were undertaken to determine the eCL as a function of [NO]. These are reported in figure 7 and also in table 2. The concentration or mixing ratios of $HO_2$ and $RO_2$ are not known in ambient air. Thus the ratio is also not known. The $RO_2^*$ values are reported to be the sum of $HO_2 + \alpha \cdot RO_2$. Using $CH_3O_2$ as a surrogate for all $RO_2$, the values of $\alpha$, which depends on [NO], have been determined by modeling and measurement in Table 2. The relevant paragraph in section 3.1.1 now reads as follows (Lines 302-306):

415     **Author's changes in manuscript**:      "Table 2 summarises the simulated PeRCEAS sensitivity for the $HO_2$ and $CH_3O_2$ detection for different NO mixing ratios in the reactor at 300 mbar. Up to 10 ppm NO ([NO] 7,29 x $10^{13}$ molecules $cm^{-3}$) the difference in sensitivity remains within the PeRCEAS uncertainty. The ratio of the $eCL_{CH3O2}$ /$eCL_{HO2}$ is defined as α. The estimated values of α from modelling and measurements are given in table 2. For the assessment of air masses the measurements of $HO_2 + α·RO_2$, where $αRO_2 ≈ α·CH_3O_2$, are compared with atmospheric model values of $HO_2 + α·RO_2$."

420

    **Comments from Referee**:      Page 6, line 197. "…to the conversion into RO2.". Include that the approach is based on O2 actinometry, as opposed to other approaches reported in the literature (such as N2O actinometry, calibrated NIST photodiodes).

**Response to RC1:** The text has been extended for clarification. Lines 247-258:

425     **Author's changes in manuscript**:      "The eCL of the DUALER reactors is determined in the laboratory by using a calibrated source of peroxy radicals. The latter uses the photolysis of water vapour at 184.9 nm (see Schultz et al., 1995). Briefly, a known water vapour - air mixture is photolysed by a low pressure mercury (Hg) lamp. A nitrous oxide ($N_2O$) absorption filter attenuates the intensity of 184.9 nm radiation. This is achieved by varying the $N_2O/N_2$ ratio in the filter absorption zone. The photolysis of $H_2O$ makes an OH and H. In air, the H reacts with $O_2$ in a termolecular reaction to make $HO_2$.

430 The photolysis of oxygen molecules yield oxygen atoms, O which react with $O_2$ in a termolecular reaction to make $O_3$ (see Reichert et al., 2003). CO is added to the gas mixture in the source to convert the OH into $HO_2$ radicals. As a result, each absorbed photon by a water vapour molecule generates two $HO_2$ molecules. Alternatively, the addition of a hydrocarbon, RH, leads to the conversion of OH to a $RO_2$, and consequently to a 1:1 mixture of $HO_2$ and $RO_2$ for calibration. The concentration of $HO_2$ or $RO_2$, and $O_3$ is thus proportional to the intensity of 184.9 nm electromagnetic radiation. As the absorption coefficient of

435 $N_2O$ (Cantrell et al., 1997) does not change significantly around 185 nm ($σ_{N2O}=14.05×10^{-20}$ $cm^2$ $molecule^{-1}$ at 25 $°C$ with a $0.02×10^{-20}$ $cm^2$ $molecule^{-1}$ $K^{-1}$, temperature dependency), different $HO_2$ and $RO_2$ radical amounts can be produced for a constant $H_2O$ concentration.."

    **Comments from Referee**:      Page 6, line 205. "…are changed stepwise every ten minutes from 8 pptv…". Also,
440 note that too much reagent added to the calibrator has the potential to affect the inlet chemistry. This can be seen by a change in the background with change in radical concentration, which is not expected if the background is mostly due to ozone.

**Response to RC1:** The radical calibration procedure does not require any change in reagents. Different radical mixing ratios are generated by attenuation of the light of the Hg/Ne UV lamp used for photolysis of $H_2O$ and $O_2$ by
445 using different concentrations of $N_2O$ as absorption filter, as described in previous publications (e.g, Reichert et al., 2003). In each step of the calibration both the amplification and the background signal change due to the effect of the light attenuation in the photolysis of $H_2O$ and $O_2$ leading to radicals and $O_3$ respectively.

The text has been extended for clarification:

    **Author's changes in manuscript**:      In line 255 "….The concentration of $HO_2$ or $RO_2$, and $O_3$ is thus proportional to
450 the intensity of 184.9 nm electromagnetic radiation, and the absorption coefficient of $N_2O$ (Cantrell et al., 1997) does not change

significantly around 185 nm ($\sigma_{N2O}$=14.05×10$^{-20}$ cm$^2$ molecule$^{-1}$ at 25 $^{\circ}$C with a 0.02×10$^{-20}$ cm$^2$ molecule$^{-1}$ K$^{-1}$, temperature dependency), different HO$_2$ and RO$_2$ radical amounts can be produced for a constant H$_2$O concentration."

**Author's changes in manuscript:** In line 271: "The O$_3$ generated by the radical source is converted in the DUALER to NO$_2$ by its reaction with NO, which is in excess. Therefore the O$_3$ entering the reactor during the radical calibration is detected as NO$_2$ in the background and amplified signals."

**Comments from Referee:** Page 6, line 206-7. "…is determined from the slope of the measured ΔNO2 levels versus the calculated radical amounts. Example data is shown in…". Suggest rewording the end of the last sentence on this page, since the concentration of NO ***within the inlet*** is 30 ppmv. Perhaps "…and added reagent NO to achieve 30 ppmv within the inlet."

**Response to RC1:** We agree with the comment of the referee but the original figure has been removed.

**Comments from Referee:** Page 7, line 208. Suggest "In Figure 7, the PeRCEAS eCL versus the inlet NO concentration…". In Figure 7, why aren't data shown for lower NO concentrations, such as used by your group in the past and by other researchers?

**Response to RC1:** This figure has been extended for eCL values at lower NO concentrations as suggested.

**Comments from Referee:** Page 7, line 212. "…concentration, eCL values increase with…".

**Response to RC1:** The text has been modified as suggested.

**Comments from Referee:** Page 7, line 16-18. Suggest describing how wall losses were determined. Are they constant or are they affected by the cleanliness of the inlet? Suggest putting all the rate coefficients used for both reactor pressures into Table 1. It is interesting that a level of 3 ppbv O3 was used, presumably because this is what comes out of the calibrator. Suggest also running the model with ambient-like levels of O3.

**Response to RC1:** The wall losses are not determined experimentally. The result of eCL calibrations before and after the measurement campaigns indicate that the effect of cleanliness of the inlet in the eCL is within the experimental error.

The text has been extended (Lines 289-297) and new simulations have been performed for clarification. Table 1 includes now all rate coefficients taken from JPL- Publication 15-10 (Burkholder et al, 2015). The model is initialised with 3ppb O$_3$ because this is the mixing ratio produced during the calibration. Sensitivity studies have shown no significant change in the simulated eCL up to 100ppb O$_3$. This is also confirmed by the experimental values shown in figure 17.

**Comments from Referee:** Page 7, line 221. Inlet pre-chamber is not defined anywhere. This should be shown in the schematic diagram discussed earlier. If the radical losses in the model do not agree with what you think they

arein the DUALER II inlet, suggest you perform experiments to determine what they are. The model of this simple chemistry should be much closer to the observations that a factor of two!

**Response to RC1:** The inlet pre-chamber is now described in detail in section 2. As mentioned in the previous response, the text has been extended for clarification (Lines 280-306) and new simulations have been performed to better reproduce the pre-chamber + reactor configuration in PeRCEAS.

**Comments from Referee**:     Page 7, line 222. Suggest "…shows measured eCL versus modeled CL for the…".

**Response to RC1:** This figure has been replaced.

**Comments from Referee**:     Page 7, line 223-4. "The $CL_{modeled}/eCL_{measured}$ ratio averages about 2 for $HO_2$…". Actually, the ratio is more than 2 for the 200 mB measurements and is about 2 for the 300 mB $HO_2$ measurements. Only the 300 mB $HO_2+CH_3O_2$ measurements are about 1.5. Does this mean that the inlet wall loss changes with reactor pressure? Were the rate coefficients in the model changed to reflect the reactor pressure? Perhaps the model wall loss values should be adjusted based on new laboratory measurements. The chemical amplifier chemistry is simple enough that a box model should be able to accurately reproduce laboratory data such as this. Add error bars to the points to represent total uncertainty in the measured and modeled values. Perhaps perform regressions of data.

**Response to RC1:** As already mentioned, the text has been modified (lines 280-301) and new simulations of the pre-chamber losses have been included. The inlet wall losses are calculated using Eq.4.

$$k_w = 1.85 \left(\frac{v^{1/3}D^{2/3}}{d^{1/3}L^{1/3}}\right)\left(\frac{S}{V}\right) \text{ (Eq.4)}$$

S is the surface area in $cm^2$, V the volume in $cm^3$, L the length and d the diameter of the flow tube in cm, v the velocity of the gas in $cm\,s^{-1}$, and D is the diffusion coefficient, which is calculated to be $D_{HO2}=0.21$ and $D_{CH3O2}=0.14$ in $cm^2\,s^{-1}$. At different pressures the velocity of the gas in the reactor changes and therefore the $k_w$.

The modeling data obtained for 300mbar is now shown in Figure 8. These agree reasonably with the corresponding experimental results.

**Comments from Referee**:     Page 7, line 226. This reviewer does not like the term "titration" in this context, even though it is widely used in the community. Suggest using "conversion" instead.

**Response to RC1:** In the case of PeRCEAS the term titration is used correctly when the conditions are selected to achieve the complete conversion of $O_3$ into $NO_2$ before reaching the detector.

The title of 3.2.2 has been replaced by "conversion of ambient $O_3$ into $NO_2$" and the text in lines 310-316 has been extended for clarification:

**Author's changes in manuscript**:     "As explained in section 3.1. the simultaneous use of two detectors measuring out of phase results in the temporal resolution of the $RO_2^*$ data being 60s. In this way, the horizontal resolution of the PeRCEAS airborne measurements, which depends on the speed and altitude of HALO, is typically between 7 and 15 km. Longer modulation cycles than 120 s result in noisy and  unrepresentative averages for ambient measurements in air masses having significant short term variability of $O_3$ and $NO_2$. To keep the temporal resolution of the $RO_2^*$ data to be equal to the mode time,

 the NO concentration

525     added at the inlet has to be sufficient for a complete titration of the sampled $O_3$ before reaching the detector. "

**Comments from Referee**:     Page 7, line 227. This reviewer disagrees that ozone in the sample has to be completely converted to $NO_2$. Why is this? It seems that partial conversion, as long as it is stable, would be fine.

**Response to RC1:**  We agree with RC1 that the partial conversion of $O_3$ in the sample, as long as stable and identical

530     for both reactor-detector lines of the PeRCEAS is sufficient to determine $RO_2^*$. However the total conversion of $O_3$ in the system enables the quantification of radical data at a 60s temporal resolution as explained in figure 4. Malfunctioning of one of the reactor-detector line yields the $RO_2^*$ to be determined at 120 s. In this case, if $O_3$ is completely converted and the simultaneous $O_3$ and $NO_2$ measurements on board are of sufficient accuracy, a $RO_2^*$ time resolution of 60s may still be feasible.

535

**Comments from Referee**:     Page 7, line 228. Figure 9 has a lot of information that could be presented in a more straightforward way. Suggest plotting the ozone lifetime (or three lifetimes) versus the reactor NO at the two pressures. This could be shown in one plot.

**Response to RC1:** The figure 9 has been changed as proposed by the referee.

540

Comments from Referee:     Page 7, line 229. There is no reason to require conversion of 100-200 ppbv of ozone to 1-2 pptv in the inlet. Conversions of 99% are more than sufficient. Suggest changing this paragraph as Figure 9 is changed.

**Response to RC1:** The text has been changed for clarification (Lines 310-322).

545

**Comments from Referee**:     Page 7, line 230. The wavelength 409 nm is mentioned, but everywhere else it indicates that the lasers operate at 408 nm.

**Response to RC1:** This is a typo and has been corrected.

550     **Comments from Referee**:     Page 7, line 236. Suggest "…which are captured by…". This reviewer disagrees that the radicals and $NO_2$ from PAN-like compounds cancels and does not lead to interference. Yes, the $NO_2$ from the decomposition should be like ambient $NO_2$ and be corrected for by the background measurement. But the radicals formed from the decomposition will amplify and appear like ambient radicals. This is an interference! Suggest rewording this paragraph. Some direct laboratory measurements of the interference would also be helpful. Figure 11

555     shows that the PAN interference is greater at lower reactor pressures. Why would this be the case. At reduced pressure, the decomposition is slower and the time is the reactor is shorter. Suggest checking the modeling.

**Response to the RC1:** The time of reaction of $CH_3CO_2$ produced by the decomposition of PAN with NO is the same in both amplification and background modes. The decomposition of PAN is therefore a potential interference, if PAN decomposes between the upper and lower gas addition points in the reactor working in amplification mode. Taking

560     into account the residence times given in Table 3 and the reactor temperatures showed now in figures 18 and 19, for most operating conditions the potential interference will remain below 2 pptv and can be considered negligible (see also answer to RC2).

The chemistry involved in the formation of $CH_3O_2$ from the PAN decomposition has now been revised, and all the rates have been taken from the recommendations in JPL 15-10, with the equilibrium rate constant from Zhang et al., (2011).

There is no significant difference in the production of $CH_3O_2$ radicals with the pressure. The differences observed in the graph are the result of the conversion of molecules·$cm^{-3}$ in mixing ratios.

The figure has been updated and the text has been modified for clarification. Lines 324-339:

**Author's changes in manuscript**: "Peroxyacyl nitrates ($RC(O)OONO_2$) such as peroxyacetylnitrate, PAN and peroxypropionyl nitrate can decompose thermally inside PeRCEAS. The extent of the decomposition to peroxy radicals and $NO_2$ depends on the time and the temperature. If the thermal decomposition occurs at shorter time scales than the modulation time, they can be a significant interfering source of radicals which are chemically amplified and lead to additional $NO_2$. In a rapidly changing background the $RO_2^*$ determination might be affected according to the temperatures and sample residence times between the gas addition points in the DUALER (Table 3).

To evaluate this effect the production of peroxy radicals from the thermal decomposition of 1 ppb PAN at different temperatures and pressures has been simulated. The results obtained with a box model (Ianini, 2003) including the reactions:

$$CH_3COO_2NO_2 \rightarrow CH_3COO_2 + NO_2 \qquad (R1)$$

$$CH_3COO_2 + NO \rightarrow CH_3 + CO_2 + NO_2 \qquad (R2)$$

$$CH_3 + O_2 + M \rightarrow CH_3O_2 \qquad (R3)$$

are depicted in figure 11. The rate coefficients used are taken from Burkholder et al., (2015).

The [$CH_3O_2$] produced does not vary significantly at the pressures investigated. As the temperature of the PeRCEAS reactors during flight generally remain under 290 K, this source of radicals is considered to be negligible for most operating conditions. The thermal stability of the PAN analogues is similar to that of PAN but they are usually at much lower concentrations than PAN in the atmosphere and also assumed to be a negligible source of error."

**Comments from Referee**: Page 8, line 259-260. It is not clear what is meant by "based on the similarity of the eCL values". Suggest rewording this sentence, and perhaps this entire paragraph to make the message clearer.

**Response to RC1:** It was concluded that the eCL dependency observed was related to the relative humidity and not to the absolute [$H_2O$] because the eCL measured by Reichert et al., (2003) at 20°C and 30°C did not differ within the experimental errors although corresponding to significantly different absolute water concentrations. The text has been modified for clarification. Lines 354-358:

**Author's changes in manuscript**: "The effect of changes in the sampled air humidity on the eCL has been reported and studied by Mihele and Hastie, (1998) and Mihele et al., (1999). Reichert et al. (2003) investigated the dependency of the eCL for ground based measurements at 20 °C and 30 °C and standard pressure, i.e., keeping the relative humidity but almost doubling the absolute water concentration. The obtained eCL values did not differ within the experimental error and confirmed

the dependency of eCL on the relative humidity. All these measurements were performed at a pressure of one atmosphere  and for 3.3 ppmv NO ([NO] 8.12 x $10^{13}$ molecules cm$^{-3}$)."

600  **Comments from Referee**:       Page 8, line 263. Suggest replacing "as shown exemplary" with "with an example shown".

**Response to RC1:** The text has been modified.

**Comments from Referee**:       Page 8, line 265. "…sample humidity do not lead to…".

605  **Response to RC1:** The text has been modified.

**Comments from Referee**:       Page 8, line 267. "…is subject to two types of errors which either are: a) intrinsically…"

**Response to RC1:** The text has been modified as suggested.

610

**Comments from Referee**:       Page 8, line 268. "…in the laboratory, or b) result…"

**Response to RC1:** The text has been modified as suggested.

**Comments from Referee**:       Page 8, line 273. Equation 3 is very similar to Equation 1. Suggest eliminating
615  Equation 3 and referring back to Equation 1 in this discussion. Perhaps change Equation 1 slightly, if needed.

**Response to RC1:** The text has been modified as suggested and Equation 3 has been eliminated.

**Comments from Referee**:       Page 9, line 279. "…Vandaele et al. [2002] with the normalized laser spectrum from the corresponding detector.". Also, "The values obtained have been…"

620  **Response to RC1:** The text has been modified as suggested.

**Comments from Referee**:       Page 9, line 284. "…depicts a sample comparison of spectra…".

**Response to RC1:** The figure has been moved to the supplementary information as suggested by RC2 and the text has been modified. Lines 387-388 :

625  **Author's changes in manuscript**:       "A sample comparison of spectra obtained for the three PeRCEAS detectors is included in the supplementary information (Figure SI-1)."

**Comments from Referee**:       Page 9, line 294. "The effective $\sigma_{NO2}$ obtained agrees within…".

**Response to RC1:** The sentence has been corrected.

630

  **Comments from Referee**:      Page 9, line 297. The tau symbol disappeared. You have a lot of ambient data. Does $\tau_0$ vary significantly as the CRDS cell mirrors are exposed to ambient air?

**Response to RC1:** The $\tau_0$ symbol disappeared during the processing of the document. This is a typo and has been corrected. The variation in the value of $\tau_0$ during the measurement depends on the composition of the air probed. As

635  shown in figure 1, the sample air goes through a 5µm filter before reaching the PeRCEAS detectors. This filter removes most of the ambient particles. As a consequence there is only a gradual change in $\tau_0$ over the 10 hours flight which is not critical for the measurement. The filter is replaced after each flight.

  **Comments from Referee**:      Page 9, line 300. "…measurement requires accurate…"

640  **Response to RC1:** The text has been modified as suggested.

  **Comments from Referee**:      Page 9, line 302-3. "…are the radical calibration…". "…which is estimated to be…".

**Response to RC1:** The source of uncertainty is actually the generation of radicals during the radical calibration.

645

  **Comments from Referee**:      Page 10, line 307. "The errors associated…".

  **Response to RC1:** The text has been modified as suggested.

  **Comments from Referee**:      Page 10, line 312. "…reactor 2, respectively, and…".

650  **Response to RC1:** The text has been modified as suggested.

  **Comments from Referee**:      Page 10, line 315-6. Delete "during the airborne measurement of $RO_2$*".

  **Response to RC1:** The text has been modified as suggested.

655  **Comments from Referee**: Page 10, line 317. "The noise in the $NO_2$ signal is enhanced by…".

  **Response to RC1:** The text has been modified as suggested.

  **Comments from Referee**:      Page 10, line 319. "…cabin temperature could increase…".

**Response to RC1:** The cabin temperature varies depending on the characteristics of the flight. The sentence has been

660  reworded.

**Comments from Referee**: Page 10, line 320. "…stability of the CRDS signal and the accuracy of the supporting measurements."

**Response to RC1:** The text refers to measurements taken before flying to check the overall performance of the instrument. Therefore they are rather "reference" than "supporting" measurements.

**Comments from Referee**: Page 10, line 323-4. Again, this reviewer does not agree with this statement. Since you are continually measuring the background and the amplified signal plus background, variations should be accounted for. Only changes happening faster than one second should have influence, unless there is something about the data analysis that is not obvious from the presentation in the paper. Suggest looking into why step changes in ozone should affect the signal for more than a few seconds. Also, step changes are extreme for ambient measurements. Even changes as the aircraft changes altitude are likely to be gradual unless a pollution layer is encountered.

**Response to RC1:** The PeRCEAS operating conditions has been clarified in text and in answers to previous questions of RC1. These clarifications have already addressed this issue.

**Comments from Referee**: Page 10, line 335. "…a standard deviation of the order of…".

**Response to RC1:** The sentence has been corrected.

**Comments from Referee**: Page 11, line 353. "…detector signals can be significantly affected…".

**Response to RC1:** The text has been changed as suggested.

**Comments from Referee**: Page 11, line 360. "…airborne measurements and is difficult to implement in…".

**Response to RC1:** The text has been changed as suggested.

**Comments from Referee**: Page 11, lines 364-370. Do you mean running the inlet at reduced pressure results in lower eCL values? Doesn't PeRCEAS continually measure the signal and background as mentioned in the second line? What do differences in detector accuracy (do you mean sensitivity) do to affect their uncertainties? Suggest adding a reference to the last sentence of this paragraph (about $RO_2$ interferences in LIF).

**Response to RC1:** As now better described in section 2 the pressure in the inlet is controlled in the pre-chamber during the flight. This pre-chamber is however associated with radical wall losses which reduce the eCL as discussed in 3.2.1. The losses in the pre-chamber depend on the residence time in the pre-chamber. The latter depends on the volume of the pre-chamber and the total flow rates.

The sentence has been extended for clarification. Lines 469-470:

**Author's changes in manuscript**: "The pressure regulation in PeRCA based airborne instruments results in lower eCL than ground based ones. This is attributed to radical losses in the pre-chamber prior to the addition of reagent gases for the radical chemical amplification."

**Response to RC1:** The reference Fuchs et al. (2011) is already cited in the introduction. However, it has additionally been included here as suggested by the referee.

700

 **Comments from Referee**:       Page 12, line 374. "Figure 20 shows sample data of $RO_2$* measured…".

**Response to RC1:** The text has been changed as suggested. The figure 20 is now Figure 18.

 **Comments from Referee**:       Page 12, line 382. Mention that the flagged values are shown in Figures 20 and 23.
705    Also mention this in the figure captions.

**Response to RC1:** The flags are now mentioned in the text and in the corresponding figure captions (now figures 18 and 19) as suggested by the referee.

 **Comments from Referee**:       Page 12, line 386. "…in more detail in Figure 23…".

710    **Response to RC1:** The text has been modified (Lines 483-487).  The corresponding figure is now Figure 19.

 **Comments from Referee**:       Page 12, line 388. Not sure what is meant by "the signal is not affected by altitude changes", since there are jumps in $\Delta NO_2$ when the altitude changes. Suggest rewording to make the point clearer.

**Response to RC1:**  Although $\Delta NO_2$ changes by altitude changes the value of $RO_2$* does not change significantly.
715    The text has been accordingly rewritten. Lines 489-492:

 **Author's changes in manuscript**:       "As can be seen in the figure, pressure fluctuations due to dynamic pressure changes have been reduced by up to 80 % in the improved PeRCEAS. Although the measured $\Delta NO_2$ is affected by altitude changes, the value of the retrieved $RO_2^{*}$ does not change significantly except for the maximum climbing rate directly after take-off.".

720

 **Comments from Referee**:       Page 12, line 402. "…over a 60 s integration…".
**Response to RC1:** The sentence has been corrected.

 **Comments from Referee**:       Page 12, line 404. While PeRCEAS may be suitable for measurements up to 12 km,
725    no data were shown at this altitude. Suggest rewording this sentence.

**Response to RC1:**  The sentence has been reworded. Line 506:

 **Author's changes in manuscript**:       "The performance of the PeRCEAS instrument has been proven to be suitable for airborne measurements during different campaigns onboard HALO."

730    **Comments from Referee**:       Page 12, line 405. "…campaigns onboard HALO.".

 **Response to RC1:**  The sentence has been changed as suggested.

**References**

735
- Line 449. "…peroxy radicals by chemical amplification…".
- Line 452. Two references are together. Need carriage return after "1993".

**Response to RC1:**  The references have been corrected.

**Tables**

740

**Comments from Referee**:      Suggest heading for "second addition point" to be changed to "reaction times", and "to detector" changed to "transfer times".

**Response to RC1:**  The headings have been changed to "reactor residence time" and "total residence time".

745  **Comments from Referee**:      Page 3, Edwards et al., Inlet pressure should be 200 mB.

**Response to RC1:**  This has been corrected.

**Figures**

**Comments from Referee**:      Most of the figures need larger symbols and bolder lines (4, 5, 6, 10, 12, 14, 17, 18,
750  19, 20, 23). In many of the plots, the legend is covered by data. Suggest enclosing legend in a box with a white background.

**Response to RC1:**  This has been corrected.

**Comments from Referee**:      See comments earlier about Figures 1-3.
755  **Response to RC1:**  These figures have been removed and replaced as suggested.

**Comments from Referee**:      Figure 2 caption. "…Top view of the…". "..the laser beam is highlighted (purple) for…". "…exiting the cavity is depicted."

**Response to RC1:**  This figure has been removed and replaced as suggested.

760

**Comments from Referee**:      Figure 4. Change y-axis to Allan Variance. Describe what the lines depict (linear fits to data less than 10 seconds?).

**Response to RC1:**  This figure has been removed and replaced as suggested. A subplot has been added to show the detector stability for 40 minutes measurement by detector FH as an example. These data are used for the Allan
765  variance study of the corresponding detector.

**Comments from Referee**:      Figure 5. It is not obvious what this figure is trying to show. It appears to this reviewer that the point is temperature changes affect the $\tau_0$ of the detector, but the retrieved $\Delta NO_2$ is affected very

little. Why not do this experiment with two detectors as done for radical measurements? This would be a more realistic representation of the actual measurement situation.

**Response to RC1:** The objective of this experiment is to illustrate the effect of temperature drifts in the $\Delta NO_2$ at the detector. Since both $NO_2$ detectors are identical in the operating conditions, it is sufficient to show the results obtained with one detector. The text has been extended for clarification

**Comments from Referee**: Figure 6. Are the equations determined from linear fits? Are they standard or bivariate fits?

**Response to RC1:** The equations are determined from linear and bivariate fits, considering the errors in x and y axis. This plot is now placed in the supplementary information.

**Comments from Referee**: Figure 7. Why are not data shown for lower values of NO? Suggest more work going from 0 to 3.5E14 NO with at least 10 points for each instrumental condition (pressure and radical type). Perhaps also show the same y-axis for both plots.

**Response to RC1:** The data series have been extended as suggested.

**Comments from Referee**: Figure 8. Since it is mentioned in the text, suggest adding 1:1.5 line. As discussed earlier, perhaps more modeling with more realistic wall loss rates needs to be done.

**Response to RC1:** The text has been extended and the results of additional modeling have been depicted in a new figure.

**Comments from Referee**: Figure 9. Changes to this figure suggested earlier (plot $O_3$ lifetime versus NO). If it is kept the same, suggest labeling each sub-figure and referring to those labels in the caption.

**Response to RC1:** The figure has been changed as suggested.

**Comments from Referee**: Figure 10. This shows that 60% of the $O_3$ is converted with NO of 3 ppmv, and 90% at 6 ppmv. This means that the instrument could be run with much lower NO levels.

**Response to RC1:** This issue has been extensively discussed in previous answers.

**Comments from Referee**: Figure 11. It is stated that PAN interference is not a problem with PeRCEAS, but this plot shows that with reaction times of 3 seconds (compared to 2.6 to 3.1 seconds for the two DUALER inlets), up to 6 pptv of peroxy radicals can be produced from 1 ppbv of PAN. Is this representative of the conditions for which the instrument has been used? This figure could be changed to plot the fraction conversion of PAN ($CH_3O_2$ produced / PAN) versus temperature for the two DUALER reaction times.

**Response to RC1:** As mentioned before, the simulations have been revised and the plots updated. The ratio $CH_3O_2$ produced/PAN is $<10^{-3}$ for the range of temperatures expected in the reactors, which are in the outer part of the fuselage. The maximum interference expected remains below 1-2 pptv for the transition times given in Table 2.

**Comments from Referee**: Figure 13. There are places where the ambient water is below the inlet water. How can this be? This figure could be changed to plot inlet H2O versus ambient H2O with the points colored by altitude.

**Response to RC1:** The figure has been changed as suggested by the referee. The relative humidity sensor at the inlet has been re-calibrated. After applying the calibration correction all the above mentioned values of inlet humidity lower than the ambient humidity have disappeared. The plot has been updated.

**Comments from Referee**: Figure 14. How can the lack of dependence on water vapor be explained, given that it is a purported to be related to one of the amplification chemistry reactions (HO2+NO)? Perhaps modelling of these data would be instructive. Also, suggest showing data down to the lowest water values possible.

**Response to RC1:** The dependency on water vapour only becomes significant at higher water number concentrations for NO 45 ppm at 300 mbar as shown in figure 14. New measurements at 10 and 30 ppmv NO at 300 mbar show a clear dependency with $[H_2O]$ which has to be taken into account in the analysis of ambient air measurements at these operating conditions. The water vapour dependence of eCL decreases significantly from 10 ppm to 45 ppm. For water vapour concentrations in the range of 4 -14 x $10^{16}$ molecule $cm^{-3}$ of relevance for HALO flights, this results in the water vapour impact on the eCL being negligible. This is shown more explicitly in the new figure 14 in the manuscript.

As described in Hastie et al., (1991) and Reichert et al., (2003), the chain length (CL) of a PERCA reactor can be expressed using a resistance model as follows:

$$\frac{1}{CL} = \left(\frac{1}{CL_{HO_2}} + \frac{1}{CL_{OH}}\right)$$

$$\frac{1}{CL} \approx \frac{1}{P_{NO_2}} \cdot \left(\frac{\sum HO_2 \text{ removal rates}}{\sum HO_2 \text{ propagating rates}} + \frac{\sum OH \text{ removal rates}}{\sum OH \text{ propagating rates}}\right)$$

And if only the predominant processes are considered, as:

$$\frac{1}{CL} \approx \frac{1}{P_{NO_2}} \cdot \left(\frac{k_{w_{HO_2}}}{k_{NO+HO_2}} + \frac{k_{w_{OH}} + k_{HONO} \cdot [NO]}{k_{CO+OH} \cdot [CO]}\right)$$

$P_{NO_2}$ is the probability of radical conversion into $NO_2$, $k_{w_{HO_2}}$ and $k_{w_{OH}}$ are the wall losses of $HO_2$ and OH respectively, $k_{NO+HO_2}$ is the rate constant for $NO_2$ production from $HO_2$ + NO reaction, $k_{HONO}$ is the rate constant for HONO formation from OH + NO reaction, $k_{CO+OH}$ is the rate constant for $HO_2$ formation from OH + CO reaction. In the presence of $H_2O$, $HO_2$ water clusters are formed as postulated by Reichert et al., (2003).

Our interpretation of these novel results is that as the NO is increased from mixing ratios of 10 to 45 ppm at 300 mbar, the $CL_{OH}$ is reduced, because the rate of the chain termination termolecular reaction of OH with NO making HONO increases, and the $CL_{HO_2}$ is increased, because the rate of the propagation reaction between NO and $HO_2$ increases. As a result the CL begins to be dominated by the $CL_{OH}$, which is independent of water vapour.

The new series of measurements are now included in Figure 14 and the text has been extended for clarification (Lines 359-371):

**Author's changes in manuscript**: "In this work radical mixtures were sampled at 25 °C for relative humidity between 2 % and 25 %. This leads to a ca. 20 times increase in the absolute [$H_2O$]. These conditions cover the [$H_2O$] expected for a larger T range (-20 – 30 °C) during airborne measurements in the free troposphere at 200 and 300 mbar inlet pressures. Figure 13 shows the [$H_2O$] in the air probed versus the [$H_2O$] in the inlet for real measurements on board of the HALO aircraft. The results in figure 14 for 45 ppm NO ([NO] $3.28 \times 10^{14}$ molecules $cm^{-3}$ at 300 mbar) indicate that variations in the sample humidity do not lead to additional uncertainty in the $RO_2^*$ retrieval as the PeRCEAS eCL remains invariable within the experimental error up to [$H_2O$] ~ $1.4 \times 10^{17}$ molecules $cm^{-3}$. In contrast, for 10 ppm and 30 ppm NO in the reactor ([NO] $7.29 \times 10^{13}$ and $2.19 \times 10^{14}$ molecules $cm^{-3}$ at 300 mbar) the eCL shows a clear dependency with the ambient [$HO_2$]. The comparison with the eCL values obtained by Reichert et al (2003) at 1 atmosphere indicate a eCL dependency on [$H_2O$], temperature and pressure having a different pattern for 45 ppm NO in the reactor. This is explained by invoking the competition in the amplification chain length, CL, between $HO_2$ and OH removal rates, as explained in Hastie et al., (1991) and Reichert et al., (2003). At [NO] ~ $3.28 \times 10^{14}$ molecules $cm^{-3}$ the CL begins to be dominated by the rate of the termination termolecular reaction of OH with NO, which is independent of water vapour. This eCL dependency has to be taken considered in the analysis of ambient air $RO_2^*$ measurements."

**Response to RC1:** The [$H_2O$] values showed are the lowest water values experimentally possible with the used set up.

**Comments from Referees**: Figure 15. Can the laser emission be adjusted so all three detectors peak at the same wavelength? This would definitely help them to behave more similarly. Suggest changing the right-hand y-axis to go from 0 to10, and to average the cross-section data to a lower resolution, say 0.1 nm, to make the plot message clearer.

**Response to RC1:** The laser emissions cannot be changed. They are multimode diode lasers with fixed emission at a particular measurement condition. The plot has been modified as suggested and moved to the SI as suggested by RC2.

**Comments from Referee**: Figure 16. Not sure this figure is necessary, since the data are plotted in Figure 17.

**Response to RC1:** This figure has been moved to the SI.

**Comments from Referee**: Figure 19. change caption "..while changing $O_3$…", "…in the source with estimated 15%...". It is not clear why the perturbations to $\Delta NO_2$ last so long (up to 40 seconds) when the background should be measured on the 1 second time scale. Is this a data processing issue? Suggest checking why this is the case.

**Response to RC1:** The $\Delta NO_2$ from each detector is calculated from individual ring down times. Consequently, variations in the background affect the calculated $\Delta NO_2$. This effect is cancelled out when averaging the $\Delta NO_2$ of both detectors to retrieve $RO_2^*$ which consequently remains unaffected. The variation of $O_3$ from the calibrator is not instantaneous but takes around 1 minute to become stable. This is the reason why ca. three $RO_2^*$ values are affected by each change in $O_3$.

**Comments from Referee**: Figure 20. There are big swings in the inlet pressure even when the altitude is not being changed. Why is this?

**Response to RC1:** The figure has been modified for clarification. The pressure spikes are related to dynamical pressure changes in the inlet, e.g. caused by changes in velocity of the aircraft and air turbulences.

**Comments from Referee**: Figures 21 and 22. These figures could be eliminated

**Response to RC1:** These figures have been removed as suggested by both referees.

**Response to RC2**

**General comments:**

**Comments from Referee**: Overall the reviewer believes 23 figures is too many for an instrument development paper of this nature. An instrument schematic could replace the first three figures (photos of the instrument, inlet and inside of aircraft) similar to Horstjann et al 2014 figure 1.

**Response to RC2**: The first three figures have been replaced by instrument schematics as suggested by the referee.

**Comments from Referee**: The authors referred to an improved inlet design (DUALER vs DUALER II) by modifying the pre-chamber design and reducing wall interaction in the inlet. This modification seems significant and likely affects the instrument performance more than discussed in this paper. A figure comparing the two inlet designs or the changes in inlet design would be useful.

**Response to RC2**: A new figure (Figure 2) showing the changes to the pre-chamber and reactors in the DUALER has been included as suggested by the referee. In addition the text has been rewritten (Lines 143-151):

**Author's changes in manuscript**: "In the DUALER, a stable pressure in the pre-chamber is achieved by a pressure regulator, which controls the flow through the bypass line. As noted the flow rate through the reactors is held constant during measurements. Consequently, when the outside air pressure changes, the bypass flow rate from the pre-chamber is changed. The outer dimensions, shape, form and weight of the DUALER are constrained by the inlet pylon in use with the research aircraft HALO. After the first version of the DUALER (from now on called DUALER I) was flown, the inner dimensions of the pre-chamber were further optimised to reduce the wall losses and turbulence in the pre-chamber. For this, in the DUALER II the volume of the pre-chamber was increased by extending its vertical extent, the length of the truncated cone on top of the reactors was reduced in 3 mm, and the volume of the reactors was increased to 130.5 ml from the 112 ml in DUALER I. These changes resulted in a higher eCL and improved pressure stability in DUALER II as compared to DUALER I. Figure 2 shows the upper part of both DUALER I and DUALER II."

**Comments from Referee**: The general description of how the inlet operates (alternating measurement modes) is somewhat confusing and is evident in the Reviewer #1's comments. A time series figure of the operation of each channel would make this clearer (ie switching from amplification mode to background mode and showing how each channels mode switching is out of phase with each other).

**Response to RC2**: A new figure (Figure 4) has been added showing the ring down times of the two detector, the retrieved $\Delta NO_2$ and $HO_2$ for a laboratory calibration of $HO_2$. The text has been rewritten (Lines 127-142):

**Author's changes in manuscript**: "Briefly, sampled air enters PeRCEAS through the DUALER pre-chamber, which is at a lower pressure than that outside of the HALO, through an orifice in a truncated cone, i.e. a nozzle. From this pre-chamber the air is pumped simultaneously through the two flow reactors and a bypass line. At the upper addition point a mixture of CO or $N_2$ and NO enters each reactor. At the lower addition point, a flow of $N_2$ or CO enters each reactor. This enables the CO and $N_2$

flows in the two reactors within the DUALER to be switched simultaneously but out of phase with one another from the upper to the lower addition point. At the addition points, the reagent gases enter the reactor through eight circular distributed 1 mm holes to facilitate the rapid mixing with the sampled air. During measurements, the pressure in the pre-chamber and both reactors is held constant. However, there is a small pressure fluctuation during the switching of flows between the upper and lower mixing point. The flow passing through each reactor enters a CRDS $NO_2$ detector. Afterwards, the sample flows together with the air from the bypass line are scrubbed for CO and $NO_x$ and, exhausted by the pump.

The DUALER inlet comprises two PeRCA chemical reactors having alternating measurement modes, which are out of phase with one another. During the first part of the measurement cycle, the first reactor and detector are in amplification mode, while simultaneously the second reactor and detector are in background mode. In the second part of the cycle, the CO addition point in both reactors is switched. Consequently, the first reactor and detector are then in background mode while the second reactor and detector are in amplification mode. In the analysis of the measurements, the amplification and background signals from both detectors are combined appropriately. This improves accuracy and temporal resolution of the resultant $RO_2^*$ data set (see 3.1). "

**Comments from Referee**:    Furthermore, a detailed description of how the mixing ratio of NO was decided on (30 ppmv) would be useful, as it differs significantly from the DUALER I (6ppmv) inlet and other groups PeRCA inlets (0.9 to 7.7 ppmv).

**Response to RC2**: The objective of the present publication is to explain the dominant factors affecting the overall performance and accuracy of PeRCEAS for the determination of the hydroperoxyl, $HO_2$, and organic peroxy radicals, $RO_2$, which react with NO to form $NO_2$, when deployed on the the HALO aircraft. The operating conditions of PeRCEAS are optimised for the specific sampling position used, cabin location of the instruments, safety requirements and the type of flight tracks and altitude profiles which were flown by HALO. Generally, these limitations are often different in different campaigns. For this reason, this manuscript does not aim at describing a unique universal set of PeRCEAS operating conditions.

**Comments from Referee**:    Generally speaking the flight data section of the paper should be focused on the improved performance of the instrument rather than flight tracks and mixing ratio figures. A comparison of DUALER I and DUALER II flight data is recommended. Considering how to show improvements between DUALER II deployments is recommended

**Response to RC2**: The performance of DUALER I and DUALER II are now compared in section 5. In addition some key improvements of performance of DUALER II compared with DUALER I are highlighted. Lines 488-493:

**Author's changes in manuscript**:    "Two hours of measurements from the flight on the 19.03.2018 are shown in Figure 19 as an example of the third airborne deployment of PeRCEAS within the EMeRGe campaign in Asia. As can be seen in the figure, pressure fluctuations due to dynamic pressure changes have been reduced by up to 80 % in the improved PeRCEAS. Although the measured $\Delta NO_2$ is affected by altitude changes, the value of the retrieved $RO_2^*$ does not change significantly except for the maximum climbing rate directly after take-off. Furthermore, the beam camera and the motorised mirror mounts enable the identification and immediate correction of small misalignments. This improves significantly the instrumental performance while simplifying maintenance."

**Specific comments:**

**Comments from Referee**: Page 4, line 120, "The optical cavity remains similar to that described in Horstjann et al…" It is useful to include mirror specifications (substrate, coating, reflectivity, diameter, etc) for a CRDS instrument, as they are critical part of theoretical instrument performance. Does the piezo optical alignment system run in a closed loop control with beam profile as a feedback parameter? If so describe this, as it seems novel.

**Response to RC2:** The piezo optical alignment system was used in the previous PeRCEAS configuration using a single mode laser as reported by Horstjann et al., (2014). The actual PeRCEAS detector described here, has a multimode diode laser. The alignment is done manually using motorized mirror mounts while the beam camera confirms the TEM$^{00}$ mode. The beam profile is checked manually in a regular basis but is not a feedback parameter for the alignment of the mirrors.

The text has been extended. Lines 153-155:

**Author's changes in manuscript**: "The optical cavity remains similar to that described in Horstjann et al., (2014), i.e., a V-resonator of ca. 100 cm$^3$ volume formed between glued highly reflective mirrors (reflectivity, R = 99.995 %, diameter, d = 0.5", radius of curvature, roc = 100 cm, AT Films, USA) on the side of a Teflon coated aluminium cuboid."

**Comments from Referee**: Page 5, line 146, "mode and modulation times…" it is not clear what mode and modulation times refer to, might be useful to define them discretely

**Response to RC2:** The text has been extended for clarification at the beginning of 3.1. (Lines 181-190) and Figure 4 has been added:

**Author's changes in manuscript**: "The mode time is defined as the time selected for the measurement in either amplification or background mode. The modulation time is the time taken for a complete measurement cycle, which comprises the sum of one amplification and one background mode. The PeRCEAS measurement cycle is illustrated in Figure 4. The $\Delta NO_2$ for each detector is calculated from the ring down time of two consecutive modes using Eq.. If the mode time is adequately selected, the $RO_2^*$ retrieved per measurement cycle is identical in both measurement lines, as the two reactors are operated out of phase with one another. The final $RO_2^*$ data is calculated as the mean of the $RO_2^*$ determined from the $\Delta NO_2$ and eCL of both detectors for a given measurement cycle. The time resolution of the $RO_2^*$ measurement is then equal to the mode time. After switching modes, a small pressure pulse leads to an oscillation of the $NO_2$ signal. Consequently, the first 20 s of each mode are not used in data analysis. The time lag arising from the time taken for the sample flow between the CRDS detector and the point of switching is typically less than 8 s (see Table 3)."

**Comments from Referee**: Page 5, line 160, "…detector temperature. For this different detector temperature gradients, $\Delta T$, where applied to modulated signals generated by varying the sampled NO2 concentration…" it's not clear why the investigators modulated NO2 while applying a temperature gradient to the detector. Would it not be easier to interpret if a constant mixing ratio gas was sampled while applying a temperature gradient? It is not clear from the text where this temperature gradient is and how it was applied. It would be useful to readers that are not familiar with optics, on why a temperature gradient of 7 degC would cause detector instability. It is also not clear

990 from the text, what was done to address this flaw in the detector design, as the authors state earlier detector stability is paramount in overall instrument performance.

**Response to RC2:** A modulated signal was applied in order to study the effect on similar signals to those measured by PeRCEAS during operation i.e., a $\Delta NO_2$ between amplification and background modes. The text has been extended for clarification. Lines 211-221:

995 **Author's changes in manuscript**: "Temperature changes of the detector affect: i) the diode laser emission, both its amplitude and wavelength, and ii) the mode matching between laser and detector, and consequently the $\tau_0$. The effect of the variations in $\tau$, resulting from changes in room or HALO cabin air temperatures, on the accuracy and precision of the $\Delta NO_2$ determination was investigated by a series of laboratory experiments. For this, modulated concentrations of $NO_2$ in the flow were generated. This was achieved by alternating between two selected $NO_2$ concentrations once per minute. The temperature of the

1000 CRDS detector, T, and $\tau$ were then measured. Detector temperature gradients over a time t, i.e., $\Delta T/\Delta t$, determined by the temperature within the detector housing close to the photodiode, were induced by controlled changes in the room temperature.

Figure 6 shows the effect of introducing temperature perturbations in a modulated $NO_2$ signal between 11.5 and 12.1 ppbv measured at 200 mbar and 23 °C. As can be seen in the figure, a temperature perturbation affects both precision and accuracy of the retrieved $\Delta NO_2$. For temperature gradients up to $\Delta T/\Delta t \approx 7$ °C h$^{-1}$ the experimental precision of the $\Delta NO_2$ determination

1005 remains within ($2\sigma$) 150 pptv (= 7.3 x $10^8$ molecules cm$^{-3}$ at 200 mbar and 23°C). "

Concerning T stability as states in 4.2, Lines 406-407:

"Generally, in-flight variations in the HALO cabin temperature affect minimally the accuracy of the $RO_2^*$ determination. "

**Comments from Referee**: Page 6, line187, "…of the sampled $O_3$ by NO to form $NO_2$ also depends on the
1010 concentration of NO added to the sample flow and the time for reaction before reaching the detector.". This would be a good place to discuss how 30 ppmv NO was decided on for a reagent mixing ratio and discuss flow rate choices for both NO and CO.

**Response to RC2:** The purpose of this section is to present the effect of changing gas concentration and flows. A
1015 balance between different and competing effects, leads to a selection of the optimal operating conditions for a specific measurement campaign. 30 ppm NO has just been selected for some of the testing series in this work as an example for suitable reagent mixing ratio. The paper has been revised and additional measurement series have been included to illustrate some of the effects investigated.

1020 **Comments from Referee**: Page 6, line 192, "3.2.1 Effective Chain Length…", This section seems to describe a well established method documented in literature. The reviewer recommends shorting the description of the method and explain better the difference in DUALER I and DUALER II eCL.

**Response to RC2:** This part of the text has been modified taking into account the comments of both RC1 and RC2.

1025     **Comments from Referee**:      Page 7, line 216, "The model was initialized with 9% CO, 3 ppb $O_3$, 50 pptv $HO_2$…" why was 3 ppb $O_3$ determined to be a representative mixing ratio for ozone? I may be misunderstanding the inlet chemistry, but it seems like missing 30 ppmv of NO would significantly affect the modeled CL. Assuming the box model initialization is correct, would it not be useful to vary the wall loss rate constants to match the eCL and determine if this wall loss is reasonable? It would also be useful to experimentally
1030     determine the wall loss of the inlet.

**Response to RC2:** The model is initialised with 3 ppbv $O_3$ because this is the mixing ratio produced during the calibration. Sensitivity studies have shown no significant change in the simulated eCL up to 100ppb $O_3$. This is also confirmed by the experimental values shown in figure 17.

Following the recommendation of RC2, the simulations have been extended varying the wall losses in the pre-
1035     chamber to match the experimental values, as now explained in the text (Lines 280-306). PeRCEAS is operated at low pressures and the pressure regulation during operation prevents the direct experimental determination of the wall losses in the pre-chamber. However these are estimated by comparison with the model used.

    **Comments from Referee**:      Page 7, line 222, "figure 8 shows eCL vs CL", the authors should include error bars
1040     on these data.

**Response to RC2:** The error bars on the experimental data are now included and the figure has been extended with more experimental and model data.

    **Comments from Referee**:      Page 7, line 228, "Figure 9 depicts the $O_3$ decay simulated for 100 to 200 ppb…"
1045     these figures are somewhat confusing to the reviewer. One could take the 99% conversion time for each NO mixing ratio curve and plot all 4 conditions (ie pressure and $O_3$ mixing ratio) on 1 figure for varying NO mixing ratio. Additionally, adding an inlet residence time reference line would be useful for helping the reader visualize what time limit you have on this reaction.

**Response to RC2:** The figures have been modified as suggested by the referee.

1050

    **Comments from Referee**:      Page 7, line 234, "PAN and PPN thermal decomposition", the reviewer believes that experimental work is justified to confirm 'this source of radicals is considered to be negligible'.The box modelling done for CL prediction was shown to not capture the actual inlet system, so it's not clear why it would do a better job with modelling PAN and PPN. Figure 11 shows up to 10 pptv interference, this does not seem negligible to the
1055     reviewer.

**Response to RC2:** The discrepancies between measured eCL and simulated CL for the reactor are attributed to errors in the estimate of the wall losses in the inlet pre-chamber and not to the overall performance of the box model. The chemistry involved in the formation of $CH_3O_2$ from the PAN decomposition has now been revised, and all the rates have been taken from the recommendations in JPL 15-10, with the equilibrium rate constant from
1060     Zhang et al (2011).

The results do not change for NO mixing ratios in reactor within 10 and 45 ppm.

During the flight the PeRCEAS reactors are located inside the pylon on the upper part of the HALO fuselage. As a consequence, the temperature in the reactors, which is measured, remains below 290K (Figure 20 and 21). Consequently the maximum interference expected for the transition times given in Table 3 will be below 2 pptv, which is within the measurement error. The figure (now Figure 11) has been updated and the text has been extended for clarification (Lines 324-339).

**Comments from Referee**:     Page 8, line 252, "Figure 12 shows the variation of the eCL for 45 ppm NO within a pressure
range…", The reviewer does not understand why this experiment was done with 45 ppm NO when the decided upon mixing ratio of NO addition seems to be 30 ppm NO for the rest of the paper. If this is a typo, it should be corrected, if not the experiment should be done at the actual mixing ratio the instrument is operated at.

**Response to RC2:**  The figure has been extended with similar measurements for 10 ppm NO. The results at 10 ppm NO confirm the recommendation of $\Delta P$= 100 mbar as the minimum operating pressure. The text (Lines 348-353) and the figure (now Figure 12) have been updated.

**Comments from Referee**:     Page 9, line 284, "Figure 15 depicts exemplary a comparison of spectra…", the reviewer does not believe including $NO_2$ absorption cross section and detector spectra is a useful figure for the main text of this paper. Remove or include in the SI.

**Response to RC2:**  The figure has now been included in the supplementary information. Line 387:

**Author's changes in manuscript**:     "A sample comparison of spectra obtained for the three PeRCEAS detectors is included in the supplementary information (Figure SI-1)."

**Comments from Referee**:     Page 9, line 289, "In addition, the effective $\sigma_{NO2}$ can be calculated by sampling known mixtures…", the reviewer does not believe including a time series of calibration gas addition to instrument is a useful figure for the main text of this paper. Remove or include in the SI.

**Response to RC2:**  The figure has now been included in the supplementary information. Lines 392-396:

**Author's changes in manuscript**:     "The result of applying Eq. (5) to the PeRCEAS detectors is depicted in Figure 15. The detectors sampled known mixtures of $NO_2$ from commercial gas cylinders in synthetic air at 200 mbar as shown in the supplementary information (Figure SI-2)."

**Comments from Referee**:     Page 9, line 294, "The result of apply Eq. 4 to the PeRCEAS detectors at 200 mbar is                                                                depicted

in Figure 17." It is more common to plot $NO_2$ number density [molecules/cm3] vs. $\alpha$, as the slope has the physical meaning of the absorption cross section of $NO_2$.

**Response to RC2:** In the case of CRDS $\alpha$ is not measured but must be calculated by using Eq.1. In contrast, the variables plotted in the Figure 15 (originally Figure 17) are taken directly from the measurement and do not require a prior knowledge of $\tau_0$. In addition the y intercept of the plot is the $1/\tau_0$ for each detector.

**Comments from Referee**: Page 9, line 302, "The main source of uncertainty…" the authors previously mention detector drift due to temperature changes (figure 5), is this not a significant source of uncertainty as well?

**Response to RC2:** Temperature drifts can be minimised during laboratory measurements. Concerning in-flight measurements, it is mentioned in section 4.3 that the temperature in the HALO cabin remains reasonably constant and therefore is not a significant source of uncertainty during the airborne measurement. An example can be seen in Figures 18 and 19. A sentence has been added to clarify this point. Line 406:

**Author's changes in manuscript**: "Generally, in-flight variations in the HALO cabin temperature affect minimally the accuracy of the $RO_2^*$ determination."

**Comments from Referee**: Page 10, line 309, "Figure 18 shows the calculated eCL from 14 radical calibrations…." Why were radical calibrations done with a NO mixing ratio of 45 ppm when the instrument is run at 30 ppm?

**Response to RC2**: In this publication there is no intention to define one unique set of operating conditions for the PeRCEAS on HALO. Several concentrations and mixing ratios for NO were investigated. At 300 mbar the longest series of measurements was selected in the figure 18 to show the best statistics in the reproducibility and stability of eCL over time. These measurements were carried out to investigate the dependence of eCL on NO. Figure 18 (now Figure 16 in the revised version) is been replaced to show the data obtained at 300 mbar and 30 ppmv as suggested by the referee. The estimates of the corresponding eCL values for 10 ppm, 30 ppm and 45 ppm NO are now listed in Table 4.

**Comments from Referee**: Page 10, line 330, "As can be seen in Figure 19…", it would be useful to plot actual $O_3$ mixing ratio rather than set point in the top panel of this figure. It would also be useful to plot chamber pressure or channel pressure, as it seems like when the $O_3$ mixing ratio is changed the $NO_2$ signal displays a large amount of noise (50 ppb). It is also unclear what D1, D2, SG and BG stand for in this figure.

**Response to RC2**: The original figure 19 is now Figure 17. The $O_3$ mixing ratio is set at the calibrated $O_3$ generator and cannot be measured during the PeRCEAS measurement, as it is converted in $NO_2$.

The variation observed in the $\Delta NO_2$ (the $NO_2$ signal is not plotted) is not related to any change in the pressure but to the change in the $O_3$ concentration in the sample, which takes approximately 1 minute to stabilise after changing the set point in the ozone generator.

The meaning of D1, D2, AP and BG are now explained in the figure caption. SG has been replaced by AP for clarification.

1135   **Comments from Referee**:    Page 11, line 364, "The pressure regulation in PeRCA based airborne instruments results in lower eCL than ground based ones." This statement should be substantiated. The reviewer sees a range of eCL for aircraft instruments from 45 to 322 and ground based instruments a range of 91 to 1010. A description of why lower pressure or pressure regulation in general affects eCL would be useful.

1140   **Response to RC2**: As now described in section 2 the pressure in the inlet in PeRCEAS is controlled in the pre-chamber during the flight. This pre-chamber prior to the reactors causes radical losses which do not have the ground based instruments without pressure chamber and pressure regulation.  The sentence has been extended for clarification. Lines 469-470:

  **Author's changes in manuscript**: "The pressure regulation in PeRCA based airborne instruments results in lower eCL than
1145 ground based ones. This is attributed to radical losses in the pre-chamber prior to the addition of reagent gases for the radical chemical amplification."

  **Comments from Referee**:    Page 11, line 367, "the detection limit and uncertainty of PeRCA based instruments are strongly depending on the variation of $O_3$ and $NO_2$ in the sampled air mass…" The reviewer believes this
1150 statement should be substantiated with uncertainty and/or detection limit analysis to show the reader the magnitude of this affect. If the changes in $O_3$ and $NO_2$ are measured (as often are in aircraft field campaigns) can a correction method not be proposed and evaluated?

  **Response to RC2**: The effect of $O_3$ changes in the $RO_2^*$ determination has been shown and quantified in figure 17. Simultaneous measurements of $O_3$ and $NO_2$ onboard can be used as a reference for the correction of the
1155 background variations in individual cases, but a method that relies on other measurements to correct the background will have to deal with different instrumental resolutions and sources of errors of the instruments involved.

  **Comments from Referee**:    Page 12, line 379, "As can be seen in figure 20…" this reviewer believes it would be
1160 useful to add the detector temperature (or deltaT used earlier) to this figure, as this was determined to be a large effect on $\Delta NO_2$ earlier in the manuscript. If $NO_2$ and $O_3$ mixing ratio data is available from the flight, this would be useful to include as well.

  **Response to RC2**: The figure 20 (now figure 18) has been modified to include the inlet and detector temperatures as suggested by the referee.

1165

  **Comments from Referee**: Page 12, line 383, "Figure 22…" the reviewer does not believe this or figure 21 add considerable value to this paper. Recommend removing or moving to SI.

**Response to RC2**: This figure has been removed as suggested by the referee.

1170    **Comments from Referee**:     Page 12, line 387, "…illustrate the improvement in the dynamical stability achieved in successive airborne deployments…" the reviewer finds it difficult to see the improvements made in the measurement by looking at time series data. Suggest thinking of a different way of presenting this conclusion. It would also be very useful to add a comparison to the previous generation of instrument somewhere in the paper.

1175    **Response to RC2**: The figures 18 and 19 have been modified to make clearer the improvement made as suggested by the referee.